# A STATISTICAL FRAMEWORK FOR PERSONALIZED FEDERATED LEARNING AND ESTIMATION: THEORY, ALGORITHMS, AND PRIVACY

**Kaan Ozkara**\*, **Antonious M. Girgis**\*, **Deepesh Data & Suhas Diggavi**
Department of Electrical and Computer Engineering
University of California, Los Angeles
`{kaan,amgirgis}@ucla.edu,deepesh.data@gmail.com,suhas@ee.ucla.edu`

## ABSTRACT

A distinguishing characteristic of federated learning is that the (local) client data could have statistical heterogeneity. This heterogeneity has motivated the design of personalized learning, where individual (personalized) models are trained, through collaboration. There have been various personalization methods proposed in literature, with seemingly very different forms and methods ranging from use of a single global model for local regularization and model interpolation, to use of multiple global models for personalized clustering, etc. In this work, we begin with a statistical framework that unifies several different algorithms as well as suggest new algorithms. We apply our framework to personalized estimation, and connect it to the classical empirical Bayes' methodology. We develop novel private personalized estimation under this framework. We then use our statistical framework to propose new personalized learning algorithms, including `AdaPeD` based on information-geometry regularization, which numerically outperforms several known algorithms. We develop privacy for personalized learning methods with guarantees for user-level privacy and composition. We numerically evaluate the performance as well as the privacy for both the estimation and learning problems, demonstrating the advantages of our proposed methods.

## 1 INTRODUCTION

The federated learning (FL) paradigm has had huge recent success both in industry and academia (McMahan et al., 2017; Kairouz et al., 2021), as it enables to leverage data available in dispersed devices for learning while maintaining data privacy. Yet, it was recently realized that for some applications, due to the statistical heterogeneity of local data, a single global learning model may perform poorly for individual clients. This motivated the need for personalized learning achieved through collaboration, and there have been a plethora of personalized models proposed in the literature as well (Fallah et al., 2020; Dinh et al., 2020; Deng et al., 2020; Mansour et al., 2020; Acar et al., 2021; Li et al., 2021; Ozkara et al., 2021; Zhang et al., 2021; Hu et al., 2020). However, the proposed approaches appear to use very different forms and methods, and there is a lack of understanding of an underlying fundamental statistical framework. Such a statistical framework could help develop theoretical bounds for performance, suggest new algorithms as well as perhaps give grounding to known methods. Our work addresses this gap.

In particular, we consider the fundamental question of how one can use collaboration to help personalized learning and estimation for users who have limited data that they want to keep private. Our proposed framework is founded on the requirement not only of personalization but also privacy, as maintaining local data privacy is what makes the federated learning framework attractive - and thus any algorithm that aims to be impactful needs to also give formal privacy guarantees. The goal of this paper is to develop a statistical framework that leads to new algorithms with provable privacy guarantees, and performance bounds. Our main contributions are (i) Development of a statistical framework for federated personalized estimation and learning (ii) Theoretical bounds and

---

\*Equal Contribution.

novel algorithms for private personalized estimation (iii) Design and privacy analysis of new private personalized learning algorithms; as elaborated below. Omitted proofs/details are in appendices.

• **Statistical framework:** We connect this problem to the classical empirical Bayes' method, pioneered by Stein (1956); James & Stein (1961); Robbins (1956), which proposed a hierarchical statistical model Gelman et al. (2013). This is modeled by an *unknown* population distribution $\mathbb{P}$ from which local parameters $\{\boldsymbol{\theta}_i\}$ are generated, which in turn generate the local data through the distribution $\mathbb{Q}(\boldsymbol{\theta}_i)$. Despite the large literature on this topic, especially in the context of statistical estimation, creating a framework for FL poses new challenges. In contrast to classical empirical Bayes' estimation, we introduce a distributed setting and develop a framework that allows information (communication and privacy) constraints[1]. This framework enables us to develop statistical performance bounds as well as suggests (private) personalized federated estimation algorithms. Moreover, we develop our framework beyond estimation, for (supervised) *distributed learning*, where clients want to build *local* predictive models with limited local (labeled) samples; we develop this framework in Section 3, which leads to new (private) personalized learning algorithms.

• **Private personalized estimation:** Our goal is to estimate individual (local) parameters, when each user has very limited (heterogeneous) data. Such a scenario motivates federated estimation of individual parameters, *privately*. More precisely, the users observe data generated by an unknown distribution parametrized by their individual (unknown) local parameters $\boldsymbol{\theta}_i$, and want to estimate their local parameters $\boldsymbol{\theta}_i$ leveraging very limited local data; see Section 2 for more details. For the hierarchical statistical model, classical results have shown that one can enhance the estimate of individual parameters based on the observations of a population of samples, despite having *independently* generated parameters from an unknown population distributions. However, this has not been studied for the distributed case, with privacy and communication constraints, which we do (see Theorem 2 for the Gaussian case and Theorem 4 for the Bernoulli case, and also for mixture population models in Appendix D). We estimate the (parametrized) population distribution under these privacy and communication constraints and use this as an empirical prior for local estimation. The effective amplification of local samples through collaboration, in Section 2, gives us theoretical insight about when collaboration is most useful, under privacy and/or communication constraints. Our results suggest how to optimally balance estimates from local and population models. We also numerically evaluate these methods, including application to polling data (see Section 4 and Appendices) to show advantages of such collaborative estimation compared to local methods.

• **Private personalized learning:** The goal here is to obtain individual learning models capable of predicting labels with limited local data in a supervised learning setting. This is the use case for federated learning with privacy guarantees. It is intimately related to the estimation problem with distinctions including (i) to design good label predictors rather than just estimate local parameters (ii) the focus on iterative methods for optimization, requiring strong compositional privacy guarantees. Therefore, the statistical formulation for learning has a similar flavor to that in estimation, where there is a population model for local (parametrized) statistics for labeled data; see Section 3 for more details. We develop several algorithms, including `AdaPeD` (in Section 3.2), `AdaMix` (in Section 3.1), and `DP-AdaPeD` (in Section 3.3), inspired by the statistical framework. `AdaPeD` uses information divergence constraints along with adaptive weighting of local models and population models. By operating in probability (rather than Euclidean) space, using information-geometry (divergence), enables `AdaPeD` to operate with different local model sizes and architectures, giving it greater flexibility than existing methods. We integrate it with *user-level* privacy to develop `DP-AdaPeD`, with strong compositional privacy guarantees (Theorem 5). `AdaMix` is inspired by mixture population distributions, which adaptively weighs multiple global models and combines it with local data for personalization. We numerically evaluate these algorithms for synthetic and real data in Section 4.

**Related Work.** Our work can be seen in the intersection of personalized learning, estimation, and privacy. Below we give a brief description of related work; a more detailed comparison which connects our framework to other personalized algorithms is given in Appendix J.

**Personalized FL:** Recent work adopted different approaches for learning personalized models, which can be explained by our statistical framework for suitable choices of population distributions as explained in Appendix J: These include, *meta-learning based methods* (Fallah et al., 2020; Acar et al., 2021; Khodak et al., 2019); *regularization* (Deng et al., 2020; Mansour et al., 2020; Hanzely

---

[1]The homogeneous case for distributed estimation is well-studied; see (Zhang, 2016) and references.

& Richtárik, 2020); *clustered FL* (Zhang et al., 2021; Mansour et al., 2020; Ghosh et al., 2020; Smith et al., 2017) (Marfoq et al., 2021); *using knowledge distillation* (Lin et al., 2020; Ozkara et al., 2021); *multi-task Learning* (Dinh et al., 2020; Hanzely & Richtárik, 2020; Smith et al., 2017; Vanhaesebrouck et al., 2017; Zantedeschi et al., 2020); and *using common representations* (Du et al., 2021; Raghu et al., 2020; Tian et al., 2020; Collins et al., 2021) and references therein. Our work enables not just a unified view of these methods, but suggests new algorithms developed in this paper, along with privacy guarantees.

After the conclusion of our work (Ozkara et al., 2022, July), we found two concurrent and independent works (Kotelevskii et al., 2022, June; Chen et al., 2022) that use a hierarchical Bayes approach to construct personalized learning algorithms, and are closest to our statistical framework. (Kotelevskii et al., 2022, June) is based on using a MCMC method[2] to estimate a population distribution; such methods could be computationally intensive (see the discussion in (Blei et al., 2003); (Chen et al., 2022) considers the unimodal Gaussian prior, and effectively does what the classical empirical Bayes suggests (see also Theorem 1). None of these works consider privacy, which we do both for estimation and learning algorithms (see Theorems 2, 4, Appendix D, and for `DP-AdaPeD` in Theorem 5). Note that MCMC methods could have detrimental privacy implications. Also, they do not include information-geometric techniques (like our `AdaPeD`) or methods inspired by mixture distributions (*e.g.,* `AdaMix`).

**Privacy for Personalized Learning.** There has been a lot of work in privacy for FL when the goal is to learn a *single* global model (see (Girgis et al., 2021b) and references therein); though there are fewer papers that address user-level privacy (Liu et al., 2020; Levy et al., 2021; Ghazi et al., 2021). There has been more recent work on applying these ideas to learn personalized models (Girgis et al., 2022; Jain et al., 2021b; Geyer et al., 2017; Hu et al., 2020; Li et al., 2020). These are for specific algorithms/models, *e.g.,* Jain et al. (2021b) focuses on the common representation model for linear regression described earlier or on item-level privacy (Hu et al., 2020; Li et al., 2020). We believe that `DP-AdaPeD` proposed in this paper is among the first user-level private personalized learning algorithms with compositional guarantees, applicable to general deep learning architectures.

## 2 PERSONALIZED ESTIMATION

We consider a client-server architecture, where there are $m$ clients. Let $\mathbb{P}(\Gamma)$ denote a global population distribution that is parameterized by an unknown $\Gamma$ and let $\boldsymbol{\theta}_1, \ldots, \boldsymbol{\theta}_m$ are sampled i.i.d. from $\mathbb{P}(\Gamma)$ and are unknown to the clients. Client $i$ is given a dataset $X_i := (X_{i1}, \ldots, X_{in})$, where $X_{ij}, j \in [n]$ are sampled i.i.d. from some distribution $\mathbb{Q}(\boldsymbol{\theta}_i)$, parameterized by $\boldsymbol{\theta}_i \in \mathbb{R}^d$. Note that heterogeneity in clients' datasets is induced through the variance in $\mathbb{P}(\Gamma)$, and if the variance of $\mathbb{P}(\Gamma)$ is zero, then all clients observe i.i.d. datasets sampled from the *same* underlying distribution.

The goal at client $i$ for all $i \in [m]$ is to estimate $\boldsymbol{\theta}_i$ through the help of the server. We focus on one-round communication schemes, where client $j$ applies a (potentially randomized) mechanism $q$ on its dataset $X_j$ and sends $q_j := q(X_j)$ to the server, who aggregates the received messages, which is denoted by $\mathsf{Agg}(q_1, \ldots, q_m)$, and broadcasts that to all clients. Based on $(X_i, \mathsf{Agg}(q_1, \ldots, q_m))$, client $i$ outputs an estimate $\widehat{\boldsymbol{\theta}}_i$ of $\boldsymbol{\theta}_i$. We measure the performance of $\widehat{\boldsymbol{\theta}}_i$ through the Bayesian risk for mean squared error (MSE), as defined below (where $\mathbb{P}$ is the true prior distribution with associated density $\pi$, $\boldsymbol{\theta}_i \sim \mathbb{P}$ is the true local parameter, and $\widehat{\boldsymbol{\theta}}_i = \widehat{\boldsymbol{\theta}}(X_i, \mathsf{Agg}(q_1, \ldots, q_m))$ is the estimator):

$$\mathbb{E}_{\boldsymbol{\theta}_i \sim \mathbb{P}} \mathbb{E}_{\widehat{\boldsymbol{\theta}}_i, q, X_1, \ldots, X_m} \|\widehat{\boldsymbol{\theta}}_i - \boldsymbol{\theta}_i\|^2 = \int \mathbb{E}_{\widehat{\boldsymbol{\theta}}_i, q, X_1, \ldots, X_m} \|\widehat{\boldsymbol{\theta}}_i - \boldsymbol{\theta}_i\|^2 \pi(\boldsymbol{\theta}_i) d\boldsymbol{\theta}_i. \tag{1}$$

The above statistical framework can model many different scenarios, and we will study in detail three settings: Gaussian and Bernoulli models (Sections 2.1, 2.2 below), and Mixture model (Appendix D).

### 2.1 GAUSSIAN MODEL

In the Gaussian setting, $\mathbb{P}(\Gamma) = \mathcal{N}(\boldsymbol{\mu}, \sigma_\theta^2 \mathbb{I}_d)$ and $\mathbb{Q}(\boldsymbol{\theta}_i) = \mathcal{N}(\boldsymbol{\theta}_i, \sigma_x^2 \mathbb{I}_d)$ for all $i \in [m]$, which implies that $\boldsymbol{\theta}_1, \ldots, \boldsymbol{\theta}_m \sim \mathcal{N}(\boldsymbol{\mu}, \sigma_\theta^2 \mathbb{I}_d)$ i.i.d. and $X_{i1}, \ldots, X_{in} \sim \mathcal{N}(\boldsymbol{\theta}_i, \sigma_x^2 \mathbb{I}_d)$ i.i.d. for $i \in [m]$. Here, $\sigma_\theta \geq 0, \sigma_x > 0$ are known, and $\boldsymbol{\mu}, \boldsymbol{\theta}_1, \ldots, \boldsymbol{\theta}_m$ are unknown. For the case of a single local

---

[2]In our understanding their numerics seem to be restricted to a unimodal Gaussian population model.

sample this is identical to the classical James-Stein estimator (James & Stein, 1961); Theorem 1 does a simple extension for multiple local samples and is actually a stepping stone for the information constrained estimation result of Theorem 2. Omitted proofs/details are provided in Appendix B.

**Our proposed estimator.** Since there is no distribution on $\boldsymbol{\mu}$, and given $\boldsymbol{\mu}$, we know the distribution of $\boldsymbol{\theta}_i$'s, and subsequently, of $X_{ij}$'s. So, we consider the maximum likelihood estimator:

$$\widehat{\boldsymbol{\theta}}_1, \ldots, \widehat{\boldsymbol{\theta}}_m, \widehat{\boldsymbol{\mu}} := \underset{\boldsymbol{\theta}_1, \ldots, \boldsymbol{\theta}_m, \boldsymbol{\mu}}{\arg \max} \, p_{\{\boldsymbol{\theta}_i, X_i\}|\boldsymbol{\mu}} \left( \boldsymbol{\theta}_1, \ldots, \boldsymbol{\theta}_m, X_1, \ldots, X_m | \boldsymbol{\mu} \right) \tag{2}$$

**Theorem 1.** *Solving* (2) *yields the following closed form expressions for* $\widehat{\boldsymbol{\mu}}$ *and* $\widehat{\boldsymbol{\theta}}_1, \ldots, \widehat{\boldsymbol{\theta}}_m$:

$$\widehat{\boldsymbol{\mu}} = \frac{1}{m} \sum_{i=1}^{m} \overline{X}_i \qquad and \qquad \widehat{\boldsymbol{\theta}}_i = a\overline{X}_i + (1-a)\widehat{\boldsymbol{\mu}}, \text{ for } i \in [m], \quad where \; a = \frac{\sigma_\theta^2}{\sigma_\theta^2 + \sigma_x^2/n}. \tag{3}$$

*The above estimator achieves the MSE:* $\mathbb{E}_{\boldsymbol{\theta}_i, X_1, \ldots, X_m} \|\widehat{\boldsymbol{\theta}}_i - \boldsymbol{\theta}_i\|^2 \leq \frac{d\sigma_x^2}{n} \left( \frac{1-a}{m} + a \right)$.

It follows that the mechanism $q$ and the aggregation function Agg for the estimators in (3) (as described in (1)) are just the average functions, where client $i$ sends $q_i = q(X_i) := \overline{X}_i = \frac{1}{n} \sum_{j=1}^{n} X_{ij}$ to the server, and the server sends $\widehat{\boldsymbol{\mu}} := \text{Agg}(q_1, \ldots, q_m) = \frac{1}{m} \sum_{i=1}^{m} q_i$ back to the clients.

**Remark 1** (Personalized estimate vs. local estimate). *When* $\sigma_\theta \to 0$, *then* $a \to 0$, *which implies that* $\widehat{\boldsymbol{\theta}}_i \to \widehat{\boldsymbol{\mu}}$ *and* $MSE \to d\sigma_x^2/mn$. *Otherwise, when* $\sigma_\theta^2$ *is large in comparison to* $\sigma_x^2/n$ *or* $n \to \infty$, *then* $a \to 1$, *which implies that* $\widehat{\boldsymbol{\theta}}_i \to \overline{X}_i$ *and* $MSE \to d\sigma_x^2/n$. *These conform to the facts that* (i) *when there is no heterogeneity, then the global average is the best estimator, and* (ii) *when heterogeneity is not small, and we have a lot of local samples, then the local average is the best estimator. Observe that the multiplicative gap between the MSE of the proposed personalized estimator and the MSE of the local estimator (based on local data only, which gives an MSE of* $d\sigma_x^2/n$*) is given by* $\left( \frac{1-a}{m} + a \right) \leq 1$ *that proves the superiority of the personalized model over the local model, which is equal to* $1/m$ *when* $\sigma_\theta = 0$ *and equal to* $0.01$ *when* $m = 10^4, n = 100$ *and* $\sigma_x^2 = 10, \sigma_\theta^2 = 10^{-3}$*, for example.*

**Remark 2** (Optimality of our personalized estimator). *In Appendix B, we show the minimax lower bound:* $\inf_{\widehat{\boldsymbol{\theta}}} \sup_{\boldsymbol{\theta} \in \Theta} \mathbb{E}_{X \sim \mathcal{N}(\boldsymbol{\theta}, \sigma_x^2)} \|\widehat{\boldsymbol{\theta}}(X) - \boldsymbol{\theta}\|^2 \geq \frac{d\sigma_x^2}{n} \left( \frac{1-a}{m} + a \right)$*, which exactly matches the upper bound on the MSE in Theorem 1, thus establishes the optimality our personalized estimator in* (3)*.*

**Privacy and communication constraints.** Observe that the scheme presented above does not protect privacy of clients' data and messages from the clients to the server can be made communication-efficient. These could be achieved by employing specific mechanisms $q$ at clients: For privacy, we can take a differentially-private $q$, and for communication-efficiency, we can take $q$ to be a quantizer. Inspired by the scheme presented above, here we consider $q$ to be a function $q : \mathbb{R}^d \to \mathcal{Y}$, that takes the average of $n$ data points as its input, and the aggregator function Agg to be the average function. Define $\widehat{\boldsymbol{\mu}}_q := \frac{1}{m} \sum_{i=1}^{m} q(\overline{X}_i)$ and consider the following personalized estimator for the $i$-th client:

$$\widehat{\boldsymbol{\theta}}_i = a\overline{X}_i + (1-a)\widehat{\boldsymbol{\mu}}_q, \qquad \text{for some } a \in [0, 1]. \tag{4}$$

**Theorem 2.** *Suppose for all* $\boldsymbol{x} \in \mathbb{R}^d$*,* $q$ *satisfies* $\mathbb{E}[q(\boldsymbol{x})] = \boldsymbol{x}$ *and* $\mathbb{E}\|q(\boldsymbol{x}) - \boldsymbol{x}\|^2 \leq d\sigma_q^2$ *for some finite* $\sigma_q$*. Then the personalized estimator in* (4) *has MSE:*

$$\mathbb{E}_{\boldsymbol{\theta}_i, q, X_1, \ldots, X_m} \|\widehat{\boldsymbol{\theta}}_i - \boldsymbol{\theta}_i\|^2 \leq \frac{d\sigma_x^2}{n} \left( \frac{1-a}{m} + a \right) \qquad where \qquad a = \frac{\sigma_\theta^2 + \sigma_q^2/m-1}{\sigma_\theta^2 + \sigma_q^2/m-1 + \sigma_x^2/n}. \tag{5}$$

*Furthermore, assuming* $\boldsymbol{\mu} \in [-r, r]^d$ *for some constant* $r$ *(but* $\boldsymbol{\mu}$ *is unknown), we have:*

1. *Communication efficiency: For any* $k \in \mathbb{N}$*, there is a* $q$ *whose output can be represented using* $k$-bits *(i.e.,* $q$ *is a quantizer) that achieves the MSE in* (5) *with probability at least* $1 - 2/mn$ *and with* $\sigma_q = \frac{b}{(2^k-1)}$*, where* $b = r + \sigma_\theta \sqrt{\log(m^2 n)} + \frac{\sigma_x}{\sqrt{n}} \sqrt{\log(m^2 n)}$*.*

2. *Privacy: For any* $\epsilon_0 \in (0, 1), \delta > 0$*, there is a* $q$ *that is user-level* $(\epsilon_0, \delta)$-locally differentially private, that achieves the MSE in* (5) *with probability at least* $1 - 2/mn$ *and with* $\sigma_q = \frac{b}{\epsilon_0} \sqrt{8 \log(2/\delta)}$*, where* $b = r + \sigma_\theta \sqrt{\log(m^2 n)} + \frac{\sigma_x}{\sqrt{n}} \sqrt{\log(m^2 n)}$*.*

### 2.2 BERNOULLI MODEL

For the Bernoulli model, $\mathbb{P}$ is supported on $[0,1]$, and $p_1, \ldots, p_m$ are sampled i.i.d. from $\mathbb{P}$, and client $i$ is given $n$ i.i.d. samples $X_{i1}, \ldots, X_{in} \sim \mathsf{Bern}(p_i)$. This setting has been studied in (Tian et al. (2017); Vinayak et al. (2019)) for estimating $\mathbb{P}$, whereas, our goal is to estimate individual parameter $p_i$ at client $i$ using the information from other clients. In order to derive a closed form MSE result, we assume that $\mathbb{P}$ is the Beta distribution.[3] Here, $\Gamma = (\alpha, \beta), p_1, \ldots, p_m$ are unknown, and client $i$'s goal is to estimate $p_i$ such that the Bayesian risk $\mathbb{E}_{p_i \sim \pi}\mathbb{E}_{\widehat{p}_i, X_1, \ldots, X_m}(\widehat{p}_i - p_i)^2$ is minimized, where $\pi$ denotes the density of the Beta distribution. Omitted proofs/details are provided in Appendix C.

Analogous to the Gaussian case, we can show that if $\alpha, \beta$ are known, then the posterior mean estimator has a closed form expression: $\widehat{p}_i = a\overline{X}_i + (1-a)\frac{\alpha}{\alpha+\beta}$, where $a = {}^{n}/_{\alpha+\beta+n}$ and ${}^{\alpha}/_{(\alpha+\beta)}$ is the mean of the beta distribution. When $\alpha, \beta$ are unknown, inspired by the above discussion, a natural approach would be to estimate the global mean $\mu = {}^{\alpha}/_{(\alpha+\beta)}$ and the weight $a = {}^{n}/_{(\alpha+\beta+n)}$, and use that in the above estimator. Note that, for $a$ we need to estimate $\alpha + \beta$, which is equal to ${}^{\mu(1-\mu)}/_{\sigma^2} - 1$, where $\sigma^2 = {}^{\alpha\beta}/_{(\alpha+\beta)^2(\alpha+\beta+1)}$ is the variance of the beta distribution. Therefore, it is enough to estimate $\mu$ and $\sigma^2$ for the personalized estimators $\{\widehat{p}_i\}$. In order to make calculations simpler, instead of making one estimate of $\mu, \sigma^2$ for all clients, we let each client make its own estimate of $\mu, \sigma^2$ (without using their own data) as: $\widehat{\mu}_i = \frac{1}{m-1}\sum_{l \neq i}\overline{X}_l$ and $\widehat{\sigma}_i^2 = \frac{1}{m-2}\sum_{l \neq i}(\overline{X}_l - \widehat{\mu}_l)^2$,[4] and then define the local weight as $\widehat{a}_i = \frac{n}{\widehat{\mu}_i(1-\widehat{\mu}_i)/\widehat{\sigma}_i^2 - 1 + n}$. Now, client $i$ uses the following personalized estimator:

$$\widehat{p}_i = \widehat{a}_i\overline{X}_i + (1-\widehat{a}_i)\widehat{\mu}_i. \tag{6}$$

**Theorem 3.** *With probability at least $1 - \frac{1}{mn}$, the MSE of the personalized estimator in* (6) *is given by:* $\mathbb{E}_{p_i \sim \pi}\mathbb{E}_{X_1, \ldots, X_m}(\widehat{p}_i - p_i)^2 \leq \mathbb{E}[\widehat{a}_i^2]\big(\frac{\alpha\beta}{n(\alpha+\beta)(\alpha+\beta+1)}\big) + \mathbb{E}[(1-\widehat{a}_i)^2]\big(\frac{\alpha\beta}{(\alpha+\beta)^2(\alpha+\beta+1)} + \frac{3\log(4m^2n)}{m-1}\big)$.

**Remark 3.** *When $n \to \infty$, then $\widehat{a}_i \to 1$, which implies that MSE tends to the MSE of the local estimator $\overline{X}_i$, which means if local samples are abundant, collaboration does not help much. When $\sigma^2 = {}^{\alpha\beta}/_{(\alpha+\beta)^2(\alpha+\beta+1)} \to 0$, i.e. there is very small heterogeneity in the system, then $\widehat{a}_i \to 0$, which implies that MSE tends to the error due to moment estimation (the last term in the MSE in Theorem 3).*

**Privacy constraints.** For any privacy parameter $\epsilon_0 > 0$ and input $x \in [0,1]$, define $q^{\mathsf{priv}} : [0,1] \to \mathbb{R}$:

$$q^{\mathsf{priv}}(x) = \begin{cases} \frac{-1}{e^{\epsilon_0}-1} & \text{w.p. } \frac{e^{\epsilon_0}}{e^{\epsilon_0}+1} - x\frac{e^{\epsilon_0}-1}{e^{\epsilon_0}+1}, \\ \frac{e^{\epsilon_0}}{e^{\epsilon_0}-1} & \text{w.p. } \frac{1}{e^{\epsilon_0}+1} + x\frac{e^{\epsilon_0}-1}{e^{\epsilon_0}+1}. \end{cases} \tag{7}$$

The mechanism $q^{\mathsf{priv}}$ is unbiased and satisfies user-level $\epsilon_0$-LDP. Thus, the $i$th client sends $q^{\mathsf{priv}}(\overline{X}_i)$ to the server, which computes $\widehat{\mu}_i^{\mathsf{priv}} = \frac{1}{m-1}\sum_{l \neq i} q^{\mathsf{priv}}(\overline{X}_l)$ and the variance $\widehat{\sigma}_i^{2(\mathsf{priv})} = \frac{1}{m-2}\sum_{l \neq i}(q^{\mathsf{priv}}(\overline{X}_l)) - \widehat{\mu}_l^{\mathsf{priv}})^2$ and sends $(\widehat{\mu}_i^{\mathsf{priv}}, \widehat{\sigma}_i^{2(\mathsf{priv})})$ to client $i$. Upon receiving this, client $i$ defines $\widehat{a}_i^{\mathsf{priv}} = \frac{n}{\widehat{\mu}_i^{\mathsf{priv}}(1-\widehat{\mu}_i^{\mathsf{priv}})/\widehat{\sigma}_i^{2(\mathsf{priv})} + n}$ and uses $\widehat{p}_i^{\mathsf{priv}} = \widehat{a}_i^{\mathsf{priv}}\overline{X}_i + (1-\widehat{a}_i^{\mathsf{priv}})\widehat{\mu}^{\mathsf{priv}}$ to estimate $p_i$.

**Theorem 4.** *With probability at least $1 - \frac{1}{mn}$, the MSE of the personalized estimator $\widehat{p}_i^{\mathsf{priv}}$ defined above is given by:* $\mathbb{E}_{p_i \sim \pi}\mathbb{E}_{q^{\mathsf{priv}}, X_1, \ldots, X_m}(\widehat{p}_i^{\mathsf{priv}} - p_i)^2 \leq \mathbb{E}[(\widehat{a}_i^{\mathsf{priv}})^2]\big(\frac{\alpha\beta}{n(\alpha+\beta)(\alpha+\beta+1)}\big) + \mathbb{E}[(1 - \widehat{a}_i^{\mathsf{priv}})^2]\big(\frac{\alpha\beta}{(\alpha+\beta)^2(\alpha+\beta+1)} + \frac{(e^{\epsilon_0}+1)^2\log(4m^2n)}{3(e^{\epsilon_0}-1)^2(m-1)}\big)$.

See Remark 4 (in Appendix B) and Remarks 6 and 7 (in Appendix C) for a discussion on privacy, communication efficiency, and client sampling.

# 3 PERSONALIZED LEARNING

Consider a client-server architecture with $m$ clients. There is an unknown global population distribution $\mathbb{P}(\Gamma)$[5] over $\mathbb{R}^d$ from which $m$ i.i.d. local parameters $\boldsymbol{\theta}_1, \ldots, \boldsymbol{\theta}_m \in \mathbb{R}^d$ are sampled. Each client

[3] Beta distribution has a density $\mathsf{Beta}(\alpha,\beta) = \frac{1}{B(\alpha,\beta)}x^{\alpha-1}(1-x)^{\beta-1}$ is defined for $\alpha, \beta > 0$ and $x \in [0,1]$, where $B(\alpha, \beta)$ is a normalizing constant. Its mean is $\frac{\alpha}{\alpha+\beta}$ and the variance is $\frac{\alpha\beta}{(\alpha+\beta)^2(\alpha+\beta+1)}$.

[4] Upon receiving $\{\overline{X}_i\}$ from all clients, the server can compute $\{\widehat{\mu}_i, \widehat{\sigma}_i^2\}$ and sends $(\widehat{\mu}_i, \widehat{\sigma}_i^2)$ to the $i$-th client.

[5] For simplicity we will consider this unknown population distribution $\mathbb{P}$ to be parametrized by unknown (arbitrary) parameters $\Gamma$.

$i \in [m]$ is provided with a dataset consisting of $n$ data points $\{(X_{i1}, Y_{i1}), \ldots, (X_{in}, Y_{in})\}$, where $Y_{ij}$'s are generated from $(X_{ij}, \boldsymbol{\theta}_i)$ using some distribution $p_{\boldsymbol{\theta}_i}(Y_{ij}|X_{ij})$. Let $Y_i := (Y_{i1}, \ldots, Y_{in})$ and $X_i := (X_{i1}, \ldots, X_{in})$ for $i \in [m]$. The underlying statistical model for our setting is given by

$$p_{\{\boldsymbol{\theta}_i, Y_i\}|\{X_i\}}(\boldsymbol{\theta}_1, \ldots, \boldsymbol{\theta}_m, Y_1, \ldots, Y_m | X_1, \ldots, X_m) = \prod_{i=1}^{m} p(\boldsymbol{\theta}_i) \prod_{i=1}^{m} \prod_{j=1}^{n} p_{\boldsymbol{\theta}_i}(Y_{ij}|X_{ij}). \quad (8)$$

Note that if we minimize the negative log likelihood of (8), we would get the optimal parameters:

$$\widehat{\boldsymbol{\theta}}_1, \ldots, \widehat{\boldsymbol{\theta}}_m := \underset{\boldsymbol{\theta}_1, \ldots, \boldsymbol{\theta}_m}{\arg\min} \sum_{i=1}^{m} \sum_{j=1}^{n} -\log(p_{\boldsymbol{\theta}_i}(Y_{ij}|X_{ij})) + \sum_{i=1}^{m} -\log(p(\boldsymbol{\theta}_i)). \quad (9)$$

Here, $f_i(\boldsymbol{\theta}_i) := \sum_{j=1}^{n} -\log(p_{\boldsymbol{\theta}_i}(Y_{ij}|X_{ij}))$ denotes the loss function at the $i$-th client, which only depends on the local data, and $R(\{\boldsymbol{\theta}_i\}) := \sum_{i=1}^{m} -\log(p(\boldsymbol{\theta}_i))$ is the regularizer that depends on the (unknown) global population distribution $\mathbb{P}$ (parametrized by unknown $\Gamma$). Note that when clients have little data and we have large number of clients, i.e., $n \ll m$ – the setting of federated learning, clients may not be able to learn good personalized models from their local data alone (if they do, it would lead to large loss). In order to learn better personalized models, clients may utilize other clients' data through collaboration, and the above regularizer (and estimates of the unknown prior distribution $\mathbb{P}$, through estimating its parameters $\Gamma$) dictates how the collaboration might be utilized. The above-described statistical framework (9) can model many different scenarios, as detailed below:

1. When $\mathbb{P}(\Gamma) \equiv \mathrm{GM}(\{p_l\}_{l=1}^{k}, \{\boldsymbol{\mu}_l\}_{l=1}^{k}, \{\sigma_{\theta,l}^2\}_{l=1}^{k})$ is a Gaussian mixture, for $\Gamma = \{(\{p_l\}_{l=1}^{k}, \{\boldsymbol{\mu}_l\}_{l=1}^{k}, \{\sigma_{\theta,l}\}_{l=1}^{k}) : p_l \geq 0, \sum_{l=1}^{k} p_l = 1, \sigma_{\theta,l} \geq 0, \boldsymbol{\mu}_l \in \mathbb{R}^d\}$, then $R(\{\theta_i\}) = -\sum_{i=1}^{m} \log\left(\sum_{l=1}^{k} p_l \exp(-\frac{\|\boldsymbol{\mu}_l - \theta_i\|_2^2}{2\sigma_{\theta,l}^2})/((2\pi\sigma_{\theta,l})^{d/2})\right)$. Here, the client models $\boldsymbol{\theta}_1, \ldots, \boldsymbol{\theta}_m$ are drawn i.i.d. from $\mathbb{P}(\Gamma)$, where $\boldsymbol{\theta}_i \sim \mathcal{N}(\boldsymbol{\mu}_l, \sigma_{\theta,l}^2 \mathbb{I}_d)$ with prob. $p_l$, for $l = 1, \ldots, k$. For $k = 1$, $R(\{\boldsymbol{\theta}_i\}) = \frac{md}{2} \log(2\pi\sigma_\theta^2) + \sum_{i=1}^{m} \frac{\|\boldsymbol{\mu} - \boldsymbol{\theta}_i\|_2^2}{2\sigma_\theta^2}$. Here, unknown $\boldsymbol{\mu}$ can be connected to the global model and $\boldsymbol{\theta}_i$'s as local models, and the alternating iterative optimization optimizes over both. This justifies the use of $\ell_2$ regularizer in earlier personalized learning works (Dinh et al., 2020; Ozkara et al., 2021; Hanzely & Richtárik, 2020; Hanzely et al., 2020; Li et al., 2021).

2. When $\mathbb{P}(\Gamma) \equiv \mathsf{Laplace}(\boldsymbol{\mu}, b)$, for $\Gamma = \{\boldsymbol{\mu}, b > 0\}$, then $R(\{\boldsymbol{\theta}_i\}) = m \log(2b) + \sum_{i=1}^{m} \frac{\|\boldsymbol{\theta}_i - \boldsymbol{\mu}\|_1}{b}$.

3. When $p_{\boldsymbol{\theta}_i}(Y_{ij}|X_{ij})$ is according to $\mathcal{N}(\boldsymbol{\theta}_i, \sigma_x^2)$, then $f_i(\boldsymbol{\theta}_i)$ is the quadratic loss as in linear regression. When $p_{\boldsymbol{\theta}_i}(Y_{ij}|X_{ij}) = \sigma(\langle\boldsymbol{\theta}_i, X_{ij}\rangle)^{Y_{ij}}(1 - \sigma(\langle\boldsymbol{\theta}_i, X_{ij}\rangle))^{(1-Y_{ij})}$, where $\sigma(z) = 1/1+e^{-z}$ for any $z \in \mathbb{R}$, then $f_i(\boldsymbol{\theta}_i)$ is the cross-entropy (or logistic) loss as in logistic regression.

### 3.1 AdaMix: Adaptive Personalization with Gaussian Mixture Prior

Now we write the full objective function for the Gaussian mixture prior model for a generic local loss function $f_i(\boldsymbol{\theta}_i)$ at client $i$ (the case of linear/logistic regression with (single) Gaussian prior and solving using alternating gradient descent is discussed in Appendices E, F.):

$$\underset{\{\boldsymbol{\theta}_i\}, \{\boldsymbol{\mu}_l\}, \{p_l\}, \{\sigma_{\theta,l}\}}{\arg\min} \sum_{i=1}^{m} F_i^{\mathrm{gm}}(\boldsymbol{\theta}_i) := \sum_{i=1}^{m} \left(f_i(\boldsymbol{\theta}_i) - \log(\sum_{l=1}^{k} p_l \exp(-\frac{\|\boldsymbol{\mu}_l - \boldsymbol{\theta}_i\|_2^2}{2\sigma_{\theta,l}^2})/((2\pi\sigma_{\theta,l})^{d/2})))\right)$$
$$(10)$$

A common example of $f_i(\boldsymbol{\theta}_i)$ is a generic neural network loss function with multi-class softmax output layer and cross entropy loss, i.e., $f_i(\boldsymbol{\theta}_i) := \sum_{j=1}^{n} -\log(p_{\boldsymbol{\theta}_i}(Y_{ij}|X_{ij}))$, where $p_{\boldsymbol{\theta}_i}(Y_{ij}|X_{ij}) = \sigma(\langle\boldsymbol{\theta}_i, X_{ij}\rangle)^{Y_{ij}}(1 - \sigma(\langle\boldsymbol{\theta}_i, X_{ij}\rangle))^{(1-Y_{ij})}$, where $\sigma(z) = 1/1+e^{-z}$ for any $z \in \mathbb{R}$. To solve (10), we can either use an alternating gradient descent approach, or we can use a clustering based approach where the server runs a (soft) clustering algorithm on received personalized models. We adopt the second approach here (described in Algorithm 1) as it provides an interesting point of view and can be combined with DP clustering algorithms. Here clients receive the global parameters from the server and do a local iteration on the personalized model (multiple local iterations can be introduced as in FedAvg (McMahan et al., 2017)), later the clients send the personalized models. Receiving the personalized models, server initiates GMM algorithm that outputs global parameters [6]

---

[6]A discrete mixture model can be proposed as a special case of GM with 0 variance. With this we can recover a similar algorithm as in Marfoq et al. (2021). Further details are presented in Appendix G.

## 3.2 ADAPED: ADAPTIVE PERSONALIZATION VIA DISTILLATION

It has been empirically observed that the knowledge distillation (KD) regularizer (between local and global models) results in better performance than the $\ell_2$ regularizer (Ozkara et al., 2021). In fact, using our framework, we can define, for the first time, a certain prior distribution that gives the KD regularizer (see Appendix H). We use the following loss function at the $i$-th client:

$$f_i(\boldsymbol{\theta}_i) + \frac{1}{2}\log(2\psi) + \frac{f_i^{\mathsf{KD}}(\boldsymbol{\theta}_i, \boldsymbol{\mu})}{2\psi}, \quad (11)$$

where $\boldsymbol{\mu}$ denotes the global model, $\boldsymbol{\theta}_i$ denotes the personalized model at client $i$, and $\psi$ can be viewed as controlling heterogeneity. The goal for each client is to minimize its local loss function, so individual components cannot be too large. For the second term, this implies that $\psi$ cannot be unbounded. For the third term, if $f_i^{\mathsf{KD}}(\boldsymbol{\theta}_i, \boldsymbol{\mu})$ is large, then $\psi$ will also increase (implying that the local parameters are too deviated from the global parameter), hence, it is better to emphasize local training loss to make the first term small. If $f_i^{\mathsf{KD}}(\boldsymbol{\theta}_i, \boldsymbol{\mu})$ is small, then $\psi$ will also decrease (implying that the local parameters are close to the global parameter), so it is better to collaborate and learn better personalized models. Such adaptive weighting quantifies the uncertainty in population distribution during training, balances the learning accordingly, and improves the empirical performance over non-adaptive methods, e.g., (Ozkara et al., 2021).

---

**Algorithm 1** Personalized Learning with Gaussian Mixture Prior (AdaMix)

**Input:** Number of iterations $T$, local datasets $(X_i, Y_i)$ for $i \in [m]$, learning rate $\eta$.

1: **Initialize** $\boldsymbol{\theta}_1^{(0)}, \ldots, \boldsymbol{\theta}_m^{(0)}$ and $\mathbb{P}^{(0)}, \boldsymbol{\mu}_1^{(0)}, \ldots, \boldsymbol{\mu}_k^{(0)}, \sigma_{\theta,1}^{(0)}, \ldots, \sigma_{\theta,k}^{(0)}$.
2: **for** $t = 1$ **to** $T$ **do**
3:    **On Clients:**
4:    **for** $i = 1$ **to** $m$: **do**
5:       Receive $\mathbb{P}^{(t-1)}, \boldsymbol{\mu}_1^{(t-1)}, \ldots, \boldsymbol{\mu}_k^{(t-1)}$, and $\sigma_{\theta,1}^{(t-1)}, \ldots, \sigma_{\theta,k}^{(t-1)}$ from the server
6:       Update the personalized parameters:

$$\boldsymbol{\theta}_i^{(t)} \leftarrow \boldsymbol{\theta}_i^{(t-1)} - \eta \nabla_{\boldsymbol{\theta}_i^{(t-1)}} F_i^{\mathsf{gm}}(\boldsymbol{\theta}_i^{(t-1)})$$

7:       Send $\boldsymbol{\theta}_i^{(t)}$ to the server
8:    **end for**
9:    **At the Server:**
10:   Receive $\boldsymbol{\theta}_1^{(t)}, \ldots, \boldsymbol{\theta}_m^{(t)}$ from the clients
11:   Update the global parameters:

$$\mathbb{P}^{(t)}, \boldsymbol{\mu}_1^{(t)}, \ldots, \boldsymbol{\mu}_k^{(t)}, \sigma_{\theta,1}^{(t)}, \ldots, \sigma_{\theta,k}^{(t)}$$
$$\leftarrow \mathsf{GMM}(\boldsymbol{\theta}_1^{(t)}, \ldots, \boldsymbol{\theta}_m^{(t)}, k)$$

12:   Broadcast $\mathbb{P}^{(t)}, \{\boldsymbol{\mu}_i^{(t)}\}_{i=1}^k, \{\sigma_{\theta,i}^{(t)}\}_{i=1}^k$ to all clients
13: **end for**

**Output:** Personalized models $\boldsymbol{\theta}_1^T, \ldots, \boldsymbol{\theta}_m^T$.

---

To optimize (11) we propose an alternating minimization approach, which we call AdaPeD; see Algorithm 2. Besides the personalized model $\boldsymbol{\theta}_i^t$, each client $i$ keeps local copies of the global model $\boldsymbol{\mu}_i^t$ and of the dissimilarity term $\psi_i^t$, and at synchronization times, server aggregates them to obtain global versions of these $\boldsymbol{\mu}^t, \psi^t$. In this way, the local training of $\boldsymbol{\theta}_i^t$ also incorporates knowledge from other clients' data through $\boldsymbol{\mu}_i^t$. In the end, clients have learned their personalized models $\{\boldsymbol{\theta}_i^T\}_{i=1}^m$.

## 3.3 DP-ADAPED: DIFFERENTIALLY PRIVATE ADAPTIVE PERSONALIZATION VIA DISTILL.

Note that client $i$ communicates $\boldsymbol{\mu}_i^t, \psi_i^t$ (which are updated by accessing the dataset for computing the gradients $\boldsymbol{h}_i^t, k_i^t$) to the server. So, to privatize $\boldsymbol{\mu}_i^t, \psi_i^t$, client $i$ adds appropriate noise to $\boldsymbol{h}_i^k, k_i^t$. In order to obtain DP-AdaPeD, we replace lines 13 and 15, respectively, by the update rules:

$$\boldsymbol{\mu}_i^{t+1} = \boldsymbol{\mu}_i^t - \eta_2\left(\frac{\boldsymbol{h}_i^t}{\max\{\|\boldsymbol{h}_i^t\|/C_1, 1\}} + \boldsymbol{\nu}_1\right) \quad \text{and} \quad \psi_i^{t+1} = \psi_i^t - \eta_3\left(\frac{k_i^t}{\max\{|k_i^t|/C_2, 1\}} + \nu_2\right),$$

where $\boldsymbol{\nu}_1 \sim \mathcal{N}(0, \sigma_{q_1}^2 \mathbb{I}_d)$ and $\nu_2 \sim \mathcal{N}(0, \sigma_{q_2}^2)$, for some $\sigma_{q_1}, \sigma_{q_2} > 0$ that depend on the desired privacy level and $C_1, C_2$, which are some predefined constants.

The theorem below (proved in Appendix I) states the Rényi Differential Privacy (RDP) guarantees.

**Theorem 5.** *After $T$ iterations, DP-AdaPeD satisfies $(\alpha, \epsilon(\alpha))$-RDP for $\alpha > 1$, where $\epsilon(\alpha) = \left(\frac{K}{m}\right)^2 6\frac{T}{\tau}\alpha\left(\frac{C_1^2}{K\sigma_{q_1}^2} + \frac{C_2^2}{K\sigma_{q_2}^2}\right)$, where $\frac{K}{m}$ denotes the sampling ratio of clients at each global iteration.*

We bound the RDP, as it gives better privacy composition than using the strong composition (Mironov et al., 2019). We can also convert our results to user-level $(\epsilon, \delta)$-DP by using the standard conversion from RDP to $(\epsilon, \delta)$-DP (Canonne et al., 2020). See Appendix A for background on privacy.

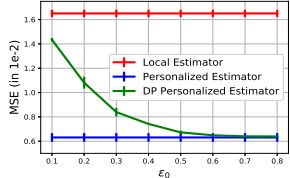
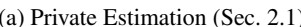

(a) Private Estimation (Sec. 2.1)

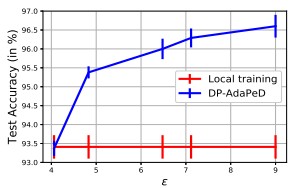

(b) Private Learning (Sec. 3.3)

Figure 1: In Fig. 1a, we plot MSE vs. $\epsilon_0$ for personalized estimation. In Fig. 1b, we plot Test Accuracy vs. $\epsilon$ on FEMNIST with client sampling 0.33, for `DP-AdaPeD` with unsampled client iterations. Since local training is private, both plots remain constant against $\epsilon$.

## 4 EXPERIMENTS

**Personalized Estimation.** We run one experiment for Bernoulli setting with real political data and the other for Gaussian setting with synthetic data. The latter one is differentially private.

● **Political tendencies on county level.** One natural application of Bernoulli setting is modeling bipartisan elections (Tian et al., 2017). We did a case study by using US presidential elections on county level between 2000-2020, with $m = 3112$ counties in our dataset. For each county the goal is to determine the political tendency parameter $p_i$. Given 6 election data we did 6-fold cross validation, with 5 elections for training and 1 election for test data. Local estimator takes an average of 5 training samples and personalized estimator is the posterior mean. To simulate a Bernoulli setting we set the data equal to 1 if Republican party won the election and 0 otherwise. We observe the personalized estimator provides MSE (averaged over 6 runs) gain of $10.7 \pm 1.9\%$ against local estimator.

● **DP personalized estimation.** To measure the performance tradeoff of the DP mechanism described in Section 2.1, we create a synthetic experiment for Gaussian setting. We let $m = 10000, n = 15$ and $\sigma_\theta = 0.1, \sigma_x = 0.5$, and create a dataset at each client as described in Gaussian setting. Applying the DP mechanism we obtain the following result in Figure 1a. Here, as expected, when privacy is low ($\epsilon_0$ is high) the private personalized estimator recovers the regular personalized estimator. For higher privacy

---

**Algorithm 2** Adaptive Personalization via Distillation (`AdaPeD`)

**Parameters:** local variances $\{\psi_i^0\}$, personalized models $\{\boldsymbol{\theta}_i^0\}$, local copies of the global model $\{\boldsymbol{\mu}_i^0\}$, learning rates $\eta_1, \eta_2, \eta_3$, synchronization gap $\tau$

1: **for** $t = 0$ **to** $T - 1$ **do**
2:     **if** $\tau$ divides $t$ **then**
3:       **On Server do:**
4:       Choose a subset $\mathcal{K}^t \subseteq [n]$ of $K$ clients
5:       Broadcast $\boldsymbol{\mu}^t$ and $\psi^t$
6:       **On Clients** $i \in \mathcal{K}^t$ (in parallel) **do**:
7:       Receive $\boldsymbol{\mu}^t, \psi^t$; set $\boldsymbol{\mu}_i^t = \boldsymbol{\mu}^t, \psi_i^t = \psi^t$
8:     **end if**
9:     **On Clients** $i \in \mathcal{K}^t$ (in parallel) **do**:
10:     Compute $\boldsymbol{g}_i^t := \nabla_{\boldsymbol{\theta}_i^t} f_i(\boldsymbol{\theta}_i^t) + \frac{\nabla_{\boldsymbol{\theta}_i^t} f_i^{\mathsf{KD}}(\boldsymbol{\theta}_i^t, \boldsymbol{\mu}_i^t)}{2\psi_i^t}$
11:     Update: $\boldsymbol{\theta}_i^{t+1} = \boldsymbol{\theta}_i^t - \eta_1 \boldsymbol{g}_i^t$
12:     Compute $\boldsymbol{h}_i^t := \nabla_{\boldsymbol{\mu}_i^t} f_i^{\mathsf{KD}}(\boldsymbol{\theta}_i^{t+1}, \boldsymbol{\mu}_i^t)/2\psi_i^t$
13:     Update: $\boldsymbol{\mu}_i^{t+1} = \boldsymbol{\mu}_i^t - \eta_2 \boldsymbol{h}_i^t$
14:     Compute $k_i^t := \frac{1}{2\psi_i^t} - f_i^{\mathsf{KD}}(\boldsymbol{\theta}_i^{t+1}, \boldsymbol{\mu}_i^{t+1})/2(\psi_i^t)^2$
15:     Update: $\psi_i^{t+1} = \psi_i^t - \eta_3 k_i^t$
16:     **if** $\tau$ divides $t + 1$ **then**
17:       Clients send $\boldsymbol{\mu}_i^t$ and $\psi_i^t$ to **Server**
18:       Server receives $\{\boldsymbol{\mu}_i^t\}_{i \in \mathcal{K}^t}$ and $\{\psi_i^t\}_{i \in \mathcal{K}^t}$
19:       Server computes $\boldsymbol{\mu}^{t+1} = \frac{1}{K} \sum_{i \in \mathcal{K}^t} \boldsymbol{\mu}_i^t$ and $\psi^{t+1} = \frac{1}{K} \sum_{i \in \mathcal{K}^t} \psi_i^t$
20:     **end if**
21: **end for**

**Output:** Personalized models $(\boldsymbol{\theta}_i^T)_{i=1}^m$

---

the private estimator's performance starts to become worse than the non-private estimator.

**Personalized Learning.** First we describe the experiment setting and then the results.

● **Experiment setting.** We consider image classification on MNIST, FEMNIST (Caldas et al., 2018), CIFAR-10, CIFAR-100 (experimental details for CIFAR-100 is given in Appendix K); and train a CNN, similar to the one considered in (McMahan et al., 2017), that has 2 convolutional and 3 fully connected layers. We set $m = 66$ for FEMNIST and $m = 50$ for MNIST, CIFAR-10, CIFAR-100. For FEMNIST, we use a subset of 198 writers so that each client has access to data from 3 authors, which results in a natural type of data heterogeneity due to writing styles of authors. On MNIST, CIFAR-10 we introduce pathological heterogeneity by letting each client sample data from 3 and 4 randomly selected classes only, respectively. We set $\tau = 10$ and vary the batch size so that each epoch consists of 60 iterations. On MNIST we train for 50 epochs, on CIFAR-10 for 250 epochs, on FEMNIST for 40 and 80 epochs, for 0.33 and 0.15 client sampling ratio, respectively. We discuss further details in Appendix K.

Table 1: Test accuracy (in %) for CNN model. The CIFAR-10, MNIST, and FEMNIST columns have client sampling ratios $\frac{K}{n}$ of 0.2, 0.1, and 0.15, respectively.

| Method | CIFAR-10 | CIFAR-100 | FEMNIST |
|---|---|---|---|
| FedAvg | $60.86 \pm 0.94$ | $30.48 \pm 0.33$ | $92.18 \pm 0.13$ |
| FedAvg+fine tuning (Jiang et al., 2019) | $63.12 \pm 0.31$ | $39.98 \pm 0.26$ | $94.12 \pm 0.26$ |
| AdaPeD (Ours) | $\mathbf{72.49} \pm 0.42$ | $\mathbf{53.11} \pm 0.34$ | $\mathbf{96.55} \pm 0.32$ |
| pFedMe (Dinh et al., 2020) | $69.53 \pm 0.16$ | $43.65 \pm 0.18$ | $94.95 \pm 0.55$ |
| Per-FedAvg (Fallah et al., 2020) | $59.95 \pm 0.79$ | $34.78 \pm 0.41$ | $93.51 \pm 0.31$ |
| QuPeD (FP) (Ozkara et al., 2021) | $71.61 \pm 0.70$ | $51.94 \pm 0.21$ | $95.99 \pm 0.08$ |
| Federated ML (Shen et al., 2020) | $71.09 \pm 0.67$ | $50.42 \pm 0.26$ | $95.12 \pm 0.18$ |

| Method | $\epsilon = 3.35$ | $\epsilon = 13.16$ | $\epsilon = 27.30$ |
|---|---|---|---|
| DP-FedAvg | $11.73 \pm 0.85$ | $29.91 \pm 1.28$ | $55.79 \pm 0.29$ |
| DP-AdaPeD (Ours) | $93.32 \pm 1.18$ | $98.51 \pm 0.90$ | $99.01 \pm 0.65$ |

Table 2: (DP-AdaPeD) Test Accuracy (in %) vs. $\epsilon$ on MNIST without client sampling.

| Method | $n = 10$ | $n = 20$ | $n = 30$ |
|---|---|---|---|
| Local Training | $39.93 \pm 0.13$ | $30.02 \pm 0.08$ | $19.97 \pm 0.07$ |
| AdaMix | $10.42 \pm 0.15$ | $3.12 \pm 0.04$ | $2.55 \pm 0.04$ |

Table 3: Mean squared error of our AdaMix algorithm and the local training for linear regression.

● **Results.** In Table 1 we compare AdaPeD against FedAvg (McMahan et al., 2017), FedAvg+ (Jiang et al., 2019) and various personalized FL algorithms: pFedMe (Dinh et al., 2020), Per-FedAvg (Fallah et al., 2020), QuPeD (Ozkara et al., 2021) without model compression, and Federated ML (Shen et al., 2020). We report further results in Appendix K. We observe AdaPeD consistently outperforms other methods. It can be seen that methods that use knowledge distillation perform better; on top of this, AdaPeD enables us adjust the dependence on collaboration according to the compatibility of global and local decisions/scores. For instance, we set $\sigma_\theta^2$ to a certain value initially, and observe it progressively decrease, which implies clients start to rely on the collaboration more and more. Interestingly, this is not always the case: for DP-AdaPeD, we first observe a decrease in $\sigma_\theta^2$ and later it increases. This suggests: while there is not much accumulated noise, clients prefer to collaborate, and as the noise accumulation on the global model increases due to DP noise, clients prefer not to collaborate. This is exactly the type of autonomous behavior we aimed with adaptive regularization.

● **DP-AdaPeD.** In Figure 1b and Table 2, we observe performance of DP-AdaPeD under different $\epsilon$ values. DP-AdaPeD outperforms DP-FedAvg because personalized models do not need to be privatized by DP mechanism, whereas the global model needs to be. Our experiments provide user-level privacy (more stringent, but appropriate in FL), as opposed to the item-level privacy.

● **DP-AdaPeD with unsampled client iterations.** When we let unsampled clients to do local iterations (free in terms of privacy cost and a realistic scenario in cross-silo settings) described in Appendix H, we can increase DP-AdaPeD's performance under more aggressive privacy constants $\epsilon$. For instance, for FEMNIST with $1/3$ client sampling we obtain the result reported in Figure 1b.

● **AdaMix.** We consider linear regression on synthetic data, with $m = 1000$ clients and each client has $n \in \{10, 20, 30\}$ local samples. Each local model $\boldsymbol{\theta}_i \in \mathbb{R}^d$ is drawn from a mixture of two Gaussian distributions $\mathcal{N}(\boldsymbol{\mu}, \Sigma)$ and $\mathcal{N}(-\boldsymbol{\mu}, \Sigma)$, where $\Sigma = 0.001 \times \mathbb{I}_d$ and $d = 50$. Each client sample $(X_{ij}, Y_{ij})$ is distributed as $X_{ij} \sim \mathcal{N}(0, \mathbb{I}_d)$ and $Y_{ij} = \langle X_{ij}, \boldsymbol{\theta}_i \rangle + w_{ij}$, where $w_{ij} \sim \mathcal{N}(0, 0.1)$. Table 3 demonstrates the superior performance of AdaMix against the local estimator.

## 5 CONCLUSION

We proposed a statistical framework leading to new personalized federated estimation and learning algorithms (e.g., AdaMix, AdaPeD); we also incorporated privacy (and communication) constraints into our algorithms and analyzed them. Open questions include information theoretic lower bounds and its comparison to proposed methods; examination of how far the proposed alternating minimization methods (such as in AdaMix, AdaPeD) are from global optima.

The work in this paper was partially supported by NSF grants 2139304, 2007714 and gift funding by Meta and Amazon.

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

# A  PRELIMINARIES

We give standard privacy definitions that we use in Section A.1, some existing results on RDP to DP conversion and RDP composition in Section A.2, and user-level differential privacy in Section A.3.

## A.1  PRIVACY DEFINITIONS

In this subsection, we define different privacy notions that we will use in this paper: local differential privacy (LDP), central different privacy (DP), and Renyi differential privacy (RDP), and their user-level counterparts.

**Definition 1** (Local Differential Privacy - LDP (Kasiviswanathan et al., 2011)). *For $\epsilon_0 \geq 0$, a randomized mechanism $\mathcal{R} : \mathcal{X} \to \mathcal{Y}$ is said to be $\epsilon_0$-local differentially private (in short, $\epsilon_0$-LDP), if for every pair of inputs $d, d' \in \mathcal{X}$, we have*

$$\Pr[\mathcal{R}(d) \in \mathcal{S}] \leq e^{\epsilon_0} \Pr[\mathcal{R}(d') \in \mathcal{S}], \qquad \forall \mathcal{S} \subset \mathcal{Y}. \tag{12}$$

Let $\mathcal{D} = \{x_1, \ldots, x_n\}$ denote a dataset comprising $n$ points from $\mathcal{X}$. We say that two datasets $\mathcal{D} = \{x_1, \ldots, x_n\}$ and $\mathcal{D}' = \{x_1', \ldots, x_n'\}$ are neighboring (and denoted by $\mathcal{D} \sim \mathcal{D}'$) if they differ in one data point, i.e., there exists an $i \in [n]$ such that $x_i \neq x_i'$ and for every $j \in [n], j \neq i$, we have $x_j = x_j'$.

**Definition 2** (Central Differential Privacy - DP (Dwork et al., 2006; Dwork & Roth, 2014)). *For $\epsilon, \delta \geq 0$, a randomized mechanism $\mathcal{M} : \mathcal{X}^n \to \mathcal{Y}$ is said to be $(\epsilon, \delta)$-differentially private (in short, $(\epsilon, \delta)$-DP), if for all neighboring datasets $\mathcal{D} \sim \mathcal{D}' \in \mathcal{X}^n$ and every subset $\mathcal{S} \subseteq \mathcal{Y}$, we have*

$$\Pr[\mathcal{M}(\mathcal{D}) \in \mathcal{S}] \leq e^{\epsilon_0} \Pr[\mathcal{M}(\mathcal{D}') \in \mathcal{S}] + \delta. \tag{13}$$

*If $\delta = 0$, then the privacy is referred to as pure DP.*

**Definition 3** (($\lambda, \epsilon(\lambda)$)-RDP (Renyi Differential Privacy) (Mironov, 2017)). *A randomized mechanism $\mathcal{M} : \mathcal{X}^n \to \mathcal{Y}$ is said to have $\epsilon(\lambda)$-Renyi differential privacy of order $\lambda \in (1, \infty)$ (in short, $(\lambda, \epsilon(\lambda))$-RDP), if for any neighboring datasets $\mathcal{D} \sim \mathcal{D}' \in \mathcal{X}^n$, the Renyi divergence between $\mathcal{M}(\mathcal{D})$ and $\mathcal{M}(\mathcal{D}')$ is upper-bounded by $\epsilon(\lambda)$, i.e.,*

$$D_\lambda(\mathcal{M}(\mathcal{D})||\mathcal{M}(\mathcal{D}')) = \frac{1}{\lambda - 1} \log \left( \mathbb{E}_{\theta \sim \mathcal{M}(\mathcal{D}')} \left[ \left( \frac{\mathcal{M}(\mathcal{D})(\theta)}{\mathcal{M}(\mathcal{D}')(\theta)} \right)^\lambda \right] \right)$$

$$\leq \epsilon(\lambda),$$

*where $\mathcal{M}(\mathcal{D})(\theta)$ denotes the probability that $\mathcal{M}$ on input $\mathcal{D}$ generates the output $\theta$. For convenience, instead of $\epsilon(\lambda)$ being an upper bound, we define it as $\epsilon(\lambda) = \sup_{\mathcal{D} \sim \mathcal{D}'} D_\lambda(\mathcal{M}(\mathcal{D})||\mathcal{M}(\mathcal{D}'))$.*

## A.2  RDP TO DP CONVERSION AND RDP COMPOSITION

As mentioned after Theorem 5, we can convert the RDP guarantees of `DP-AdaPeD` to its DP guarantees using existing conversion results from literature. To the best of our knowledge, the following gives the best conversion.

**Lemma 1** (From RDP to DP (Canonne et al., 2020; Balle et al., 2020)). *Suppose for any $\lambda > 1$, a mechanism $\mathcal{M}$ is $(\lambda, \epsilon(\lambda))$-RDP. Then, the mechanism $\mathcal{M}$ is $(\epsilon, \delta)$-DP, where $\epsilon, \delta$ are define below:*

*For a given $\delta \in (0, 1)$ :*

$$\epsilon = \min_\lambda \epsilon(\lambda) + \frac{\log(1/\delta) + (\lambda - 1)\log(1 - 1/\lambda) - \log(\lambda)}{\lambda - 1}$$

*For a given $\epsilon > 0$ :*

$$\delta = \min_\lambda \frac{\exp((\lambda - 1)(\epsilon(\lambda) - \epsilon))}{\lambda - 1} \left(1 - \frac{1}{\lambda}\right)^\lambda.$$

The main strength of RDP in comparison to other privacy notions comes from composition. The following result states that if we adaptively compose two RDP mechanisms with the same order, their privacy parameters add up in the resulting mechanism.

**Lemma 2** (Adaptive composition of RDP (Mironov, 2017, Proposition 1)). *For any $\lambda > 1$, let $\mathcal{M}_1 : \mathcal{X} \to \mathcal{Y}_1$ be a $(\lambda, \epsilon_1(\lambda))$-RDP mechanism and $\mathcal{M}_2 : \mathcal{Y}_1 \times \mathcal{X} \to \mathcal{Y}$ be a $(\lambda, \epsilon_2(\lambda))$-RDP mechanism. Then, the mechanism defined by $(\mathcal{M}_1, \mathcal{M}_2)$ satisfies $(\lambda, \epsilon_1(\lambda) + \epsilon_2(\lambda))$-RDP.*

### A.3 User-level Differential Privacy Levy et al. (2021)

Consider a set of $m$ users, each having a local dataset of $n$ samples. Let $\mathcal{D}_i = \{x_{i1}, \ldots, x_{in}\}$ denote the local dataset at the $i$-th user for $i \in [m]$, where $x_{ij} \in \mathcal{X}$ and $\mathcal{X} \subset \mathbb{R}^d$. We define $\mathcal{D} = (\mathcal{D}_1, \ldots, \mathcal{D}_m) \in (\mathcal{X}^n)^m$ as the entire dataset.

We have already defined DP, LDP, and RDP in Section A.1 w.r.t. the item-level privacy. Here, we extend those definition w.r.t. the user-level privacy. In order to do that, we need a generic neighborhood relation between datasets: We say that two datasets $\mathcal{D}, \mathcal{D}'$ are neighboring with respect to distance metric dis if we have $\mathsf{dis}(\mathcal{D}, \mathcal{D}') \leq 1$.

**Item-level DP/RDP vs. User-level DP/RDP.** By choosing $\mathsf{dis}(\mathcal{D}, \mathcal{D}') = \sum_{i=1}^{m} \sum_{j=1}^{n} \mathbb{1}\{x_{ij} \neq x'_{ij}\}$, we recover the standard definition of the DP/RDP from Definitions 2, 3, which we call *item-level* DP/RDP. In the item-level DP/RDP, two datasets $\mathcal{D}, \mathcal{D}'$ are neighboring if they differ in a single item. On the other hand, by choosing $\mathsf{dis}(\mathcal{D}, \mathcal{D}') = \sum_{i=1}^{m} \mathbb{1}\{\mathcal{D}_i \neq \mathcal{D}'_i\}$, we call it *user-level* DP/RDP, where two datasets $\mathcal{D}, \mathcal{D}' \in (\mathcal{X}^n)^m$ are neighboring when they differ in a local dataset of any single user. Observe that when each user has a single item ($n = 1$), then both item-level and user-level privacy are equivalent.

**User-level Local Differential Privacy (LDP).** When we have a single user (i.e., $m = 1$ and $\mathcal{D} = \mathcal{X}^n$), by choosing $\mathsf{dis}(\mathcal{D}, \mathcal{D}') = \mathbb{1}\{\mathcal{D} \neq \mathcal{D}'\}$ for $\mathcal{D}, \mathcal{D}' \in \mathcal{X}^n$, we call it *user-level LDP*. In this case each user privatize her own local dataset using a private mechanism.

We can define user-level LDP/DP/RDP analogously to their item-level counterparts using the neighborhood relation dis defined above.

## B Personalized Estimation – Gaussian Model

### B.1 Proof of Theorem 1

We will derive the optimal estimator and prove the MSE for one dimensional case, i.e., for $d = 1$; the final result can be obtained by applying these to each of the $d$ coordinates separately.

The posterior estimators of the local means $\theta_1, \ldots, \theta_m$ and the global mean $\mu$ are obtained by solving the following optimization problem:

$$
\begin{aligned}
\hat{\theta}_1, \ldots, \hat{\theta}_m, \hat{\mu} &= \underset{\theta_1, \ldots, \theta_m, \mu}{\arg\max}\, p_{\mathbf{X}|\theta}\left(X_1, \ldots, X_m | \theta_1, \ldots, \theta_m\right) p_{\theta|\mu}(\theta_1, \ldots, \theta_m | \mu) \\
&= \underset{\theta_1, \ldots, \theta_m, \mu}{\arg\min}\, -\log\left(p_{\mathbf{X}|\theta}\left(X_1, \ldots, X_m | \theta_1, \ldots, \theta_m\right)\right) - \log\left(p_{\theta|\mu}(\theta_1, \ldots, \theta_m | \mu)\right) \\
&= \underset{\theta_1, \ldots, \theta_m, \mu}{\arg\min}\, C + \sum_{i=1}^{m} \sum_{j=1}^{n} \frac{\left(X_i^j - \theta_i\right)^2}{\sigma_x^2} + \sum_{i=1}^{m} \frac{(\theta_i - \mu)^2}{\sigma_\theta^2},
\end{aligned}
$$

where the second equality is obtained from the fact that the $\log$ function is a monotonic function, and $C$ is a constant independent of the variables $\theta = (\theta_1, \ldots, \theta_m)$. Observe that the objective function $F(\theta, \mu) = \sum_{i=1}^{m} \sum_{j=1}^{n} \frac{\left(X_i^j - \theta_i\right)^2}{\sigma_x^2} + \sum_{i=1}^{m} \frac{(\theta_i - \mu)^2}{\sigma_\theta^2}$ is jointly convex in $(\theta, \mu)$. Thus, the optimal is obtained by setting the derivative to zero as it is an unbounded optimization problem.

$$
\begin{aligned}
\left.\frac{\partial F}{\partial \theta_i}\right|_{\mu = \hat{\mu}, \theta_i = \hat{\theta}_i} &= \frac{\sum_{j=1}^{n} 2(\hat{\theta}_i - X_i^j)}{\sigma_x^2} + \frac{2(\hat{\theta}_i - \hat{\mu})}{\sigma_\theta^2} = 0, \qquad \forall i \in [m] \\
\left.\frac{\partial F}{\partial \mu}\right|_{\mu = \hat{\mu}, \theta_i = \hat{\theta}_i} &= \frac{\sum_{i=1}^{m} 2(\hat{\mu} - \hat{\theta}_i)}{\sigma_\theta^2} = 0.
\end{aligned}
$$

By solving these $m + 1$ equations in $m + 1$ unknowns, we get:

$$
\hat{\theta}_i = \alpha \left(\frac{1}{n} \sum_{j=1}^{n} X_i^j\right) + (1 - \alpha) \left(\frac{1}{mn} \sum_{i=1}^{m} \sum_{j=1}^{n} X_i^j\right), \tag{14}
$$

where $\alpha = \frac{\sigma_\theta^2}{\sigma_\theta^2 + \frac{\sigma_x^2}{n}}$. By letting $\overline{X}_i = \frac{1}{n}\sum_{j=1}^n X_i^j$ for all $i \in [m]$, we can write $\hat{\theta}_i = \alpha \overline{X}_i + (1 - \alpha)\frac{1}{m}\sum_{i=1}^m \overline{X}_i$.

Observe that $\mathbb{E}\left[\hat{\theta}_i | \theta\right] = \alpha\theta_i + \frac{1-\alpha}{m}\sum_{l=1}^m \theta_l$, where $\theta = (\theta_1, \ldots, \theta_m)$. Thus, the estimator (14) is an unbiased estimate of $\{\theta_i\}$. Substituting the $\hat{\theta}_i$ in the MSE, we get that

$$
\begin{aligned}
\mathbb{E}_{X_1,\ldots,X_m}\left[\left(\hat{\theta}_i - \theta_i\right)^2\right] &= \mathbb{E}_\theta\left[\mathbb{E}_{X_1,\ldots,X_m}\left[\left(\hat{\theta}_i - \theta_i\right)^2 |\theta\right]\right] \\
&= \mathbb{E}_\theta\left[\mathbb{E}_{X_1,\ldots,X_m}\left[\left(\hat{\theta}_i - \mathbb{E}\left[\hat{\theta}_i|\theta\right] + \mathbb{E}\left[\hat{\theta}_i|\theta\right] - \theta_i\right)^2 |\theta\right]\right] \\
&= \mathbb{E}_\theta\left[\mathbb{E}_{X_1,\ldots,X_m}\left[\left(\hat{\theta}_i - \mathbb{E}\left[\hat{\theta}_i|\theta\right]\right)^2 |\theta\right]\right] + \mathbb{E}_\theta\left[\mathbb{E}_{X_1,\ldots,X_m}\left[\left(\mathbb{E}\left[\hat{\theta}_i|\theta\right] - \theta_i\right)^2 |\theta\right]\right]
\end{aligned}
$$
$$\tag{15}$$

**Claim 1.**

$$\mathbb{E}_\theta\left[\mathbb{E}_{X_1,\ldots,X_m}\left[\left(\hat{\theta}_i - \mathbb{E}\left[\hat{\theta}_i|\theta\right]\right)^2 |\theta\right]\right] = \alpha^2\frac{\sigma_x^2}{n} + (1-\alpha)^2\frac{\sigma_x^2}{mn} + 2\alpha(1-\alpha)\frac{\sigma_x^2}{mn}$$

$$\mathbb{E}_\theta\left[\mathbb{E}_{X_1,\ldots,X_m}\left[\left(\mathbb{E}\left[\hat{\theta}_i|\theta\right] - \theta_i\right)^2 |\theta\right]\right] = (1-\alpha)^2\mathbb{E}_\theta\left[\left(\frac{1}{m}\sum_{k=1}^m \theta_k - \theta_i\right)^2\right] \leq (1-\alpha)^2\frac{\sigma_\theta^2(m-1)}{m}$$

*Proof.* For the first equation:

$$
\begin{aligned}
\mathbb{E}_\theta\left[\mathbb{E}_{X_1,\ldots,X_m}\left[\left(\hat{\theta}_i - \mathbb{E}\left[\hat{\theta}_i|\theta\right]\right)^2 |\theta\right]\right] &= \mathbb{E}_\theta\left[\mathbb{E}_{X_1,\ldots,X_m}\left[\left(\alpha(\overline{X}_i - \theta_i) + (1-\alpha)\frac{1}{m}\sum_{k=1}^m(\overline{X}_k - \theta_k)\right)^2 |\theta\right]\right] \\
&= \alpha^2\mathbb{E}\left[\mathbb{E}\left[(\overline{X}_i - \theta_i)^2 | \theta\right]\right] + (1-\alpha)^2\mathbb{E}\left[\mathbb{E}\left[\left(\frac{1}{m}\sum_{k=1}^m(\overline{X}_k - \theta_k)\right)^2 | \theta\right]\right] \\
&\quad + 2\alpha(1-\alpha)\mathbb{E}\left[\mathbb{E}\left[\frac{1}{m}\sum_{k=1}^m(\overline{X}_i - \theta_i)(\overline{X}_k - \theta_k) | \theta\right]\right] \\
&= \alpha^2\frac{\sigma_x^2}{n} + (1-\alpha)^2\frac{\sigma_x^2}{mn} + 2\alpha(1-\alpha)\frac{\sigma_x^2}{mn}
\end{aligned}
$$

For the second equation, first note that $\mathbb{E}\left[\hat{\theta}_i|\theta\right] - \theta_i = \alpha\theta_i + \frac{1-\alpha}{m}\sum_{k=1}^m \theta_k - \theta_i = (1-\alpha)\left(\frac{1}{m}\sum_{k=1}^m \theta_k - \theta_i\right)$:

$$
\begin{aligned}
\mathbb{E}_\theta\left[\mathbb{E}_{X_1,\ldots,X_m}\left[\left(\mathbb{E}\left[\hat{\theta}_i|\theta\right] - \theta_i\right)^2 |\theta\right]\right] &= (1-\alpha)^2\mathbb{E}\left[\left(\frac{1}{m}\sum_{k=1}^m \theta_k - \theta_i\right)^2\right] \\
&= \frac{(1-\alpha)^2}{m^2}\mathbb{E}\left[\left(\sum_{k\neq i}(\theta_k - \theta_i)\right)^2\right] \\
&= \frac{(1-\alpha)^2}{m^2}\left[\sum_{k\neq i}\mathbb{E}(\theta_k - \theta_i)^2 + \sum_{k\neq i, l\neq i, k\neq l}\mathbb{E}(\theta_k - \theta_i)(\theta_l - \theta_i)\right] \\
&\leq \frac{(1-\alpha)^2}{m^2}\left[\sum_{k\neq i}[\mathbb{E}(\theta_k - \mu)^2 + \mathbb{E}(\theta_i - \mu)^2] + \sum_{k\neq i, l\neq i, k\neq l}\mathbb{E}(\theta_k - \theta_i)(\theta_l - \theta_i)\right] \\
&= \frac{(1-\alpha)^2}{m^2}\left[2(m-1)\sigma_\theta^2 + \sum_{k\neq i, l\neq i, k\neq l}\mathbb{E}(\theta_k - \theta_i)(\theta_l - \theta_i)\right]
\end{aligned}
$$

$$= \frac{(1-\alpha)^2}{m^2} \left[ 2(m-1)\sigma_\theta^2 + \sum_{k \neq i, l \neq i, k \neq l} \mathbb{E}(\mu - \theta_i)^2 \right] \quad \text{(Since } \mathbb{E}[\theta_k] = \mu \text{ for all } k \in [m])$$

$$= \frac{(1-\alpha)^2}{m^2} \left[ 2(m-1)\sigma_\theta^2 + (m-1)(m-2)\sigma_\theta^2 \right]$$

$$= (1-\alpha)^2 \frac{\sigma_\theta^2 (m-1)}{m}$$

This concludes the proof of Claim 1. $\qquad\square$

Substituting the result of Claim 1 into (15), we get

$$\mathbb{E}_{X_1, \ldots, X_m} \left[ \left( \hat{\theta}_i - \theta_i \right)^2 \right] \leq \alpha^2 \frac{\sigma_x^2}{n} + (1-\alpha)^2 \frac{\sigma_x^2}{mn} + 2\alpha(1-\alpha) \frac{\sigma_x^2}{mn} + (1-\alpha)^2 \frac{\sigma_\theta^2 (m-1)}{m} \quad (16)$$

$$\stackrel{(a)}{=} \frac{\sigma_x^2}{n} \left( \alpha^2 + \frac{(1-\alpha)^2 + 2\alpha(1-\alpha)}{m} + \alpha(1-\alpha)\frac{m-1}{m} \right)$$

$$= \frac{\sigma_x^2}{n} \left( \alpha + \frac{1-\alpha}{m} \right),$$

where in (a) we used $\alpha = \frac{\sigma_\theta^2}{\sigma_\theta^2 + \frac{\sigma_x^2}{n}}$ for the last term to write $(1-\alpha)^2 \frac{\sigma_\theta^2 (m-1)}{m} = \frac{\sigma_x^2}{n} \alpha(1-\alpha)\frac{m-1}{m}$.

Observe that the estimator in (14) is a weighted summation between two estimators: the local estimator $\overline{X}_i = \frac{1}{n} \sum_{j=1}^n X_i^j$, and the global estimator $\hat{\mu} = \frac{1}{m} \sum_{i=1}^m \overline{X}_i$. Thus, the MSE in (a) consists of four terms: 1) The variance of the local estimator ($\frac{\sigma_x^2}{n}$). 2) The variance of the global estimator ($\frac{\sigma_x^2}{nm}$). 3) The correlation between the local estimator and the global estimator ($\frac{\sigma_x^2}{nm}$). 4) The bias term $\mathbb{E}_\theta \left[ \mathbb{E}_{X_1, \ldots, X_m} \left[ \left( \mathbb{E} \left[ \hat{\theta}_i | \theta \right] - \theta_i \right)^2 | \theta \right] \right]$. This completes the proof of Theorem 1.

## B.2 Proof of Theorem 2, Equation (5)

Similar to the proof of Theorem 1, here also we will derive the optimal estimator and prove the MSE for the one dimensional case, and the final result can be obtained by applying these to each of the $d$ coordinates separately.

Let $\theta = (\theta_1, \ldots, \theta_m)$ denote the personalized models vector. For given a constraint function $q$, we set the personalized model as follows:

$$\hat{\theta}_i = \alpha \left( \frac{1}{n} \sum_{j=1}^n X_i^j \right) + (1-\alpha) \left( \frac{1}{m} \sum_{i=1}^m q(\overline{X}_i) \right) \qquad \forall i \in [m], \qquad (17)$$

where $\overline{X}_i = \frac{1}{n} \sum_{j=1}^n X_i^j$. From the second condition on the function $q$, we get that

$$\mathbb{E} \left[ \hat{\theta}_i | \theta \right] = \alpha \theta_i + \frac{1-\alpha}{m} \sum_{l=1}^m \theta_l, \qquad (18)$$

Thus, by following similar steps as the proof of Theorem 1, we get that:

$$\mathbb{E} \left[ \left( \hat{\theta}_i - \theta_i \right)^2 \right] = \mathbb{E} \left[ \mathbb{E} \left[ \left( \hat{\theta}_i - \theta_i \right)^2 | \theta \right] \right]$$

$$= \mathbb{E} \left[ \mathbb{E} \left[ \left( \hat{\theta}_i - \mathbb{E} \left[ \hat{\theta}_i | \theta \right] + \mathbb{E} \left[ \hat{\theta}_i | \theta \right] - \theta_i \right)^2 | \theta \right] \right]$$

$$= \mathbb{E} \left[ \mathbb{E} \left[ \left( \hat{\theta}_i - \mathbb{E} \left[ \hat{\theta}_i | \theta \right] \right)^2 | \theta \right] \right] + \mathbb{E} \left[ \mathbb{E} \left[ \left( \mathbb{E} \left[ \hat{\theta}_i | \theta \right] - \theta_i \right)^2 | \theta \right] \right]$$

$$\stackrel{(a)}{=} \alpha^2 \frac{\sigma_x^2}{n} + (1-\alpha)^2 \mathbb{E} \left[ \left( \frac{1}{m} \sum_{l=1}^m q(\overline{X}_l) - \theta_l \right)^2 | \theta \right]$$

$$+ 2\alpha(1-\alpha)\mathbb{E}\left[(\overline{X}_i - \theta_i)\left(\frac{1}{m}\sum_{l=1}^{m} q(\overline{X}_l) - \theta_l\right)|\theta\right] + (1-\alpha)^2\mathbb{E}\left[\left(\frac{1}{m}\sum_{k=1}^{m}\theta_k - \theta_i\right)^2\right]$$

$$\overset{(b)}{=} \alpha^2\frac{\sigma_x^2}{n} + \frac{(1-\alpha)^2\left(\frac{\sigma_x^2}{n} + \sigma_q^2\right)}{m} + \frac{2\alpha(1-\alpha)\sigma_x^2}{mn} + (1-\alpha)^2\mathbb{E}\left[\left(\frac{1}{m}\sum_{k=1}^{m}\theta_k - \theta_i\right)^2\right]$$

$$\leq \alpha^2\frac{\sigma_x^2}{n} + \frac{(1-\alpha)^2\left(\frac{\sigma_x^2}{n} + \sigma_q^2\right)}{m} + 2\alpha(1-\alpha)\frac{\sigma_x^2}{mn} + (1-\alpha)^2\frac{\sigma_\theta^2(m-1)}{m}$$

$$\overset{(c)}{=} \frac{\sigma_x^2}{n}\left(\alpha^2 + \frac{(1-\alpha)^2 + 2\alpha(1-\alpha)}{m} + \alpha(1-\alpha)\frac{m-1}{m}\right)$$

$$= \frac{\sigma_x^2}{n}\left(\alpha + \frac{1-\alpha}{m}\right), \tag{19}$$

where step (a) follows by substituting the expectation of the personalized model from (18). Step (b) follows from the first and third conditions of the function $q$. Step (c) follows by choosing $\alpha = \frac{\sigma_\theta^2 + \frac{\sigma_q^2}{m-1}}{\sigma_\theta^2 + \frac{\sigma_q^2}{m-1} + \frac{\sigma_x^2}{n}}$. This derives the result stated in (5) in Theorem 2.

### B.2.1   PROOF OF THEOREM 2, PART 1

The proof consists of two steps. First, we use the concentration property of the Gaussian distribution to show that the local sample means $\{\overline{X}_i\}$ are bounded within a small range with high probability. Second, we apply an unbiased stochastic quantizer on the projected sample mean.

The local samples $X_i^1, \ldots, X_i^n$ are drawn i.i.d. from a Gaussian distribution with mean $\theta_i$ and variance $\sigma_x^2$, and hence, we have that $\overline{X}_i \sim \mathcal{N}(\theta_i, \frac{\sigma_x^2}{n})$. Thus, from the concentration property of the Gaussian distribution, we get that $\Pr[|\overline{X}_i - \theta_i| > c_1] \leq \exp\left(-\frac{nc_1^2}{\sigma_x^2}\right)$ for all $i \in [m]$. Similarly, the models $\theta_1, \ldots, \theta_m$ are drawn i.i.d. from a Gaussian distribution with mean $\mu \in [-r, r]$ and variance $\sigma_\theta^2$, hence,, we get $\Pr[|\theta_i - \mu| > c_2] \leq \exp\left(-\frac{c_2^2}{\sigma_\theta^2}\right)$ for all $i \in [m]$. Let $\mathcal{E} = \{\overline{X}_i \in [-a, a] : \forall i \in [m]\}$, where $a = r + c_1 + c_2$. Thus, from the union bound, we get that $\Pr[\mathcal{E}] > 1 - m(e^{-\frac{nc_1^2}{\sigma_x^2}} + e^{-\frac{c_2^2}{\sigma_\theta^2}})$. By setting $c_1 = \sqrt{\frac{\sigma_x^2}{n}\log(m^2n)}$ and $c_2 = \sqrt{\sigma_\theta^2\log(m^2n)}$, we get that $a = r + \frac{\sigma_x}{\sqrt{n}}\sqrt{\log(m^2n)} + \sigma_\theta\sqrt{\log(m^2n)}$, and $\Pr[\mathcal{E}] = 1 - \frac{2}{mn}$.

Let $q_k : [-a, a] \to \mathcal{Y}_k$ be a quantization function with $k$-bits, where $\mathcal{Y}_k$ is a discrete set of cardinality $|\mathbb{Y}_k| = 2^k$. For given $x \in [-a, a]$, the output of the function $q_k$ is given by:

$$q_k(x) = \frac{2a}{2^k - 1}\left(\lfloor\tilde{x}\rfloor + \mathsf{Bern}\left(\tilde{x} - \lfloor\tilde{x}\rfloor\right)\right) - a, \tag{20}$$

where $\mathsf{Bern}(p)$ is a Bernoulli random variable with bias $p$, and $\tilde{x} = \frac{2^k - 1}{2a}(x + a) \in [0, 2^k - 1]$. Observe that the output of the function $q_k$ requires only $k$-bits for transmission. Furthermore, the function $q_k$ satisfies the following conditions:

$$\mathbb{E}\left[q_k(x)\right] = x, \tag{21}$$

$$\sigma_{q_k}^2 = \mathbb{E}\left[(q_k(x) - x)^2\right] \leq \frac{a^2}{(2^k - 1)^2}. \tag{22}$$

Let each client applies the function $q_k$ on the projected local mean $\tilde{X}_i = \mathsf{Proj}_{[-a,a]}\left[\overline{X}_i\right]$ and sends the output to the server for all $i \in [m]$. Conditioned on the event $\mathcal{E}$, i.e., $\overline{X}_i \in [-a, a] \quad \forall i \in [m]$, and using (19), we get that

$$MSE = \mathbb{E}_{\theta,\mathbf{X}}\left[\left(\hat{\theta}_i - \theta_i\right)^2\right] \leq \frac{\sigma_x^2}{n}\left(\frac{1-\alpha}{m} + \alpha\right), \tag{23}$$

where $\alpha = \frac{\sigma_\theta^2 + \frac{a^2}{(2^k-1)^2(m-1)}}{\sigma_\theta^2 + \frac{a^2}{(2^k-1)^2(m-1)} + \frac{\sigma_x^2}{n}}$ and $a = r + \frac{\sigma_x}{\sqrt{n}}\sqrt{\log(m^2 n)} + \sigma_\theta \sqrt{\log(m^2 n)}$. Note that the event $\mathcal{E}$ happens with probability at least $1 - \frac{2}{mn}$.

### B.2.2 PROOF OF THEOREM 2, PART 2

We define the (random) mechanism $q_p : [-a, a] \to \mathcal{R}$ that takes an input $x \in [-a, a]$ and generates a user-level $(\epsilon_0, \delta)$-LDP output $y \in \mathbb{R}$, where $y = q_p(x)$ is given by:

$$q_p(x) = x + \nu, \tag{24}$$

where $\nu \sim \mathcal{N}(0, \sigma_{\epsilon_0}^2)$ is a Gaussian noise. By setting $\sigma_{\epsilon_0}^2 = \frac{8a^2 \log(2/\delta)}{\epsilon_0^2}$, we get that the output of the function $q_p(x)$ is $(\epsilon_0, \delta)$-LDP from Dwork & Roth (2014). Furthermore, the function $q_p$ satisfies the following conditions:

$$\mathbb{E}\left[q_p(x)\right] = x, \tag{25}$$

$$\sigma_{q_p}^2 = \mathbb{E}\left[(q_p(x) - x)^2\right] \leq \frac{8a^2 \log(2/\delta)}{\epsilon_0^2}. \tag{26}$$

Similar to the proof of Theorem 2, Part 1, let each client applies the function $q_p$ on the projected local mean $\tilde{X}_i = \mathsf{Proj}_{[-a,a]}\left[\overline{X}_i\right]$ and sends the output to the server for all $i \in [m]$. Conditioned on the event $\mathcal{E}$, i.e., $\overline{X}_i \in [-a, a] \quad \forall i \in [m]$, and using (19), we get that

$$\mathrm{MSE} = \mathbb{E}_{\theta, \mathbf{X}}\left[\left(\hat{\theta}_i - \theta_i\right)^2\right] \leq \frac{\sigma_x^2}{n}\left(\frac{1-\alpha}{m} + \alpha\right), \tag{27}$$

where $\alpha = \frac{\sigma_\theta^2 + \frac{8a^2 \log(2/\delta)}{\epsilon_0^2(m-1)}}{\sigma_\theta^2 + \frac{8a^2 \log(2/\delta)}{\epsilon_0^2(m-1)} + \frac{\sigma_x^2}{n}}$ and $a = r + \frac{\sigma_x}{\sqrt{n}}\sqrt{\log(m^2 n)} + \sigma_\theta \sqrt{\log(m^2 n)}$. Note that the event $\mathcal{E}$ happens with probability at least $1 - \frac{2}{mn}$.

**Remark 4** (Privacy with communication efficiency). *Note that our private estimation algorithm for the Gaussian case adds Gaussian noise (which is a real number) but that can also be made communication-efficient by alternatively adding a discrete Gaussian noise (Canonne et al., 2020).*

### B.3 LOWER BOUND

Here we discuss the lower bound using Fisher information technique similar to Barnes et al. (2020). In particular we use a Bayesian version of Cramer-Rao lower bound and van Trees inequality Gill & Levit (1995). Let us denote $f(X|\theta)$ as the data generating conditional density function and $\pi(\theta)$ as the prior distribution that generates $\theta$. Let us denote $\mathbb{E}_\theta$ as the expectation with respect to the randomness of $\theta$ and $\mathbb{E}$ as the expectation with respect to randomness of $X$ and $\theta$. First we define two types of Fisher information:

$$I_X(\theta) = \mathbb{E}_\theta \nabla_\theta \log(f(X|\theta)) \nabla_\theta \log(f(X|\theta))^T$$
$$I(\pi) = \mathbb{E} \nabla_\theta \log(\pi(\theta)) \nabla_\theta \log(\pi(\theta))^T$$

namely Fisher information of estimating $\theta$ from samples $X$ and Fisher information of prior $\pi$. Here the logarithm is elementwise. For van Trees inequality we need the following regularity conditions:

- $f(X|\cdot)$ and $\pi(\cdot)$ are absolutely continuous and $\pi(\cdot)$ vanishes at the end points of $\Theta$.
- $\mathbb{E}_\theta \nabla_\theta \log(f(X|\theta)) = 0$
- We also assume both density functions are continuously differentiable.

These assumptions are satisfied for the Gaussian setting for any finite mean $\mu$, they are satisfied for Bernoulli setting as long as parameters $\alpha$ and $\beta$ are larger than 1. Assuming local samples $X$ are generated i.i.d with $f(x|\theta)$, the van Trees inequality for one dimension is as follows:

$$\mathbb{E}(\widehat{\theta}(X) - \theta)^2 \geq \frac{1}{n\mathbb{E}I_x(\theta) + I(\pi)}$$

where $I_X(\theta) = \mathbb{E}_\theta \log(f(X|\theta))'^2$ and $I(\pi) = \mathbb{E}\log(\pi(\theta))'^2$. Assuming $\theta \in \mathbb{R}^d$ and each dimension is independent from each other, by Gill & Levit (1995) we have:

$$\mathbb{E}\|\widehat{\theta}(X) - \theta\|^2 \geq \frac{d^2}{n\mathbb{E}\mathrm{Tr}(I_x(\theta)) + \mathrm{Tr}(I(\pi))} \tag{28}$$

Note, the lower bound on the average risk directly translates as a lower bound on $\sup_{\theta \in \Theta} \mathbb{E}_X \|\widehat{\theta}(X) - \theta\|^2$. Before our proof we have a useful fact:

**Fact 1.** *Given some random variable $X \sim N(Y, \sigma_y^2)$ where $Y \sim N(Z, \sigma_z^2)$ we have $X \sim N(z, \sigma_z^2 + \sigma_y^2)$.*

*Proof.* We will give the proof in one dimension, however, it can easily be extended to multidimensional case where each dimension is independent. For all $t \in \mathbb{R}$ we have,

$$\mathbb{E}_X[\exp(itX)] = \mathbb{E}_Y \mathbb{E}_X[\exp(itX)|Y] = \mathbb{E}_Y[\exp(itY - \frac{\sigma_x^2 t^2}{2})]$$

$$= \exp(-\frac{\sigma_x^2 t^2}{2})\mathbb{E}_Y[\exp(itY)]$$

$$= \exp(-\frac{\sigma_x^2 t^2}{2})\exp(itz - \frac{\sigma_y^2 t^2}{2})$$

$$= \exp(itz - \frac{(\sigma_x^2 + \sigma_y^2)t^2}{2})$$

where the last line is the characteristic function of a Gaussian with mean $z$ and variance $\sigma_x^2 + \sigma_y^2$. $\square$

**Gaussian case with perfect knowledge of prior.** In this setting we know that $\theta_i \sim N(\mu\mathbf{1}, \sigma_\theta^2\mathrm{I}_d)$, hence, $I(\pi) = \frac{1}{\sigma_\theta^2}\mathrm{I}_d$, similarly $I_X(\theta) = \frac{1}{\sigma_x^2}\mathrm{I}_d$. Then,

$$\sup_{\theta_i} \mathbb{E}\|\widehat{\theta}_i(X) - \theta_i\|^2 \geq \frac{d^2}{n\mathbb{E}\frac{d}{\sigma_x^2} + \frac{d}{\sigma_\theta^2}} = \frac{d\sigma_\theta^2\sigma_x^2}{n\sigma_\theta^2 + \sigma_x^2} \tag{29}$$

**Gaussian case with estimated population mean.** In this setting instead of a true prior we have a prior whose mean is the average of all data spread across clients, i.e., we assume $\theta_i \sim N(\widehat{\mu}, \sigma_\theta^2\mathrm{I}_d)$ where $\widehat{\mu} = \frac{1}{mn}\sum_{i,j}^{m,n} X_i^j$. We additionally know that there is a Markov relation such that $X_i^j|\theta_j \sim N(\theta_j, \sigma_x^2\mathrm{I}_d)$ and $\theta_j \sim N(\mu, \sigma_\theta^2\mathrm{I}_d)$. While the true prior is parameterized with mean $\mu$, $\theta_i$ in this form is not parameterized by $\mu$ but by $\widehat{\mu}$ which itself has randomness due $X_i^j$. However, using Fact 1 twice we can write $\theta_i \sim N(\mu, (\sigma_\theta^2 + \frac{\sigma_\theta^2}{m} + \frac{\sigma_x^2}{mn})\mathrm{I}_d)$. Then using the van Trees inequality similar to the lower bound in perfect case we can obtain:

$$\sup_{\theta_i \in \Theta} \mathbb{E}_X\|\widehat{\theta}_i(X) - \theta_i\|^2 \geq d\frac{\sigma_\theta^2\sigma_x^2 + \frac{\sigma_x^4}{mn}}{n\sigma_\theta^2 + \sigma_x^2} \tag{30}$$

## C  PERSONALIZED ESTIMATION – BERNOULLI MODEL

### C.1  WHEN $\alpha, \beta$ ARE KNOWN

Analogous to the Gaussian case, we can show that if $\alpha$, $\beta$ are known, then the posterior mean estimator has a closed form expression: $\widehat{p}_i = a\overline{X}_i + (1 - a)\frac{\alpha}{\alpha+\beta}$ (where $a = {}^n/_{\alpha+\beta+n}$) and achieves the MSE: $\mathbb{E}_{p_i \sim \pi}\mathbb{E}_{\widehat{p}_i, X_1, \dots, X_m}(\widehat{p}_i - p_i)^2 \le \frac{\alpha\beta}{n(\alpha+\beta)(\alpha+\beta+1)}\frac{n}{\alpha+\beta+n}$. We show this below.

For a client $i$, let $\pi(p_i)$ be distributed as $\text{Beta}(\alpha, \beta)$. In this setting, we model that each client generates local samples according to $\text{Bern}(p_i)$. Consequently, each client has a Binomial distribution regarding the sum of local data samples. Estimating Bernoulli parameter $p_i$ is related to Binomial distribution $\text{Bin}(n, p_i)$ (the sum of data samples) $Z_i$ since it is the sufficient statistic of Bernoulli distribution. The distribution for Binomial variable $Z_i$ given $p_i$ is $P(Z_i = z_i | p_i) = \binom{n}{z_i}p_i^{z_i}(1 - p_i)^{n-z_i}$. It is a known fact that for any prior, the Bayesian MSE risk minimizer is the posterior mean $\mathbb{E}[p_i | Z_i = z_i]$.

When $p_i \sim \text{Beta}(\alpha, \beta)$, we have posterior

$$
\begin{aligned}
f(p_i | Z_i = z_i) &= \frac{P(z_i | p_i)}{P(z_i)}\pi(p_i) \\
&= \frac{\binom{n}{z_i}p_i^{z_i}(1 - p_i)^{n-z_i}}{P(z_i)}\frac{p_i^{\alpha-1}(1 - p_i)^{\beta-1}}{B(\alpha, \beta)} \\
&= \frac{\binom{n}{z_i}}{P(z_i)}\frac{B(\alpha + z_i, \beta + n - z_i)}{B(\alpha, \beta)}\frac{p_i^{\alpha+z_i-1}(1 - p_i)^{\beta+n-z_i-1}}{B(\alpha + z_i, \beta + n - z_i)},
\end{aligned}
$$

where $B(\alpha, \beta) = \frac{\Gamma(\alpha)\Gamma(\beta)}{\Gamma(\alpha+\beta)}$, and

$$
\begin{aligned}
P(z_i) &= \int P(z_i | p_i)\pi(p_i)dp_i \\
&= \int \binom{n}{z_i}p_i^{z_i}(1 - p_i)^{n-z_i}\frac{p_i^{\alpha-1}(1 - p_i)^{\beta-1}}{B(\alpha, \beta)}dp_i \\
&= \binom{n}{z_i}\frac{B(z_i + \alpha, n - z_i + \beta)}{B(\alpha, \beta)}\underbrace{\int \frac{p_i^{\alpha+z_i-1}(1 - p_i)^{\beta+n-z_i-1}}{B(\alpha + z_i, \beta + n - z_i)}dp_i}_{\text{integral of a Beta distribution}} \\
&= \binom{n}{z_i}\frac{B(z_i + \alpha, n - z_i + \beta)}{B(\alpha, \beta)}
\end{aligned}
$$

Thus, we get that the posterior distribution $f(p_i | Z_i = z_i) = \frac{p_i^{\alpha+z_i-1}(1-p_i)^{\beta+n-z_i-1}}{B(\alpha+z_i, \beta+n-z_i)}$ is a beta distribution $\text{Beta}(z_i + \alpha, n - z_i + \beta)$. As a result, the posterior mean is given by:

$$
\widehat{p}_i = \frac{\alpha + Z_i}{\alpha + \beta + n} = a\left(\frac{Z_i}{n}\right) + (1 - a)\left(\frac{\alpha}{\alpha + \beta}\right), \tag{31}
$$

where $a = \frac{n}{\alpha+\beta+n}$. Observe that $\mathbb{E}_{p_i \sim \text{Beta}(\alpha, \beta)}[p_i] = \frac{\alpha}{\alpha+\beta}$, i.e., the estimator is a weighted summation between the local estimator $\frac{z_i}{n}$ and the global estimator $\mu = \frac{\alpha}{\alpha+\beta}$.

We have $R_{p_i}(\widehat{p}_i) = \mathbb{E}_\pi\mathbb{E}(\widehat{p}_i - p_i)^2$. The MSE of the posterior mean is given by:

$$
\begin{aligned}
\text{MSE} &= \mathbb{E}[(\hat{p}_i - p_i)^2] \\
&= \mathbb{E}\left[\left(a\left(\frac{z_i}{n} - p_i\right) + (1 - a)(\mu - p_i)\right)^2\right] \\
&= a^2\mathbb{E}\left[\left(\frac{z_i}{n} - p_i\right)^2\right] + (1 - a)^2\mathbb{E}\left[(\mu - p_i)^2\right] \\
&= a^2\mathbb{E}_{p_i \sim \pi(p_i)}\left[\frac{p_i(1 - p_i)}{n}\right] + (1 - a)^2\frac{\alpha\beta}{(\alpha + \beta)^2(\alpha + \beta + 1)}
\end{aligned}
$$

$$= a^2 \frac{\alpha\beta}{n(\alpha+\beta)(\alpha+\beta+1)} + (1-a)^2 \frac{\alpha\beta}{(\alpha+\beta)^2(\alpha+\beta+1)}$$

$$= \frac{\alpha\beta}{n(\alpha+\beta)(\alpha+\beta+1)} \left( \frac{n}{\alpha+\beta+n} \right).$$

The last equality is obtained by setting $a = \frac{n}{\alpha+\beta+n}$.

**Remark 5.** *Note that $\overline{X}_i := \frac{Z_i}{n}$ is the estimator based only on the local data and $\alpha/(\alpha+\beta)$ is the true global mean, and $\widehat{p}_i = a\overline{X}_i + (1-a)\frac{\alpha}{\alpha+\beta}$, where $a = n/\alpha+\beta+n$ (see (31)) is the estimator based on all the data. Observe that when $n \to \infty$, then $a \to 1$, which implies that $\widehat{p}_i \to \overline{X}_i$. Otherwise, when $\alpha + \beta$ is large (i.e., the variance of the beta distribution is small), then $a \to 0$, which implies that $\widehat{p}_i \to \alpha/(\alpha+\beta)$. Both these conclusions conform to the conventional wisdom as mentioned in the Gaussian case. It can be shown that the local estimate $\overline{X}_i$ achieves the Bayesian risk of $\mathbb{E}_{p_i \sim Beta(\alpha,\beta)}\mathbb{E}_{X_i}[(\overline{X}_i - p_i)^2] = \mathbb{E}_{p_i \sim Beta(\alpha,\beta)}(p_i(1-p_i))/n = \alpha\beta/n(\alpha+\beta)(\alpha+\beta+1)$, which implies that the personalized estimation with perfect prior always outperforms the local estimate with a multiplicative gain $a = n/(n+\alpha+\beta) \le 1$.*

## C.2 WHEN $\alpha, \beta$ ARE UNKNOWN: PROOF OF THEOREM 3

The personalized model of the $i$th client with unknown parameters $\alpha, \beta$ is given by:

$$\hat{p}_i = \overline{a}_i \overline{X}_i + (1-\overline{a}_i)(\hat{\mu}_i), \tag{32}$$

where $\overline{a}_i = \frac{n}{\frac{\hat{\mu}_i(1-\hat{\mu}_i)}{\hat{\sigma}_i^2}+n}$, the empirical mean $\hat{\mu}_i = \frac{1}{m-1}\sum_{l\neq i}\overline{X}_l$, and the empirical variance $\hat{\sigma}_i^2 = \frac{1}{m-2}\sum_{l\neq i}(\overline{X}_l - \hat{\mu}_i)^2$. From (Tian et al., 2017, Lemma 1), with probability $1 - \frac{1}{m^2 n}$, we get that

$$|\mu - \hat{\mu}_i| \le \sqrt{\frac{3\log(4m^2n)}{m-1}}$$

$$|\sigma^2 - \hat{\sigma}_i^2| \le \sqrt{\frac{3\log(4m^2n)}{m-1}},$$

where $\mu = \frac{\alpha}{\alpha+\beta}$, $\sigma^2 = \frac{\alpha\beta}{(\alpha+\beta)^2(\alpha+\beta+1)}$ are the true mean and variance of the beta distribution, respectively. Let $c = \sqrt{\frac{3\log(4m^2n)}{m-1}}$. Conditioned on the event $\mathcal{E} = \{|\mu - \hat{\mu}_i| \le c, |\sigma^2 - \hat{\sigma}_i^2| \le c : \forall i \in [m]\}$ that happens with probability at least $1 - \frac{1}{mn}$, we get that:

$$\mathbb{E}\left[(\hat{p}_i - p_i)^2 | Z_{-i}\right] = \overline{a}^2 \mathbb{E}\left[\left(\frac{Z_i}{n} - p_i\right)^2\right] + (1-\overline{a})^2 \mathbb{E}\left[(\hat{\mu}_i - p_i)^2 | Z_{-i}\right]$$

$$= \overline{a}^2 \left(\frac{\alpha\beta}{n(\alpha+\beta)(\alpha+\beta+1)}\right) + (1-\overline{a})^2 \left(\mathbb{E}\left[(\mu - p_i)^2\right] + (\mu - \hat{\mu}_i)^2\right)$$

$$= \overline{a}^2 \left(\frac{\alpha\beta}{n(\alpha+\beta)(\alpha+\beta+1)}\right) + (1-\overline{a})^2 \left(\frac{\alpha\beta}{(\alpha+\beta)^2(\alpha+\beta+1)} + (\mu - \hat{\mu}_i)^2\right)$$

$$\le \overline{a}^2 \left(\frac{\alpha\beta}{n(\alpha+\beta)(\alpha+\beta+1)}\right) + (1-\overline{a})^2 \left(\frac{\alpha\beta}{(\alpha+\beta)^2(\alpha+\beta+1)} + c^2\right),$$

where the expectation is with respect to $z_i \sim \text{Binom}(p_i, n)$ and $p_i \sim \text{Beta}(\alpha, \beta)$ and $Z_{-i} = \{z_1, \ldots, z_{i-1}, z_{i+1}, \ldots, z_m\}$ denotes the entire dataset except the $i$th client data ($z_i$). By taking the expectation with respect to the datasets $Z_{-i}$, we get that the MSE is bounded by:

$$\text{MSE} \le \mathbb{E}\left[\overline{a}^2\right]\left(\frac{\alpha\beta}{n(\alpha+\beta)(\alpha+\beta+1)}\right) + \mathbb{E}\left[(1-\overline{a})^2\right]\left(\frac{\alpha\beta}{(\alpha+\beta)^2(\alpha+\beta+1)} + \frac{3\log(4m^2n)}{m-1}\right),$$

with probability at least $1 - \frac{1}{mn}$. This completes the proof of Theorem 3.

## C.3 WITH PRIVACY CONSTRAINTS: PROOF OF THEOREM 4

First, we prove some properties of the private mechanism $q_p$. Observe that for any two inputs $x, x' \in [0, 1]$, we have that:

$$\frac{\Pr[q_p(x) = y]}{\Pr[q_p(x') = y]} = \frac{\frac{e^{\epsilon_0}}{e^{\epsilon_0}+1} - x\frac{e^{\epsilon_0}-1}{e^{\epsilon_0}+1}}{\frac{e^{\epsilon_0}}{e^{\epsilon_0}+1} - x'\frac{e^{\epsilon_0}-1}{e^{\epsilon_0}+1}} \le e^{\epsilon_0}, \tag{33}$$

for $y = \frac{-1}{e^{\epsilon_0}-1}$. Similarly, we can prove (33) for the output $y = \frac{e^{\epsilon_0}}{e^{\epsilon_0}-1}$. Thus, the mechanism $q_p$ is user-level $\epsilon_0$-LDP. Furthermore, for given $x \in [0, 1]$, we have that

$$\mathbb{E}[q_p(x)] = x. \tag{34}$$

Thus, the output of the mechanism $q_p$ is an unbiased estimate of the input $x$. From the Hoeffding's inequality for bounded random variables, we get that:

$$\Pr[|\hat{\mu}_i^{(p)} - \mu| > t] \le 2\exp\left(\frac{-3(e^{\epsilon_0}-1)^2(m-1)t^2}{(e^{\epsilon_0}+1)^2}\right)$$
$$\Pr[|\hat{\sigma}_i^{2(p)} - \sigma_2| > t] \le 2\exp\left(\frac{-3(e^{\epsilon_0}-1)^2(m-1)t^2}{(e^{\epsilon_0}+1)^2}\right) \tag{35}$$

Thus, we have that the event $\mathcal{E} = \{|\hat{\mu}_i^{(p)} - \mu| \le c_p, |\hat{\sigma}_i^{2(p)} - \sigma^2| \le c_p : \forall i \in [m]\}$ happens with probability at least $1 - \frac{1}{mn}$, where $c_p = \sqrt{\frac{(e^{\epsilon_0}+1)^2 \log(4m^2n)}{3(e^{\epsilon_0}-1)^2(m-1)}}$. By following the same steps as the non-private estimator, we get the fact that the MSE of the private model is bounded by:

$$\text{MSE} \le \mathbb{E}[\bar{a}^2]\left(\frac{\alpha\beta}{n(\alpha+\beta)(\alpha+\beta+1)}\right)$$
$$+ \mathbb{E}[(1-\bar{a})^2]\left(\frac{\alpha\beta}{(\alpha+\beta)^2(\alpha+\beta+1)} + \frac{(e^{\epsilon_0}+1)^2\log(4m^2n)}{3(e^{\epsilon_0}-1)^2(m-1)}\right), \tag{36}$$

where $\bar{a}^{(p)} = \frac{n}{\frac{\hat{\mu}_i^{(p)}(1-\hat{\mu}_i^{(p)})}{\hat{\sigma}_i^{2(p)}}+n}$ and the expectation is with respect to the clients data $\{z_1, \ldots, z_{i-1}, z_{i+1}, \ldots, z_m\}$ and the randomness of the private mechanism $q_p$. This completes the proof of Theorem 4.

**Remark 6** (Privacy with communication efficiency). *Note that our private estimation algorithm for the Bernoulli case is already communication-efficient as each client sends only one bit to the server.*

**Remark 7** (Client sampling). *For simplicity, in the theoretical analysis in Gaussian and Bernoulli models, we assume that all clients participate in the estimation process. However, a simple modification to our analysis also handles the case where only $K$ out of $m$ clients participate: in all our theorem statements we would have to modify to have $K$ instead $m$. Note that we do client sampling for our experiments in Table 1.*

## D PERSONALIZED ESTIMATION – MIXTURE MODEL

Consider a set of $m$ clients, where the $i$-th client has a local dataset $X_i = (X_{i1}, \ldots, X_{in})$ of $n$ samples for $i \in [m]$, where $X_{ij} \in \mathbb{R}^d$. The local samples $X_i$ of the $i$-th client are drawn i.i.d. from a Gaussian distribution $\mathcal{N}(\boldsymbol{\theta}_i, \sigma_x^2 \mathbb{I}_d)$ with unknown mean $\boldsymbol{\theta}_i$ and known variance $\sigma_x^2 \mathbb{I}_d$.

In this section, we assume that the personalized models $\boldsymbol{\theta}_1, \ldots, \boldsymbol{\theta}_m$ are drawn i.i.d. from a discrete distribution $\mathbb{P} = [p_1, \ldots, p_k]$ for given $k$ candidates $\boldsymbol{\mu}_1, \ldots, \boldsymbol{\mu}_k \in \mathbb{R}^d$. In other works, $\Pr[\boldsymbol{\theta}_i = \boldsymbol{\mu}_l] = p_l$ for $l \in [k]$ and $i \in [m]$. The goal of each client is to estimate her personalized model $\{\boldsymbol{\theta}_i\}$ that minimizes the mean square error defined as follows:

$$\text{MSE} = \mathbb{E}_{\{\boldsymbol{\theta}_i, X_i\}}\|\boldsymbol{\theta}_i - \hat{\boldsymbol{\theta}}_i\|^2, \tag{37}$$

where the expectation is taken with respect to the personalized models $\boldsymbol{\theta}_i$ and the local samples $\{X_{ij} \sim \mathcal{N}(\boldsymbol{\theta}_i, \sigma_x^2 \mathbb{I}_d)\}$. Furthermore, $\hat{\boldsymbol{\theta}}_i$ denotes the estimate of the personalized model $\boldsymbol{\theta}_i$ for $i \in [m]$.

First, we start with a simple case when the clients have perfect knowledge of the prior distribution, i.e., the $i$-th client knows the $k$ Gaussian distributions $\mathcal{N}\left(\boldsymbol{\mu}_1, \sigma_{\boldsymbol{\theta}}^2\right), \ldots, \mathcal{N}\left(\boldsymbol{\mu}_k, \sigma_{\boldsymbol{\theta}}^2\right)$ and the prior distribution $\boldsymbol{\alpha} = [\alpha_1, \ldots, \alpha_k]$. This will serve as a stepping stone to handle the more general case when the prior distribution is unknown.

### D.1 WHEN THE PRIOR DISTRIBUTION IS KNOWN

In this case, the $i$-th client does not need the data of the other clients as she has a perfect knowledge about the prior distribution.

**Theorem 6.** *For given a perfect knowledge $\boldsymbol{\alpha} = [\alpha_1, \ldots, \alpha_k]$ and $\mathcal{N}\left(\boldsymbol{\mu}_1, \sigma_{\boldsymbol{\theta}}^2\right), \ldots, \mathcal{N}\left(\boldsymbol{\mu}_k, \sigma_{\boldsymbol{\theta}}^2\right)$, the optimal personalized estimator that minimizes the MSE is given by:*

$$\hat{\boldsymbol{\theta}}_i = \sum_{l=1}^{k} a_l^{(i)} \boldsymbol{\mu}_l, \tag{38}$$

*where $\alpha_l^{(i)} = \dfrac{p_l \exp\left(-\frac{\sum_{j=1}^n \|X_{ij} - \boldsymbol{\mu}_l\|^2}{2\sigma_x^2}\right)}{\sum_{s=1}^k p_s \exp\left(-\frac{\sum_{j=1}^n \|X_{ij} - \boldsymbol{\mu}_s\|^2}{2\sigma_x^2}\right)}$ denotes the weight associated to the prior model $\boldsymbol{\mu}_l$ for $l \in [k]$.*

*Proof.* Let $\boldsymbol{\theta}_i \sim \mathbb{P}$, where $\mathbb{P} = [p_1, \ldots, p_k]$ and $p_l = \Pr[\boldsymbol{\theta}_i = \boldsymbol{\mu}_l]$ for $l \in [k]$. The goal is to design an estimator $\hat{\boldsymbol{\theta}}_i$ that minimizes the MSE given by:

$$\text{MSE} = \mathbb{E}_{\boldsymbol{\theta}_i \sim \mathbb{P}} \mathbb{E}_{\{X_{ij} \sim \mathcal{N}(\boldsymbol{\theta}_i, \sigma_x^2)\}} \left[\|\hat{\boldsymbol{\theta}}_i - \boldsymbol{\theta}_i\|^2\right]. \tag{39}$$

Let $X_i = (X_{i1}, \ldots, X_{in})$. By following the standard proof of the minimum MSE, we get that:

$$\mathbb{E}_{\boldsymbol{\theta}_i} \mathbb{E}_{X_i} \left[\|\hat{\boldsymbol{\theta}}_i - \boldsymbol{\theta}_i\|^2\right] = \mathbb{E}_{X_i} \mathbb{E}_{\boldsymbol{\theta}_i|X_i} \left[\left.\|\hat{\boldsymbol{\theta}}_i - \mathbb{E}[\boldsymbol{\theta}_i|X_i] + \mathbb{E}[\boldsymbol{\theta}_i|X_i] - \boldsymbol{\theta}_i\|^2\right| X_i\right]$$

$$= \mathbb{E}_{X_i} \mathbb{E}_{\boldsymbol{\theta}_i|X_i} \left[\left.\|\mathbb{E}[\boldsymbol{\theta}_i|X_i] - \boldsymbol{\theta}_i\|^2\right| X_i\right] + \mathbb{E}_{X_i} \mathbb{E}_{\boldsymbol{\theta}_i|X_i} \left[\left.\|\mathbb{E}[\boldsymbol{\theta}_i|X_i] - \hat{\boldsymbol{\theta}}_i\|^2\right| X_i\right]$$

$$\geq \mathbb{E}_{X_i} \mathbb{E}_{\boldsymbol{\theta}_i|X_i} \left[\left.\|\mathbb{E}[\boldsymbol{\theta}_i|X_i] - \boldsymbol{\theta}_i\|^2\right| X_i\right], \tag{40}$$

where the last inequality is achieved with equality when $\hat{\boldsymbol{\theta}}_i = \mathbb{E}[\boldsymbol{\theta}_i|X_i]$. The distribution on $\boldsymbol{\theta}_i$ given the local dataset $X_i$ is given by:

$$\Pr[\boldsymbol{\theta}_i = \boldsymbol{\mu}_l|X_i] = \frac{f(X_i|\boldsymbol{\theta}_i = \boldsymbol{\mu}_l) \Pr[\boldsymbol{\theta}_i = \boldsymbol{\mu}_l]}{f(X_i)}$$

$$= \frac{f(X_i|\boldsymbol{\theta}_i = \boldsymbol{\mu}_l) \Pr[\boldsymbol{\theta}_i = \boldsymbol{\mu}_l]}{\sum_{s=1}^k f(X_i|\boldsymbol{\theta}_i = \boldsymbol{\mu}_s) \Pr[\boldsymbol{\theta}_i = \boldsymbol{\mu}_s]} \tag{41}$$

$$= \frac{p_l \exp\left(-\frac{\sum_{j=1}^n \|X_{ij} - \boldsymbol{\mu}_l\|^2}{2\sigma_x^2}\right)}{\sum_{s=1}^k p_s \exp\left(-\frac{\sum_{j=1}^n \|X_{ij} - \boldsymbol{\mu}_s\|^2}{2\sigma_x^2}\right)} = \alpha_l^{(i)}$$

As a result, the optimal estimator is given by:

$$\hat{\boldsymbol{\theta}}_i = \mathbb{E}[\boldsymbol{\theta}_i|X_i] = \sum_{l=1}^{k} \alpha_l^{(i)} \boldsymbol{\mu}_l. \tag{42}$$

This completes the proof of Theorem 6. $\qquad \square$

The optimal personalized estimation in (38) is a weighted summation over all possible candidates vectors $\boldsymbol{\mu}_1, \ldots, \boldsymbol{\mu}_k$, where the weight $\alpha_l^{(i)}$ increases if the prior $p_l$ increases and/or the local samples $\{X_{ij}\}$ are close to the model $\boldsymbol{\mu}_l$ for $l \in [k]$. Observe that the optimal estimator $\hat{\boldsymbol{\theta}}_i$ in Theorem 6 that minimizes the MSE is completely different from the local estimator $\left(\frac{1}{n} \sum_{j=1}^n X_{ij}\right)$. Furthermore, it is easy to see that the local estimator has the MSE $\left(\frac{d\sigma_x^2}{n}\right)$ which increases linearly with the data dimension $d$. On the other hand, the MSE of the optimal estimator in Theorem 6 is a function of the prior distribution $\mathbb{P} = [p_1, \ldots, p_k]$, the prior vectors $\boldsymbol{\mu}_1, \ldots, \boldsymbol{\mu}_k$, and the local variance $\sigma_x^2$.

## D.2 WHEN THE PRIOR DISTRIBUTION IS UNKNOWN

Now, we consider a more practical case when the prior distribution $\mathbb{P} = [p_1, \ldots, p_k]$ and the candidates $\boldsymbol{\mu}_1, \ldots, \boldsymbol{\mu}_k$ are unknown to the clients. In this case, the clients collaborate with each other by their local data to estimate the priors $\mathbb{P}$ and $\boldsymbol{\mu}_1, \ldots, \boldsymbol{\mu}_k$, and then, each client uses the estimated priors to design her personalized model as in (38).

We present Algorithm 3 based on alternating minimization. The algorithm starts by initializing the local models $\{\boldsymbol{\theta}_i^{(0)} := \frac{1}{n} \sum_{j=1}^n X_{ij}\}$. Then, the algorithm works in rounds alternating between estimating the priors $\mathbb{P}^{(t+1)} = [p_1^{(t+1)}, \ldots, p_k^{(t+1)}], \boldsymbol{\mu}_1^{(t+1)}, \ldots, \boldsymbol{\mu}_k^{(t+1)}$ for given local models $\{\boldsymbol{\theta}_i^{(t)}\}$ and estimating the personalized models $\{\boldsymbol{\theta}_i^{(t+1)}\}$ for given global priors $\mathbb{P}^{(t+1)}$ and $\boldsymbol{\mu}_1^{(t+1)}, \ldots, \boldsymbol{\mu}_k^{(t+1)}$. Observe that for given the prior information $\mathbb{P}^{(t)}, \{\boldsymbol{\mu}_l^t\}$, each client updates her personalized model in Step 6 which is the optimal estimator for given priors according to Theorem 6. On the other hand, for given personalized models $\{\boldsymbol{\theta}_i^{(t)}\}$, we estimate the priors $\mathbb{P}^{(t)}, \{\boldsymbol{\mu}_l^t\}$ using clustering algorithm with $k$ sets in Step 11. The algorithm Cluster takes $m$ vectors $\boldsymbol{a}_1, \ldots, \boldsymbol{a}_m$ and an integer $k$ as its input, and its goal is to generate a set of $k$ cluster centers $\boldsymbol{\mu}_1, \ldots, \boldsymbol{\mu}_k$ that minimizes $\sum_{i=1}^m \min_{l \in k} \|\boldsymbol{a}_i - \boldsymbol{\mu}_l\|^2$. Furthermore, these clustering algorithms can also return the prior distribution $\mathbb{P}$, by setting $p_l := \frac{|\mathcal{S}_l|}{m}$, where $\mathcal{S}_l \subset \{\boldsymbol{a}_1, \ldots, \boldsymbol{a}_m\}$ denotes the set of vectors that are belongs to the $l$-th cluster. There are lots of algorithms that do clustering, but perhaps, Lloyd's algorithm Lloyd (1982) and Ahmadian Ahmadian et al. (2019) are the most common algorithms for $k$-means clustering. Our Algorithm 3 can work with any clustering algorithm.

---

**Algorithm 3** Alternating Minimization for Personalized Estimation

---

**Input:** Number of iterations $T$, local datasets $(X_{i1}, \ldots, X_{in})$ for $i \in [m]$.
 1: **Initialize** $\boldsymbol{\theta}_i^0 = \frac{1}{n} \sum_{j=1}^n X_{ij}$ for $i \in [m]$.
 2: **for** $t = 1$ **to** $T$ **do**
 3:  **On Clients:**
 4:  **for** $i = 1$ **to** $m$: **do**
 5:    Receive $\mathbb{P}^{(t)}, \boldsymbol{\mu}_1^{(t)}, \ldots, \boldsymbol{\mu}_k^{(t)}$ from the server
 6:    Update the personalized model:

$$\boldsymbol{\theta}_i^t \leftarrow \sum_{l=1}^k \alpha_l^{(i)} \boldsymbol{\mu}_l^{(t)} \qquad \text{and} \qquad \alpha_l^{(i)} = \frac{p_l^{(t)} \exp\left(-\frac{\sum_{j=1}^n \|X_{ij} - \boldsymbol{\mu}_l^{(t)}\|^2}{2\sigma_x^2}\right)}{\sum_{s=1}^k p_s^{(t)} \exp\left(-\frac{\sum_{j=1}^n \|X_{ij} - \boldsymbol{\mu}_s^{(t)}\|^2}{2\sigma_x^2}\right)}$$

 7:    Send $\boldsymbol{\theta}_i^t$ to the server
 8:  **end for**
 9:  **At the Server:**
10:   Receive $\boldsymbol{\theta}_1^{(t)}, \ldots, \boldsymbol{\theta}_m^{(t)}$ from the clients
11:   Update the global parameters: $\mathbb{P}^{(t)}, \boldsymbol{\mu}_1^{(t)}, \ldots, \boldsymbol{\mu}_k^{(t)} \leftarrow \mathsf{Cluster}\left(\boldsymbol{\theta}_1^{(t)}, \ldots, \boldsymbol{\theta}_m^{(t)}, k\right)$
12:   Broadcast $\mathbb{P}^{(t)}, \boldsymbol{\mu}_1^{(t)}, \ldots, \boldsymbol{\mu}_k^{(t)}$ to all clients
13: **end for**
**Output:** Personalized models $\boldsymbol{\theta}_1^T, \ldots, \boldsymbol{\theta}_m^T$.

---

## D.3 PRIVACY/COMMUNICATION CONSTRAINTS

In the personalized estimation Algorithm 3, each client shares her personalized estimator $\boldsymbol{\theta}_i^{(t)}$ to the server at each iteration which is not communication-efficient and violates the privacy. In this section we present ideas on how to design communication-efficient and/or private Algorithms for personalized estimation.

**Lemma 3.** *Let $\boldsymbol{\mu}_1, \ldots \boldsymbol{\mu}_k \in \mathbb{R}^d$ be unknown means such that $\|\boldsymbol{\mu}_i\|_2 \leq r$ for each $i \in [k]$. Let $\boldsymbol{\theta}_1, \ldots, \boldsymbol{\theta}_m \sim \mathbb{P}$, where $\mathbb{P} = [p_1, \ldots, p_k]$ and $p_l = \Pr[\boldsymbol{\theta}_i = \boldsymbol{\mu}_l]$. For $i \in [m]$, let $X_{i1}, \ldots, X_{in} \sim$*

$\mathcal{N}(\boldsymbol{\theta}_i, \sigma_x^2)$, i.i.d. Then, with probability at least $1 - \frac{1}{mn}$, the following bound holds for all $i \in [m]$:

$$\left\| \frac{1}{n} \sum_{j=1}^{n} X_{ij} \right\|_2 \leq 4\sqrt{d\frac{\sigma_x^2}{n}} + 2\sqrt{\log(m^2 n)\frac{\sigma_x^2}{n}} + r. \tag{43}$$

*Proof.* Observe that the vector $(\overline{X}_i - \theta_i) = \frac{1}{n}\sum_{i=1}^{n} X_{ij} - \theta_i$ is a sub-Gaussian random vector with proxy $\frac{\sigma_x^2}{n}$. As a result, we have that:

$$\|\overline{X}_i - \theta_i\|_2 \leq 4\sqrt{d\frac{\sigma_x^2}{n}} + 2\sqrt{\log(1/\eta)\frac{\sigma_x^2}{n}}, \tag{44}$$

with probability at least $1 - \eta$ from Wainwright (2019). Since $\mu_1, \ldots, \mu_k \in \mathbb{R}^d$ are such that $\|\boldsymbol{\mu}_i\|_2 \leq r$ for each $i \in [k]$, we have:

$$\|\overline{X}_i\|_2 \leq 4\sqrt{d\frac{\sigma_x^2}{n}} + 2\sqrt{\log(1/\eta)\frac{\sigma_x^2}{n}} + r, \tag{45}$$

with probability $1 - \eta$ from the triangular inequality. Thus, by choosing $\eta = \frac{1}{m^2 n}$ and using the union bound, this completes the proof of Lemma 3. $\qquad\square$

Lemma 3 shows that the average of the local samples $\{\overline{X}_i\}$ has a bounded $\ell_2$ norm with high probability. Thus, we can design a communication-efficient estimation Algorithm as follows: Each client clips her personal model $\boldsymbol{\theta}_i^{(t)}$ within radius $4\sqrt{d\frac{\sigma_x^2}{n}} + 2\sqrt{\log(m^2 n)\frac{\sigma_x^2}{n}} + r$. Then, each client applies a vector-quantization scheme (e.g., Bernstein et al. (2018); Alistarh et al. (2017); Girgis et al. (2021a)) to the clipped vector before sending it to the server.

To design a private estimation algorithm with discrete priors, each client clips her personalized estimator $\boldsymbol{\theta}_i^{(t)}$ within radius $4\sqrt{d\frac{\sigma_x^2}{n}} + 2\sqrt{\log(m^2 n)\frac{\sigma_x^2}{n}} + r$. Then, we can use a differentially private algorithm for clustering (see e.g., Stemmer (2020) for clustering under LDP constraints and Ghazi et al. (2020) for clustering under central DP constraints.). Since, we run $T$ iterations in Algorithm 3, we can obtain the final privacy analysis $(\epsilon, \delta)$ using the strong composition theorem Dwork & Roth (2014).

## E  PERSONALIZED LEARNING – LINEAR REGRESSION

In this section, we present the personalized linear regression problem. Consider A set of $m$ clients, where the $i$-th client has a local dataset consisting of $n$ samples $(X_{i1}, Y_{i1}), \ldots, (X_{in}, Y_{in})$, where $X_{ij} \in \mathbb{R}^d$ denotes the feature vector and $Y_{ij} \in \mathbb{R}$ denotes the corresponding response. Let $Y_i = (Y_{i1}, \ldots, Y_{i1}) \in \mathbb{R}^n$ and $X_i = (X_{i1}, \ldots, X_{in}) \in \mathbb{R}^{n \times d}$ denote the response vector and the feature matrix at the $i$-th client, respectively. Following the standard regression, we assume that the response vector $Y_i$ is obtained from a linear model as follows:

$$Y_i = X_i \boldsymbol{\theta}_i + \boldsymbol{w}_i, \tag{46}$$

where $\boldsymbol{\theta}_i$ denotes personalized model of the $i$-th client and $\boldsymbol{w}_i \sim \mathcal{N}\left(0, \sigma_x^2 \mathbb{I}_n\right)$ is a noise vector. The clients' parameters $\boldsymbol{\theta}_1, \ldots, \boldsymbol{\theta}_m$ are drawn i.i.d. from a Gaussian distribution $\boldsymbol{\theta}_1, \ldots, \boldsymbol{\theta}_m \sim \mathcal{N}(\boldsymbol{\mu}, \sigma_\theta^2 \mathbb{I}_d)$, i.i.d.

Our goal is to solve the optimization problem stated in (9) (for the linear regression setup) and learn the optimal personalized parameters $\{\widehat{\boldsymbol{\theta}}_i\}$. The following theorem characterizes the exact form of the optimal $\{\widehat{\boldsymbol{\theta}}_i\}$ and computes their minimum mean squared error w.r.t. the true parameters $\{\boldsymbol{\theta}_i\}$.

**Theorem 7.** *The optimal personalized parameters at client $i$ with known $\boldsymbol{\mu}, \sigma_\theta^2, \sigma_x^2$ is given by:*

$$\widehat{\boldsymbol{\theta}}_i = \left( \frac{\mathbb{I}}{\sigma_\theta^2} + \frac{X_i^T X_i}{\sigma_x^2} \right)^{-1} \left( \frac{X_i^T Y_i}{\sigma_x^2} + \frac{\boldsymbol{\mu}}{\sigma_\theta^2} \right). \tag{47}$$

*The mean squared error (MSE) of the above $\widehat{\boldsymbol{\theta}}_i$ is given by:*

$$\mathbb{E}_{\boldsymbol{w}_i, \boldsymbol{\theta}_i} \left\| \widehat{\boldsymbol{\theta}}_i - \boldsymbol{\theta}_i \right\|^2 = \mathsf{Tr}\left( \left( \frac{\mathbb{I}}{\sigma_\theta^2} + \frac{X_i^T X_i}{\sigma_x^2} \right)^{-1} \right), \tag{48}$$

*Proof.* The personalized model with perfect prior is obtained by solving the optimization problem stated in (9), which is given below for convenience. Note that for linear regression with Gaussian prior, we have $\mathbb{P}(\Gamma) \equiv \mathcal{N}(\boldsymbol{\mu}, \sigma_\theta^2 \mathbb{I}_d)$ and $p_{\boldsymbol{\theta}_i}(Y_{ij}|X_{ij})$ according to $\mathcal{N}(\boldsymbol{\theta}_i, \sigma_x^2)$.

$$\widehat{\boldsymbol{\theta}}_i = \arg\min_{\boldsymbol{\theta}_i} \sum_{j=1}^{n} -\log(p_{\boldsymbol{\theta}_i}(Y_{ij}|X_{ij})) - \log(p(\boldsymbol{\theta}_i)).$$

$$= \arg\min_{\boldsymbol{\theta}_i} \sum_{j=1}^{n} \frac{(Y_{ij} - X_{ij}\boldsymbol{\theta}_i)^2}{2\sigma_x^2} + \frac{\|\boldsymbol{\theta}_i - \boldsymbol{\mu}\|^2}{2\sigma_\theta^2}.$$

$$= \arg\min_{\boldsymbol{\theta}_i} \frac{\|Y_i - X_i\boldsymbol{\theta}_i\|^2}{2\sigma_x^2} + \frac{\|\boldsymbol{\theta}_i - \boldsymbol{\mu}\|^2}{2\sigma_\theta^2}.$$

By taking the derivative with respect to $\boldsymbol{\theta}_i$, we get

$$\frac{\partial}{\partial\boldsymbol{\theta}_i} = \frac{X_i^T(X_i\boldsymbol{\theta}_i - Y_i)}{\sigma_x^2} + \frac{\boldsymbol{\theta}_i - \boldsymbol{\mu}}{\sigma_\theta^2}. \tag{49}$$

Equating the above partial derivative to zero, we get that the optimal personalized parameters $\widehat{\boldsymbol{\theta}}_i$ is given by:

$$\widehat{\boldsymbol{\theta}}_i = \left(\frac{\mathbb{I}}{\sigma_\theta^2} + \frac{X_i^T X_i}{\sigma_x^2}\right)^{-1} \left(\frac{X_i^T Y_i}{\sigma_x^2} + \frac{\boldsymbol{\mu}}{\sigma_\theta^2}\right). \tag{50}$$

Taking the expectation w.r.t. $w_i$, we get:

$$\mathcal{E}_{\boldsymbol{w}_i}[\widehat{\boldsymbol{\theta}}_i] = \left(\frac{\mathbb{I}}{\sigma_\theta^2} + \frac{X_i^T X_i}{\sigma_x^2}\right)^{-1} \left(\frac{X_i^T X_i\boldsymbol{\theta}_i}{\sigma_x^2} + \frac{\boldsymbol{\mu}}{\sigma_\theta^2}\right), \tag{51}$$

Thus, we can bound the MSE as following:

$$\mathcal{E}_{\boldsymbol{w}_i,\boldsymbol{\theta}_i} \left\|\widehat{\boldsymbol{\theta}}_i - \boldsymbol{\theta}_i\right\|^2 = \mathbb{E}_{\boldsymbol{w}_i,\boldsymbol{\theta}_i} \left\|\widehat{\boldsymbol{\theta}}_i - \mathbb{E}_{\boldsymbol{w}_i}[\widehat{\boldsymbol{\theta}}_i] + \mathbb{E}_{\boldsymbol{w}_i}[\widehat{\boldsymbol{\theta}}_i] - \boldsymbol{\theta}_i\right\|^2$$

$$= \mathbb{E}_{\boldsymbol{w}_i,\boldsymbol{\theta}_i} \left\|\widehat{\boldsymbol{\theta}}_i - \mathbb{E}_{\boldsymbol{w}_i}[\widehat{\boldsymbol{\theta}}_i]\right\|^2 + \mathbb{E}_{\boldsymbol{w}_i,\boldsymbol{\theta}_i} \left\|\mathbb{E}_{\boldsymbol{w}_i}[\widehat{\boldsymbol{\theta}}_i] - \boldsymbol{\theta}_i\right\|^2 + 2\mathbb{E}_{\boldsymbol{w}_i,\boldsymbol{\theta}_i} \left\langle\widehat{\boldsymbol{\theta}}_i - \mathbb{E}_{\boldsymbol{w}_i}[\widehat{\boldsymbol{\theta}}_i], \mathbb{E}_{\boldsymbol{w}_i}[\widehat{\boldsymbol{\theta}}_i] - \boldsymbol{\theta}_i\right\rangle$$

$$= \mathbb{E}_{\boldsymbol{w}_i,\boldsymbol{\theta}_i} \left\|\widehat{\boldsymbol{\theta}}_i - \mathbb{E}_{\boldsymbol{w}_i}[\widehat{\boldsymbol{\theta}}_i]\right\|^2 + \mathbb{E}_{\boldsymbol{w}_i,\boldsymbol{\theta}_i} \left\|\mathbb{E}_{\boldsymbol{w}_i}[\widehat{\boldsymbol{\theta}}_i] - \boldsymbol{\theta}_i\right\|^2$$

In the last equality, we used $\mathbb{E}_{\boldsymbol{w}_i,\boldsymbol{\theta}_i} \left\langle\widehat{\boldsymbol{\theta}}_i - \mathbb{E}_{\boldsymbol{w}_i}[\widehat{\boldsymbol{\theta}}_i], \mathbb{E}_{\boldsymbol{w}_i}[\widehat{\boldsymbol{\theta}}_i] - \boldsymbol{\theta}_i\right\rangle =$ $\mathbb{E}_{\boldsymbol{\theta}_i} \left\langle\mathbb{E}_{\boldsymbol{w}_i}[\widehat{\boldsymbol{\theta}}_i] - \mathbb{E}_{\boldsymbol{w}_i}[\widehat{\boldsymbol{\theta}}_i], \mathbb{E}_{\boldsymbol{w}_i}[\widehat{\boldsymbol{\theta}}_i] - \boldsymbol{\theta}_i\right\rangle = 0$, where the first equality holds because $\mathbb{E}_{\boldsymbol{w}_i}[\widehat{\boldsymbol{\theta}}_i] - \boldsymbol{\theta}_i$ is independent of $\boldsymbol{w}_i$.

Letting $\boldsymbol{M} = \frac{\mathbb{I}}{\sigma_\theta^2} + \frac{X_i^T X_i}{\sigma_x^2}$, and Tr denoting the trace operation, we get

$$\mathcal{E}_{\boldsymbol{w}_i,\boldsymbol{\theta}_i} \left\|\widehat{\boldsymbol{\theta}}_i - \boldsymbol{\theta}_i\right\|^2 = \mathsf{Tr}\left(\boldsymbol{M}^{-1}\mathbb{E}_{\boldsymbol{w}_i}\left[\left(\frac{X_i^T \boldsymbol{w}_i}{\sigma_x^2}\right)\left(\frac{X_i^T \boldsymbol{w}_i}{\sigma_x^2}\right)^T\right]\boldsymbol{M}^{-1}\right)$$

$$+ \mathsf{Tr}\left(\boldsymbol{M}^{-1}\mathbb{E}_{\boldsymbol{\theta}_i}\left[\left(\frac{\boldsymbol{\theta}_i - \boldsymbol{\mu}}{\sigma_\theta^2}\right)\left(\frac{\boldsymbol{\theta}_i - \boldsymbol{\mu}}{\sigma_\theta^2}\right)^T\right]\boldsymbol{M}^{-1}\right)$$

$$= \mathsf{Tr}\left(\boldsymbol{M}^{-1}\frac{X_i^T X_i}{\sigma_x^2}\boldsymbol{M}^{-1}\right) + \mathsf{Tr}\left(\boldsymbol{M}^{-1}\frac{\mathbb{I}}{\sigma_\theta^2}\boldsymbol{M}^{-1}\right)$$

$$= \mathsf{Tr}\left(\boldsymbol{M}^{-1}\right).$$

This completes the proof of Theorem 7. $\qquad\square$

Observe that the local model of the $i$-th client, i.e., estimating $\boldsymbol{\theta}_i$ only from the local data $(Y_i, X_i)$, is given by:

$$\widehat{\boldsymbol{\theta}}_i^{(l)} = \left(X_i^T X_i\right)^{-1} X_i^T Y_i, \tag{52}$$

---

**Algorithm 4** Linear Regression GD

---

**Input:** Number of iterations $T$, local datasets $(Y_i, X_i)$ for $i \in [m]$, learning rate $\eta$.

1: **Initialize** $\boldsymbol{\theta}_i^0$ for $i \in [m]$, $\boldsymbol{\mu}^0$, $\sigma_x^{2,0}$, $\sigma_\theta^{2,0}$.
2: **for** $t = 1$ **to** $T$ **do**
3:     **On Clients:**
4:     **for** $i = 1$ **to** $m$: **do**
5:         Receive and set $\boldsymbol{\mu}_i^t = \boldsymbol{\mu}^t, \sigma_{\theta,i}^{2,t} = \sigma_\theta^{2,t}, \sigma_{x,i}^{2,t} = \sigma_x^{2,t}$

6:         Update the personalized model: $\boldsymbol{\theta}_i^t \leftarrow \boldsymbol{\theta}_i^{t-1} + \eta \left( \sum_{j=1}^n \frac{X_{ij}(Y_{ij} - X_{ij}\boldsymbol{\theta}_i^{t-1})}{\sigma_{x,i}^{2,t-1}} + \frac{\boldsymbol{\mu}_i^{t-1} - \boldsymbol{\theta}_i^{t-1}}{\sigma_{\theta,i}^{2,t-1}} \right)$

7:         Update local version of mean: $\boldsymbol{\mu}_i^t \leftarrow \boldsymbol{\mu}_i^{t-1} - \eta \left( \frac{\boldsymbol{\mu}_i^{t-1} - \boldsymbol{\theta}_i^{t-1}}{\sigma_{\theta,i}^{2,t-1}} \right)$

8:         Update local variance: $\sigma_{x,i}^{2,t} \leftarrow \sigma_{x,i}^{2,t-1} - \eta \left( \frac{n}{2\sigma_{x,i}^{2,t-1}} - \sum_{j=1}^n \frac{(Y_{ij} - X_{ij}\boldsymbol{\theta}_i^{t-1})^2}{2(\sigma_{x,i}^{2,t-1})^2} \right)$

9:         Update global variance: $\sigma_{\theta,i}^{2,t} \leftarrow \sigma_{\theta,i}^{2,t-1} - \eta \left( \frac{d}{2\sigma_{\theta,i}^{2,t-1}} - \frac{\|\boldsymbol{\mu}_i^{t-1} - \boldsymbol{\theta}_i^{t-1}\|^2}{2(\sigma_{\theta,i}^{2,t-1})^2} \right)$

10:     **end for**
11:     **At the Server:**
12:     Aggregate mean: $\boldsymbol{\mu}^t = \frac{1}{m} \sum_{i=1}^m \boldsymbol{\mu}_i^t$
13:     Aggregate global variance: $\sigma_\theta^{2,t} = \frac{1}{m} \sum_{i=1}^m \sigma_{\theta,i}^{2,t}$
14:     Aggregate local variance: $\sigma_x^{2,t} = \frac{1}{m} \sum_{i=1}^m \sigma_{x,i}^{2,t}$
15:     Broadcast $\boldsymbol{\mu}^t, \sigma_\theta^{2,t}, \sigma_x^{2,t}$
16: **end for**

**Output:** Personalized models $\boldsymbol{\theta}_1^T, \ldots, \boldsymbol{\theta}_m^T$.

---

where we assume the matrix $X_i^T X_i$ has a full rank (otherwise, we take the pseudo inverse). This local estimate achieves the MSE given by:

$$\mathbb{E} \left\| \widehat{\boldsymbol{\theta}}_i^{(l)} - \boldsymbol{\theta}_i \right\|^2 = \mathsf{Tr}\left( \left( X_i^T X_i \right)^{-1} \right) \sigma_x^2, \tag{53}$$

we can prove it by following similar steps as the proof of Theorem 7. When $\sigma_\theta^2 \to \infty$, we can easily see that the local estimate (52) matches the personalized estimate in (47).

To make the regression problem more practical, we assume that the mean $\boldsymbol{\mu}$, the local variance $\sigma_x^2$, and the global variance $\sigma_\theta^2$ are unknown. Hence, we estimate the personalized parameters by minimizing the negative log likelihood:

$$\widehat{\boldsymbol{\theta}}_1, \ldots, \widehat{\boldsymbol{\theta}}_m = \underset{\{\boldsymbol{\theta}_i\}, \boldsymbol{\mu}, \sigma_x^2, \sigma_\theta^2}{\arg\min} \sum_{i=1}^m \sum_{j=1}^n - \log\left( p_{\boldsymbol{\theta}_i}\left( Y_{ij} | X_{ij} \right) \right) + \sum_{i=1}^m - \log\left( p\left( \boldsymbol{\theta}_i \right) \right)$$

$$= \arg\min \frac{nm}{2} \log(2\pi\sigma_x^2) + \sum_{i=1}^m \sum_{j=1}^n \frac{(Y_{ij} - X_{ij}\theta_i)^2}{2\sigma_x^2} + \frac{md}{2} \log(2\pi\sigma_\theta^2) + \sum_{i=1}^m \frac{\|\boldsymbol{\theta}_i - \boldsymbol{\mu}\|^2}{2\sigma_\theta^2}. \tag{54}$$

Instead of solving the above optimization problem explicitly, we can optimize it through gradient descent (GD) and the resulting algorithm is presented in Algorithm 4. In addition to keeping the personalized models $\{\boldsymbol{\theta}_i^t\}$, each client also maintains local copies of $\{\boldsymbol{\mu}_i^t, \sigma_{\theta,i}^t, \sigma_{x,i}^t\}$ and updates all these parameters by taking appropriate gradients of the objective in (54) and synchronize them with the server to update the global copy of these parameters $\{\boldsymbol{\mu}^t, \sigma_\theta^t, \sigma_x^t\}$.

## F    PERSONALIZED LEARNING – LOGISTIC REGRESSION

As described in Section 3, by taking $\mathbb{P}(\Gamma) \equiv \mathcal{N}(\boldsymbol{\mu}, \sigma_\theta^2 \mathbb{I}_d)$ and $p_{\theta_i}(Y_{ij} | X_{ij}) = \sigma(\langle \boldsymbol{\theta}_i, X_{ij} \rangle)^{Y_{ij}} (1 - \sigma(\langle \boldsymbol{\theta}_i, X_{ij} \rangle))^{(1-Y_{ij})}$, where $\sigma(z) = {}^1\!/_{1+e^{-z}}$ for any $z \in \mathbb{R}$, then the overall optimization problem becomes:

$$\underset{\{\boldsymbol{\theta}_i\}, \boldsymbol{\mu}, \sigma_\theta}{\arg\min} \sum_{i=1}^{m} \sum_{j=1}^{n} \left[ Y_{ij} \log\left( \frac{1}{1 + e^{-\langle \boldsymbol{\theta}_i, X_{ij} \rangle}} \right) + (1 - Y_{ij}) \log\left( \frac{1}{1 + e^{\langle \boldsymbol{\theta}_i, X_{ij} \rangle}} \right) \right]$$

$$+ \frac{md}{2} \log(2\pi\sigma_\theta^2) + \sum_{i=1}^{m} \frac{\|\boldsymbol{\mu} - \boldsymbol{\theta}_i\|_2^2}{2\sigma_\theta^2}. \tag{55}$$

When $\boldsymbol{\mu}$ and $\sigma_\theta^2$ are unknown, we would like to learn them by gradient descent, as in the linear regression case. The corresponding algorithm is described in Algorithm 5.

---

**Algorithm 5** Logistic Regression SGD

---

**Input:** Number of iterations $T$, local datasets $(Y_i, X_i)$ for $i \in [m]$, learning rate $\eta$.

1: **Initialize** $\boldsymbol{\theta}_i^0$ for $i \in [m]$, $\boldsymbol{\mu}^0$, $\sigma_\theta^{2,0}$.
2: **for** $t = 1$ **to** $T$ **do**
3:     **On Clients:**
4:     **for** $i = 1$ **to** $m$: **do**
5:         Receive $(\boldsymbol{\mu}^t, \sigma_\theta^{2,t})$ from the server and set $\boldsymbol{\mu}_i^t := \boldsymbol{\mu}^t, \sigma_{\theta,i}^{2,t} := \sigma_\theta^{2,t}$
6:         Update the personalized model:

$$\boldsymbol{\theta}_i^t \leftarrow \boldsymbol{\theta}_i^{t-1} - \eta \left( \sum_{j=1}^{n} \nabla_{\theta_i^{t-1}} l_{CE}^{(p)}(\boldsymbol{\theta}_i^{t-1}, (X_i^j, Y_i^j)) + \frac{\boldsymbol{\mu}_i^{t-1} - \boldsymbol{\theta}_i^{t-1}}{\sigma_{\theta,i}^{2,t-1}} \right),$$

        where $l_{CE}^{(p)}$ denotes the cross-entropy loss.

7:         Update local version of mean: $\boldsymbol{\mu}_i^t \leftarrow \boldsymbol{\mu}_i^{t-1} - \eta \left( \frac{\boldsymbol{\mu}_i^{t-1} - \boldsymbol{\theta}_i^{t-1}}{\sigma_{\theta,i}^{2,t-1}} \right)$
8:         Update global variance: $\sigma_{\theta,i}^{2,t} \leftarrow \sigma_{\theta,i}^{2,t-1} - \eta \left( \frac{d}{2\sigma_{\theta,i}^{2,t-1}} - \frac{\|\boldsymbol{\mu}_i^{t-1} - \boldsymbol{\theta}_i^{t-1}\|^2}{2(\sigma_{\theta,i}^{2,t-1})^2} \right)$
9:         Send $(\boldsymbol{\mu}_i^t, \sigma_{\theta,i}^{2,t})$ to the server
10:    **end for**
11:    **At the Server:**
12:     Receive $\{(\boldsymbol{\mu}_i^t, \sigma_{\theta,i}^{2,t})\}$ from the clients
13:     Aggregate mean: $\boldsymbol{\mu}^t = \frac{1}{m} \sum_{i=1}^{m} \boldsymbol{\mu}_i^t$
14:     Aggregate global variance: $\sigma_\theta^{2,t} = \frac{1}{m} \sum_{i=1}^{m} \sigma_{\theta,i}^{2,t}$
15:     Broadcast $(\boldsymbol{\mu}^t, \sigma_\theta^{2,t})$ to all clients
16: **end for**
**Output:** Personalized models $\boldsymbol{\theta}_1^T, \ldots, \boldsymbol{\theta}_m^T$.

---

## G  PERSONALIZED LEARNING – MIXTURE MODEL

In this section, we present the linear regression problem as a generalization to the estimation problem with discrete priors. This model falls into the framework studied in Marfoq et al. (2021) and is illustrated to show how our framework also captures it.

Consider a set of $m$ clients, where the $i$-th client has a local dataset $(X_{i1}, Y_{i1}), \ldots, (X_{in}, Y_{in})$ of $m$ samples, where $X_{ij} \in \mathbb{R}^d$ denotes the feature vector and $Y_{ij} \in \mathbb{R}$ denotes the corresponding response. Let $Y_i = (Y_{i1}, \ldots, Y_{i1}) \in \mathbb{R}^n$ and $X_i = (X_{i1}, \ldots, X_{in}) \in \mathbb{R}^{n \times d}$ denote the response vector and the feature matrix at the $i$-th client, respectively. Following the standard regression, we assume that the response vector $Y_i$ is obtained from a linear model as follows:

$$Y_i = X_i \boldsymbol{\theta}_i + \boldsymbol{w}_i, \tag{56}$$

where $\boldsymbol{\theta}_i$ denotes personalized model of the $i$-th client and $\boldsymbol{w}_i \sim \mathcal{N}\left(0, \sigma_x^2 \mathbb{I}_n\right)$ is a noise vector. The clients models are drawn i.i.d. from a discrete distribution $\boldsymbol{\theta}_1, \ldots, \boldsymbol{\theta}_m \sim \mathbb{P}$, where $\mathbb{P} = [p_1, \ldots, p_k]$ such that $p_l = \Pr[\boldsymbol{\theta}_i = \boldsymbol{\mu}_l]$ for $i \in [m]$ and $l \in [k]$.

Our goal is to solve the optimization problem stated in (9) (for the linear regression with the above discrete prior) and learn the optimal personalized parameters $\{\widehat{\boldsymbol{\theta}}_i\}$.

We assume that the discrete distribution $\mathbb{P}$ and the prior candidates $\{\boldsymbol{\mu}_l\}_{l=1}^k$ are unknown to the clients. Inspired from Algorithm 3 for estimation with discrete priors, we obtain Algorithm 6 for learning with discrete prior. Note that this is *not* a new algorithm, and is essentially the algorithm proposed in Marfoq et al. (2021) applied to linear regression. We wanted to show how our framework captures mixture model in Marfoq et al. (2021) through this example.

**Description of Algorithm 6.** Client $i$ initializes its personalized parameters $\boldsymbol{\theta}_i^{(0)} = (X_i^T X_i)^{-1} X_i^T Y_i$, which is the optimal as a function of the local dataset at the $i$-th client without any prior knowledge. In any iteration $t$, for a given prior information $\mathbb{P}^{(t)}, \{\boldsymbol{\mu}_l^{(t)}\}$, the $i$-th client updates the personalized model as $\boldsymbol{\theta}_i^t = \sum_{l=1}^k \alpha_l^{(i)} \boldsymbol{\mu}_l^{(t)}$, where the weights $\alpha_l^{(i)} \propto p_l^{(t)} \exp\left(-\frac{\|X_i \boldsymbol{\mu}_l^{(t)} - Y_i\|^2}{2\sigma_x^2}\right)$ and sends its current estimate of the personalized parameter $\boldsymbol{\theta}_i^t$ to the server. Upon receiving $\boldsymbol{\theta}_1^t, \dots, \boldsymbol{\theta}_m^t$, server will run Cluster algorithm to update the global parameters $\mathbb{P}, \boldsymbol{\mu}_1^{(t)}, \dots, \boldsymbol{\mu}_k^{(t)}$, and broadcasts them to the clients.

---

**Algorithm 6** Alternating Minimization for Personalized Learning

---

**Input:** Number of iterations $T$, local datasets $(X_i, Y_i)$ for $i \in [m]$.

1: **Initialize** $\boldsymbol{\theta}_i^0 = (X_i^T X_i)^{-1} X_i^T Y_i$ for $i \in [m]$ (if $X_i^T X_i$ is not full-rank, take the pseudo-inverse).

2: **for** $t = 1$ **to** $T$ **do**
3:    **On Clients:**
4:    **for** $i = 1$ **to** $m$: **do**
5:       Receive $\mathbb{P}^{(t)}, \boldsymbol{\mu}_1^{(t)}, \dots, \boldsymbol{\mu}_k^{(t)}$ from the server
6:       Update the personalized parameters and the coefficients:

$$\boldsymbol{\theta}_i^t \leftarrow \sum_{l=1}^k \alpha_l^{(i)} \boldsymbol{\mu}_l^{(t)} \qquad \text{and} \qquad \alpha_l^{(i)} = \frac{p_l^{(t)} \exp\left(-\frac{\|X_i \boldsymbol{\mu}_l^{(t)} - Y_i\|^2}{2\sigma_x^2}\right)}{\sum_{s=1}^k p_s^{(t)} \exp\left(-\frac{\|X_i \boldsymbol{\mu}_s^{(t)} - Y_i\|^2}{2\sigma_x^2}\right)}$$

7:       Send $\boldsymbol{\theta}_i^{(t)}$ to the server
8:    **end for**
9:    **At the Server:**
10:   Receive $\boldsymbol{\theta}_1^{(t)}, \dots, \boldsymbol{\theta}_m^{(t)}$ from the clients
11:   Update the global parameters: $\mathbb{P}^{(t)}, \boldsymbol{\mu}_1^{(t)}, \dots, \boldsymbol{\mu}_k^{(t)} \leftarrow \mathsf{Cluster}\left(\boldsymbol{\theta}_1^{(t)}, \dots, \boldsymbol{\theta}_m^{(t)}, k\right)$
12:   Broadcast $\mathbb{P}^{(t)}, \boldsymbol{\mu}_1^{(t)}, \dots, \boldsymbol{\mu}_k^{(t)}$ to all clients
13: **end for**
**Output:** Personalized models $\boldsymbol{\theta}_1^T, \dots, \boldsymbol{\theta}_m^T$.

---

# H   PERSONALIZED LEARNING – ADAPED

## H.1   KNOWLEDGE DISTILLATION POPULATION DISTRIBUTION

In this section we discuss what type of a population distribution can give rise to algorithms/problems that include a knowledge distillation (KD) (or KL divergence) penalty term between local and global models. From Section 3, Equation (9), consider $p_{\boldsymbol{\theta}_i}(y|x)$ as a randomized mapping from input space $\mathcal{X}$ to output class $\mathcal{Y}$, parameterized by $\boldsymbol{\theta}_i$. For simplicity, consider the case where $|\mathcal{X}|$ is finite, e.g. for MNIST it could be all possible $28 \times 28$ black and white images. Every $p_{\boldsymbol{\theta}_i}(y|x)$ corresponds to a probability matrix (parameterized by $\boldsymbol{\theta}_i$) of size $|\mathcal{Y}| \times |\mathcal{X}|$, where the $(y, x)$'th represents the probability of the class $y$ (row) given the data sample $x$ (column). Therefore, each column is a probability vector. Since we want to sample the probability matrix, it suffices to restrict our attention to any set of $|\mathcal{Y}| - 1$ rows, as the remaining row can be determined by these $|\mathcal{Y}| - 1$ rows.

Similarly, for a global parameter $\boldsymbol{\mu}$, let $p_{\boldsymbol{\mu}}(y|x)$ define a randomized mapping from $\mathcal{X}$ to $\mathcal{Y}$, parameterized by the global parameter $\boldsymbol{\mu}$. Note that for a fixed global parameter $\boldsymbol{\mu}$, the randomized map $p_{\boldsymbol{\mu}}(y|x)$ is fixed, whereas, our goal is to sample $p_{\boldsymbol{\theta}_i}(y|x)$ for $i = 1, \ldots, m$, one for each client. For simplicity of notation, define $p_{\boldsymbol{\theta}_i} := p_{\boldsymbol{\theta}_i}(y|x)$ and $p_{\boldsymbol{\mu}} := p_{\boldsymbol{\mu}}(y|x)$ to be the corresponding probability matrices, and let the distribution for sampling $p_{\boldsymbol{\theta}_i}(y|x)$ be denoted by $p_{p_{\boldsymbol{\mu}}}(p_{\boldsymbol{\theta}_i})$. Note that different mappings $p_{\boldsymbol{\theta}_i}(y|x)$ correspond to different $\boldsymbol{\theta}_i$'s, so we define $p(\boldsymbol{\theta}_i)$ (in Equation (9)) as $p_{p_{\boldsymbol{\mu}}}(p_{\boldsymbol{\theta}_i})$, which is the density of sampling the probability matrix $p_{\boldsymbol{\theta}_i}(y|x)$.

For the KD population distribution, we define this density $p_{p_{\boldsymbol{\mu}}}(p_{\boldsymbol{\theta}_i})$ as:

$$p_{p_{\boldsymbol{\mu}}}(p_{\boldsymbol{\theta}_i}) = c(\psi) e^{-\psi D_{\mathsf{KL}}(p_{\boldsymbol{\mu}}(y|x) \| p_{\boldsymbol{\theta}_i}(y|x))} \tag{57}$$

where $\psi$ is an 'inverse variance' type of parameter, $c(\psi)$ is a normalizing function that depends on $(\psi, p_{\boldsymbol{\mu}})$, and $D_{\mathsf{KL}}(p_{\boldsymbol{\mu}}(y|x) \| p_{\boldsymbol{\theta}_i}(y|x)) = \sum_{x \in \mathcal{X}} p(x) \sum_{y \in \mathcal{Y}} p_{\boldsymbol{\mu}}(y|x) \log \left( \frac{p_{\boldsymbol{\mu}}(y|x)}{p_{\boldsymbol{\theta}_i}(y|x)} \right)$ is the conditional KL divergence, where $p(x)$ denotes the probability of sampling a data sample $x \in \mathcal{X}$. Now all we need is to find $c(\psi)$ given a fixed $\boldsymbol{\mu}$ (and therefore fixed $p_{\boldsymbol{\mu}}(y|x)$). Here we consider $D_{\mathsf{KL}}(p_{\boldsymbol{\mu}} \| p_{\boldsymbol{\theta}_i})$, but our analysis can be extended to $D_{\mathsf{KL}}(p_{\boldsymbol{\theta}_i} \| p_{\boldsymbol{\mu}})$ or $\| p_{\boldsymbol{\theta}_i} - p_{\boldsymbol{\mu}} \|_2$ as well.

For simplicity and to make the calculations easier, we consider a binary classification task with $\mathcal{Y} = \{0, 1\}$ and define $p_{\boldsymbol{\mu}}(x) := p_{\boldsymbol{\mu}}(y = 1 | X = x)$ and $q_i(x) := p_{\boldsymbol{\theta}_i}(y = 1 | X = x)$. We have:

$$\begin{aligned} D_{\mathsf{KL}}(p_{\boldsymbol{\mu}}(y|x) \| p_{\boldsymbol{\theta}_i}(y|x)) = \sum_x p(x) \Big( & p_{\boldsymbol{\mu}}(x)(\log p_{\boldsymbol{\mu}}(x) - \log q_i(x)) \\ & + (1 - p_{\boldsymbol{\mu}}(x))(\log(1 - p_{\boldsymbol{\mu}}(x)) - \log(1 - q_i(x))) \Big). \end{aligned}$$

Hence, after some algebra we have,

$$p_{p_{\boldsymbol{\mu}}}(p_{\boldsymbol{\theta}_i}) = c(\psi) e^{\psi \sum_x p(x) H(p_{\boldsymbol{\mu}}(x))} e^{\psi \sum_x p(x)(p_{\boldsymbol{\mu}}(x) \log(q_i(x)) + (1 - p_{\boldsymbol{\mu}}(x)) \log(1 - q_i(x)))}$$

Then,

$$c(\psi) \prod_x \left[ \int_0^1 e^{\psi p(x) H(p_{\boldsymbol{\mu}}(x))} e^{\psi p(x)(p_{\boldsymbol{\mu}}(x) \log(q_i(x)) + (1 - p_{\boldsymbol{\mu}}(x)) \log(1 - q_i(x)))} dq_i(x) \right] = 1.$$

Note that

$$\int_0^1 e^{\psi p(x)(p_{\boldsymbol{\mu}}(x) \log(q_i(x)) + (1 - p_{\boldsymbol{\mu}}(x)) \log(1 - q_i(x)))} dq_i(x) = B \left( 1 + \frac{p_{\boldsymbol{\mu}}(x)}{\psi p(x)}, 1 + \frac{1 - p_{\boldsymbol{\mu}}(x)}{\psi p(x)} \right)$$

Accordingly, after some algebra, we can obtain $c(\psi) = \frac{e^{-\psi \sum_x p(x) H(p_{\boldsymbol{\mu}}(x))}}{\prod_x B \left( 1 + \frac{p_{\boldsymbol{\mu}}(x)}{\psi p(x)}, 1 + \frac{1 - p_{\boldsymbol{\mu}}(x)}{\psi p(x)} \right)}$, where $H$ is binary Shannon entropy. Substituting this in (57), we get

$$p_{p_{\boldsymbol{\mu}}}(p_{\boldsymbol{\theta}_i}) = \frac{e^{-\psi \sum_x p(x) H(p_{\boldsymbol{\mu}}(x))}}{\prod_x B(1 + \frac{p_{\boldsymbol{\mu}}(x)}{\psi p(x)}, 1 + \frac{1 - p_{\boldsymbol{\mu}}(x)}{\psi p(x)})} e^{-\psi D_{\mathsf{KL}}(p_{\boldsymbol{\mu}}(y|x) \| p_{\boldsymbol{\theta}_i}(y|x))}$$

which is the population distribution that can result in a KD type regularizer. Note that when we take the negative logarithm of the population distribution we obtain KL divergence loss and an additional term that depends on $\psi$ and $p_{\boldsymbol{\mu}}$. This is the form seen in Section 3.2 Equation (11) for AdaPeD algorithm. For numerical purpose, we take the additional term $-\log \left( \frac{e^{-\psi \sum_x p(x) H(p_{\boldsymbol{\mu}}(x))}}{\prod_x B(1 + \frac{p_{\boldsymbol{\mu}}(x)}{\psi p(x)}, 1 + \frac{1 - p_{\boldsymbol{\mu}}(x)}{\psi p(x)})} \right)$ to be simple $\frac{1}{2} \log(2\psi)$. As mentioned in Section 3.2, this serves the purpose of regularizing $\psi$. This is in contrast to the objective considered in Ozkara et al. (2021), which only has the KL divergence loss as the regularizer, without the additional term.

## H.2 AdaPeD with Unsampled Client Iterations

When there is a flexibility in computational resources for doing local iterations, unsampled clients can do local training on their personalized models to speed-up convergence at no cost to privacy. This can be used in cross-silo settings, such as cross-institutional training for hospitals, where privacy is

crucial and there are available computing resources most of the time. We propose the algorithm for AdaPeD with with unsampled client iterations in Algorithm 7:

---

**Algorithm 7** Adaptive Personalization via Distillation (`AdaPeD`) with unsampled client iterations

---

**Parameters:** local variances $\{\psi_i^0\}$, personalized models $\{\boldsymbol{\theta}_i^0\}$, local copies of the global model $\{\boldsymbol{\mu}_i^0\}$, synchronization gap $\tau$, learning rates $\eta_1, \eta_2, \eta_3$, number of sampled clients $K$.

1: **for** $t = 0$ **to** $T - 1$ **do**
2:     **if** $\tau$ divides $t$ **then**
3:         **On Server do:**
4:         Choose a subset $\mathcal{K}^t \subseteq [n]$ of $K$ clients
5:         Broadcast $\boldsymbol{\mu}^t$ and $\psi^t$
6:         **On Clients** $i \in \mathcal{K}^t$ (in parallel) **do**:
7:         Receive $\boldsymbol{\mu}^t$ and $\psi^t$; set $\boldsymbol{\mu}_i^t = \boldsymbol{\mu}^t$, $\psi_i^t = \psi^t$
8:     **end if**
9:     **On Clients** $i \notin \mathcal{K}^t$ (in parallel) **do**:
10:       Compute $\boldsymbol{g}_i^t := \nabla_{\boldsymbol{\theta}_i^t} f_i(\boldsymbol{\theta}_i^t) + \frac{\nabla_{\boldsymbol{\theta}_i^t} f_i^{\mathsf{KD}}(\boldsymbol{\theta}_i^t, \boldsymbol{\mu}_i^{t_i'})}{2\psi_i^{t_i'}}$ where $t_i'$ is the last time index where client $i$

          received global parameters from the server
11:       Update: $\boldsymbol{\theta}_i^{t+1} = \boldsymbol{\theta}_i^t - \eta_1 \boldsymbol{g}_i^t$
12:     **On Clients** $i \in \mathcal{K}^t$ (in parallel) **do**:
13:       Compute $\boldsymbol{g}_i^t := \nabla_{\boldsymbol{\theta}_i^t} f_i(\boldsymbol{\theta}_i^t) + \frac{\nabla_{\boldsymbol{\theta}_i^t} f_i^{\mathsf{KD}}(\boldsymbol{\theta}_i^t, \boldsymbol{\mu}_i^t)}{2\psi_i^t}$
14:       Update: $\boldsymbol{\theta}_i^{t+1} = \boldsymbol{\theta}_i^t - \eta_1 \boldsymbol{g}_i^t$
15:       Compute $\boldsymbol{h}_i^t := \frac{\nabla_{\boldsymbol{\mu}_i^t} f_i^{\mathsf{KD}}(\boldsymbol{\theta}_i^{t+1}, \boldsymbol{\mu}_i^t)}{2\psi_i^t}$
16:       Update: $\boldsymbol{\mu}_i^{t+1} = \boldsymbol{\mu}_i^t - \eta_2 \boldsymbol{h}_i^t$
17:       Compute $k_i^t := \frac{1}{2\psi_i^t} - \frac{f_i^{\mathsf{KD}}(\boldsymbol{\theta}_i^{t+1}, \boldsymbol{\mu}_i^{t+1})}{2(\psi_i^t)^2}$
18:       Update: $\psi_i^{t+1} = \psi_i^t - \eta_3 k_i^t$
19:     **if** $\tau$ divides $t + 1$ **then**
20:       Clients send $\boldsymbol{\mu}_i^t$ and $\psi_i^t$ to **Server**
21:       Server receives $\{\boldsymbol{\mu}_i^t\}_{i \in \mathcal{K}^t}$ and $\{\psi_i^t\}_{i \in \mathcal{K}^t}$
22:       Server computes $\boldsymbol{\mu}^{t+1} = \frac{1}{K} \sum_{i \in \mathcal{K}^t} \boldsymbol{\mu}_i^t$ and $\psi^{t+1} = \frac{1}{K} \sum_{i \in \mathcal{K}^t} \psi_i^t$
23:     **end if**
24: **end for**
**Output:** Personalized models $(\boldsymbol{\theta}_i^T)_{i=1}^m$

---

Of course, when a client is not sampled for a long period of rounds this approach can become similar to a local training; hence, it might be reasonable to put an upper limit on the successive number of local iterations for each client.

## I    PERSONALIZED LEARNING – DP-ADAPED

**Proof of Theorem 5**

**Theorem** (Restating Theorem 5). *After $T$ iterations, `DP-AdaPeD` satisfies $(\alpha, \epsilon(\alpha))$-RDP for $\alpha > 1$, where $\epsilon(\alpha) = \left(\frac{K}{m}\right)^2 6 \left(\frac{T}{\tau}\right) \alpha \left(\frac{C_1^2}{K\sigma_{q_1}^2} + \frac{C_2^2}{K\sigma_{q_2}^2}\right)$, where $\frac{K}{m}$ denotes the sampling ratio of the clients at each global iteration.*

*Proof.* In this section, we provide the privacy analysis of `DP-AdaPeD`. We first analyze the RDP of a single global round $t \in [T]$ and then, we obtain the results from the composition of the RDP over total $T$ global rounds. Recall that privacy leakage can happen through communicating $\{\boldsymbol{\mu}_i\}$ and $\{\psi_i^t\}$ and we privatize both of these. In the following, we do the privacy analysis of privatizing $\{\boldsymbol{\mu}_i\}$ and a similar analysis could be done for $\{\psi_i^t\}$ as well.

At each synchronization round $t \in [T]$, the server updates the global model $\boldsymbol{\mu}^{t+1}$ as follows:

$$\boldsymbol{\mu}^{t+1} = \frac{1}{K} \sum_{i \in \mathcal{K}t} \boldsymbol{\mu}_i^t, \tag{58}$$

where $\boldsymbol{\mu}_i^t$ is the update of the global model at the $i$-th client that is obtained by running $\tau$ local iterations at the $i$-th client. At each of the local iterations, the client clips the gradient $\boldsymbol{h}_i^t$ with threshold $C_1$ and adds a zero-mean Gaussian noise vector with variance $\sigma_{q_1}^2 \mathbb{I}_d$. When neglecting the noise added at the local iterations, the norm-2 sensitivity of updating the global model $\boldsymbol{\mu}_i^{t+1}$ at the synchronization round $t$ is bounded by:

$$\Delta \boldsymbol{\mu} = \max_{\mathcal{K}^t, \mathcal{K}'^t} \|\boldsymbol{\mu}^{t+1} - \boldsymbol{\mu}'^{t+1}\|_2^2 \leq \frac{\tau C_1^2}{K^2}, \tag{59}$$

where $\mathcal{K}^t, \mathcal{K}'^t \subset [m]$ are neighboring sets that differ in only one client. Additionally, $\boldsymbol{\mu}^{t+1} = \frac{1}{K} \sum_{i \in \mathcal{K}t} \boldsymbol{\mu}_i^t$ and $\boldsymbol{\mu}'^{t+1} = \frac{1}{K} \sum_{i \in \mathcal{K}'t} \boldsymbol{\mu}_i^t$. Since we add i.i.d. Gaussian noises with variance $\sigma_{q_1}^2$ at each local iteration at each client, and then, we take the average of theses vectors over $K$ clients, it is equivalent to adding a single Gaussian vector to the aggregated vectors with variance $\frac{\tau \sigma_{q_1}^2}{K}$. Thus, from the RDP of the sub-sampled Gaussian mechanism in (Mironov et al., 2019, Table 1), Bun et al. (2018), we get that the global model $\boldsymbol{\mu}^{t+1}$ of a single global iteration of DP-AdaPeD is $(\alpha, \epsilon_t^{(1)}(\alpha))$-RDP, where $\epsilon_t(\alpha)$ is bounded by:

$$\epsilon_t^{(1)}(\alpha) = \left(\frac{K}{m}\right)^2 \frac{6\alpha C_1^2}{K\sigma_{q_1}^2}. \tag{60}$$

Similarly, we can show that the global parameter $\psi^{t+1}$ at any synchronization round of DP-AdaPeD is $(\alpha, \epsilon_t^{(2)}(\alpha))$-RDP, where $\epsilon_t(\alpha)$ is bounded by:

$$\epsilon_t^{(2)}(\alpha) = \left(\frac{K}{m}\right)^2 \frac{6\alpha C_2^2}{K\sigma_{q_2}^2}. \tag{61}$$

Using adaptive RDP composition (Mironov, 2017, Proposition 1), we get that each synchronization round of DP-AdaPeD is $(\alpha, \epsilon_t^{(1)}(\alpha) + \epsilon_t^{(2)}(\alpha))$-RDP. Thus, by running DP-AdaPeD over $T/\tau$ synchronization rounds and from the composition of the RDP, we get that DP-AdaPeD is $(\alpha, \epsilon(\alpha))$-RDP, where $\epsilon(\alpha) = \left(\frac{T}{\tau}\right)\left(\epsilon_t^{(1)}(\alpha) + \epsilon_t^{(2)}(\alpha)\right)$. This completes the proof of Theorem 5. □

## J    EXPANDED RELATED WORK AND CONNECTIONS TO EXISTING METHODS

In Section 1, we mentioned that the our framework has connections to several personalized FL methods. In this appendix we provide a few more details related to these connections.

**Regularization:** As noted earlier using (9) with the Gaussian population prior connects to the use of $\ell_2$ regularizer in earlier personalized learning works Dinh et al. (2020); Ozkara et al. (2021); Hanzely & Richtárik (2020); Hanzely et al. (2020); Li et al. (2021), which also iterates between local and global model estimates. This form can be explicitly seen in Appendix E, where in Algorithm 4, we see that the Gaussian prior along with iterative optimization yields the regularized form seen in these methods. In these cases[8], $\mathbb{P}(\Gamma) \equiv \mathcal{N}(\boldsymbol{\mu}, \sigma_\theta^2 \mathbb{I}_d)$ for unknown parameters $\Gamma = \{\boldsymbol{\mu}\}$. Note that since the parameters of the population distribution are unknown, these need to be estimated during the iterative learning process. In the algorithm, 4 it is seen the $\boldsymbol{\mu}$ plays the role of the global model (and is truly so for the linear regression problem studied in Appendix E).

**Clustered FL:** If one uses a *discrete* mixture model for the population distribution then the iterative algorithm suggested by our framework connects to (Zhang et al., 2021; Mansour et al., 2020; Ghosh et al., 2020; Smith et al., 2017; Marfoq et al., 2021). In particular, consider a population model with parameters in the $m$-dimensional probability simplex $\{\boldsymbol{\alpha} : \boldsymbol{\alpha} = [\alpha_1, \ldots, \alpha_k], \alpha_i \geq 0, \forall i, \sum_i \alpha_i = 1\}$ which describing a distribution. If there are $m$ (unknown) discrete distributions $\{\mathcal{D}_1, \ldots, \mathcal{D}_m\}$, one can consider these as the unknown description of the population model in addition to $\boldsymbol{\alpha}$. Therefore,

---

[8]One can generalize these by including $\sigma_\theta^2$ in the unknown parameters.

each local data are generated either as a mixture as in (Marfoq et al., 2021) or by choosing one of the unknown discrete distributions with probability $\boldsymbol{\alpha}$ dictating the probability of choosing $\mathcal{D}_i$, when hard clustering is used (*e.g.,* (Mansour et al., 2020)). Each node $j$ chooses a mixture probability $\boldsymbol{\alpha}^{(j)}$ uniformly from the $m$-dimensional probability simplex. In the former case, it uses this mixture probability to generate a local mixture distribution. In the latter it chooses $\mathcal{D}_i$ with probability $\boldsymbol{\alpha}_i^{(j)}$.

As mentioned earlier, not all parametrized distributions can be written as a mixture of *finite* number distributions, which is the assumption for discrete mixtures. Consider a unimodal Gaussian population distribution (as also studied in Appendix E). Since $\mathbb{P}(\Gamma) \equiv \mathcal{N}(\boldsymbol{\mu}, \sigma_\theta^2 \mathbb{I}_d)$, for node $i$, we sample $\boldsymbol{\theta}_i \sim \mathbb{P}(\Gamma)$. We see that the actual data distribution for this node is $p_{\boldsymbol{\theta}_i}(y|\boldsymbol{x}) = \mathcal{N}(\boldsymbol{\theta_i}^\top \boldsymbol{x}, \sigma^2)$. Clearly the set of such distributions $\{p_{\boldsymbol{\theta}_i}(y|\boldsymbol{x})\}$ *cannot* be written as any finite mixture as $\boldsymbol{\theta}_i \in \mathbb{R}^d$ and $p_{\boldsymbol{\theta}_i}(y|\boldsymbol{x})$ is a unimodal Gaussian distribution, with same parameter $\boldsymbol{\theta_i}$ for all data generated in node $i$. Essentially the generative framework of finite mixtures (as in (Marfoq et al., 2021)) could be restrictive as it does not capture such parametric models.

**Knowledge distillation:** The population distribution related to a regularizer based on Kullback-Leibler divergence (knowledge distillation) has been shown in Appendix H. Therefore this can be cast in terms of information geometry where the probability falls of exponentially with in this geometry. Hence these connect to methods such as Lin et al. (2020); Li & Wang (2019); Shen et al. (2020); Ozkara et al. (2021), but the exact regularizer used does not take into account the full parametrization, and one can therefore improve upon these methods.

**FL with Multi-task Learning (MTL):** In this framework, a *fixed* relationship between tasks is usually assumed (Smith et al., 2017). Therefore one can model this as a Gaussian model with *known* parameters relating the individual models. The individual models are chosen from a joint Gaussian with particular (known) covariance dictating the different models, and therefore giving the quadratic regularization used in FL-MTL (Smith et al., 2017). In this the parameters of the Gaussian model are known and fixed.

**Common representations:** The works in Du et al. (2021); Jain et al. (2021b) use a linear model where $y \sim \mathcal{N}(\boldsymbol{x}^\top \boldsymbol{\theta}_i, \sigma^2)$ can be considered a local generative model for node $i$. The common representation approach assumes that $\boldsymbol{\theta}_i = \sum_{j=1}^k \boldsymbol{B}\boldsymbol{w}_j^{(i)}$, for some $k \ll d$, where $\boldsymbol{\theta}_i \in \mathbb{R}^d$. Therefore, one can parametrize a population by this (unknown) common basis $\boldsymbol{B}$, and under a mild assumption that the weights are bounded, we can choose a uniform measure in this bounded cube to choose $\boldsymbol{w}^{(i)}$ for each node $i$. The alternating optimization iteratively discovers the global common representation and the local weights as done in Du et al. (2021); Jain et al. (2021b) (and references therein). This common linear representation approach was generalized in Du et al. (2021); Collins et al. (2021) to neural networks, where a set of parameters to obtain a common representation ("head") at each client was obtained and each local client appendd it with a "tail" combining the representation to obtain the final model. This also fits into our statistical framework, where the common representation (head) parameters are chosen from a population model (like the common subspace in the linear case) and the tail parameters are independently chosen (again as in the linear case).

**Empirical and Hierarchical Bayes:** As mentioned in Section 1, our work is inspired by empirical Bayes framework, introduced in (Stein, 1956; Robbins, 1956; James & Stein, 1961), which is the origin of hierarchical Bayes methods; see also (Gelman et al., 2013, pp. 132). (Stein, 1956; James & Stein, 1961) studied jointly estimating Gaussian individual parameters, generated by an unknown (parametrized) Gaussian population distribution. They showed a surprising result that one can enhance the estimate of individual parameters based on the observations of a population of Gaussian random variables with *independently* generated parameters from an unknown (parametrized) Gaussian population distribution. Effectively, this methodology advocated *estimating* the unknown population distribution using the individual independent samples, and then using it effectively as an empirical prior for individual estimates.[9] This was studied for Bernoulli variables with heterogeneously generated individual parameters by Lord (1967) and the optimal error bounds for maximum likelihood estimates for population distributions were recently developed in (Vinayak et al., 2019). Hierarchical Bayes, builds on empirical Bayes framework and is sometimes associated with a fully Bayes method. Our choice to use empirical Bayes framework as the foundation is also because

---

[9]This was shown to uniformly improve the mean-squared error averaged over the population, compared to an estimate using just the single local sample.

it is more computationally feasible than a fully Bayes method. The subtle difference between the two is that empirical Bayes uses a point estimate of a (parametrized) prior, whereas, the terminology hierarchical Bayes often refers to a fully Bayes method where the (non-parametric) prior is estimated by computationally intensive methods like MCMC (see the discussion in (Blei et al., 2003)). As mentioned in Section 1, a contribution of our work is to connect a well studied statistical framework of empirical (hierarchical) Bayes to model heterogeneity in personalized federated learning. This statistical model yields a framework for personalized FL and leads to new algorithms and bounds especially in the local data starved regime.

# K    ADDITIONAL DETAILS AND RESULTS FOR EXPERIMENTS

## K.1    IMPLEMENTATION DETAILS

In this section we give further details on implementation and setting of the experiments that were used in Section 4.

**CIFAR-100 Experiment Setting.** We do additional experiments on CIFAR-100. CIFAR-100 is a dataset consisting of 100 classes and 20 superclasses. Each superclass corresponds to a category of 5 classes (e.g. superclass flowers correspond to orchids, poppies, roses, sunflowers, tulips). To introduce heterogeneity we let each client sample data samples from 2 super classes (the classification task is still to classify among 100 classes). For classification on CIFAR-100 dataset we consider a 5-layer CNN with 2 convolutional layers of 64 filters and 5x5 filter size, following that we have 2 fully connected layers with activation sizes of 384,192 and finally an output layer of dimension 100. We set number of local epochs to be 2, batch size to be 25 per client; number of clients is 50, client participation $\frac{K}{n} = 0.2$, and number of epochs 200 (100 communication rounds). In this new dataset the classification task is more complex given the increased number of labels.

**Hyperparameters.** We implemented Per-FedAvg and pFedMe based on the code from GitHub,[10]. Other implementations were not available online, so we implemented ourselves. For each of the methods we tuned learning rate in the set $\{0.3, 0.2, 0.15, 0.125, 0.1, 0.075, 0.05\}$ and have a decaying learning schedule such that learning rate is multiplied by 0.99 at each epoch. We use weight decay of $1e-4$. For MNIST and FEMNIST experiments for both personalized and global models we used a 5-layer CNN, the first two layers consist of convolutional layers of filter size $5 \times 5$ with 6 and 16 filters respectively. Then we have 3 fully connected layers of dimension $256 \times 120$, $120 \times 84$, $84 \times 10$ and lastly a softmax operation. For CIFAR-10 experiments we use a similar CNN, the only difference is the first fully connected layer is of dimension $400 \times 120$.

- `AdaPeD`[11]: We fine-tuned $\psi$ in between $0.5 - 5$ with 0.5 increments and set it to 3.5. We set $\eta_3 = 5e-2$. We manually prevent $\psi$ becoming smaller than 0.5 so that local loss does not become dominated by the KD loss. We use $\eta_2 = 0.1$ and $\eta_1 = 0.1$. [12] When taking the derivative with respect to $\psi$ we observed sometimes multiplying the right term (consist of KD loss function) by some constant (5 in our experiments) gives better performance.

- Per-FedAvg Fallah et al. (2020) and pFedMe Dinh et al. (2020): For Per-FedAvg, we used 0.075 as the learning rate and $\alpha = 0.001$. For pFedMe we used the same learning rate schedule for main learning rate, $K = 3$ for the number of local iterations; and we used $\lambda = 0.5$, $\eta = 0.2$.

- QuPeD Ozkara et al. (2021): We choose $\lambda_p = 0.25$, $\eta_1 = 0.1$ and $\eta_3 = 0.1$ as stated.

- Federated Mutual Learning Shen et al. (2020): Since authors do not discuss the hyperparameters in the paper, we used $\alpha = \beta = 0.25$, global model has the same learning schedule as the personalized models.

## K.2    ADDITIONAL EXPERIMENTS

**Convergence plots for `AdaPeD`.**    We put the experimental convergence plots (test accuracy vs no. of iteration) for `AdaPeD` in Figure 2.

---

[10]`https://github.com/CharlieDinh/pFedMe`

[11]For federated experiments we have used PyTorch's Data Distributed Parallel package.

[12]We use `https://github.com/tao-shen/FEMNIST_pytorch` to import FEMNIST dataset.

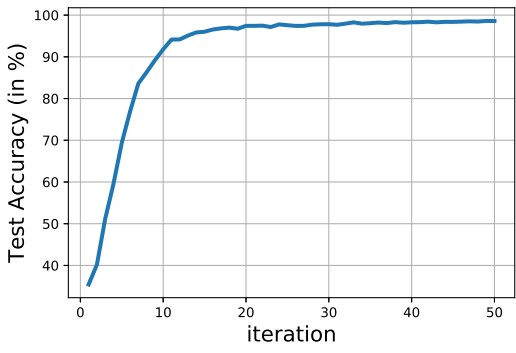
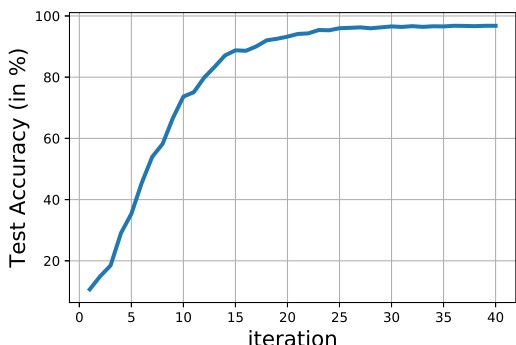

(a) `AdaPeD` Test Accuracy (in %) vs iteration on MNIST with 0.1 sampling ratio.

(b) `AdaPeD` Test Accuracy (in %) vs iteration on FEMNIST with 0.33 sampling ratio.

Figure 2: Convergence plots (test accuracy vs no. of iteration) for `AdaPeD`.

**Personalized estimation: synthetic experiments in Bernoulli setting.** For this setting, for $\mathbb{P}$ we consider three distributions that (Tian et al., 2017) considered: normal, uniform and '3-spike' which have equal weight at 1/4, 1/2, 3/4. Additionally, we consider a Beta prior. We compute squared error of personalized estimators and local estimators ($\frac{Z_i}{n}$) w.r.t. the true $p_i$ and report the average over all clients. We use $m = 10000$ clients and 14 local samples similar to (Tian et al., 2017). Personalized estimator provides a decrease in MSE by $37.1 \pm 3.9\%, 12.0 \pm 1.6\%, 24.3 \pm 2.8\%, 34.0 \pm 3.7\%$, respectively, for each aforementioned population distribution compared to their corresponding local estimators. Furthermore, as theoretically noted, less spread out prior distributions (low data heterogeneity) results in higher MSE gap between personalized and local estimators.

**Linear regression.** For this, we create a setting similar to (Jain et al., 2021a). We set $m = 10,000$, $n = 10$; and sample client true models according to a Gaussian centered at some randomly chosen $\mu$ with variance $\sigma_\theta^2$. We randomly generate design matrices $X_i$ and create $Y_i$ at each client by adding a zero mean Gaussian noise with true variance $\sigma_x^2$ to $X_i\theta_i$. We set true values $\sigma_x^2 = 0.01, \sigma_x^2 = 0.05$ and we sample each component of $\mu$ from a Gaussian with 0 mean and 0.1 standard deviation and each component of $X$ from a Gaussian with 0 mean and variance 0.05, both i.i.d. We measure the average MSE over all clients with and compare personalized and local methods. When $d = 50$, personalized regression has an MSE gain of $8.0 \pm 0.8\%, 14.8 \pm 1.2\%$, and when $d = 100$, $9.2 \pm 1.1\%, 12.3 \pm 2.0\%$ compared to local and FedAvg regression, respectively. Moreover, compared to personalized regression where $\mu, \sigma_\theta, \sigma_x$ are known, alternating algorithm only results in $1\%$ and $4.7\%$ increase in MSE respectively for $d = 50$ and $d = 100$.

**Estimation Experiments.** We provide more results for the estimation setting discussed in Figure 1a. In Figure 3a we have a setting with 1000 clients and 5 local samples and in Figure 3b 500 clients and 5 local samples per client. We observe with as the number of clients increase DP-Personalized Estimator can converge to Personalized Estimator with less privacy budget. We also observe compared to Figure 1a, less number of local samples increases the performance discrepancy between personalized and local estimator.

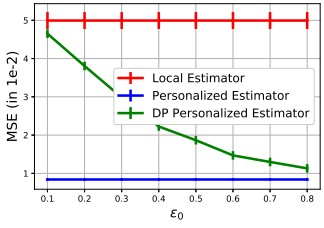
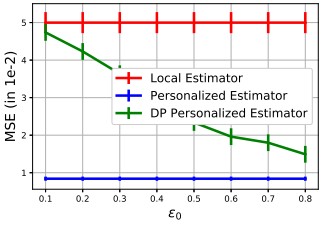

(a) Private Estimation with m=1000, n=5

(b) Private Estimation with m=500, n=5

Figure 3: In Figure 1a, we plot MSE vs. $\epsilon_0$ for personalized estimation with different number of clients, this is the same setting as Figure 1a except the number of clients and local samples.

**Additional Learning Experiments with Different Number of Clients.** We do additional experiments with different number of clients. On FEMNIST we use the same model and same data sample per client as in Section4, number of clients is 30, total number of epochs is 30 and we fix the local iteration to be 40 per epoch, we do full client sampling to simulate a cross-silo environment. As seen in Table 4, AdaPeD continues to outperform the competing methods following the trend in Section 4.

Table 4: Test accuracy (in %) for FEMNIST with $m = 30$ clients.

| Method | FEMNIST |
|---|---|
| FedAvg | $95.91 \pm 0.78$ |
| FedAvg+fine tuning Jiang et al. (2019) | $96.22 \pm 0.57$ |
| AdaPeD (Ours) | $\mathbf{98.10} \pm 0.09$ |
| pFedMe (Dinh et al., 2020) | $96.03 \pm 0.50$ |
| Per-FedAvg (Fallah et al., 2020) | $96.71 \pm 0.14$ |
| QuPeD (FP) (Ozkara et al., 2021) | $97.72 \pm 0.16$ |
| Federated ML (Shen et al., 2020) | $96.80 \pm 0.13$ |

On CIFAR-10 we use the same model as in Section4, and divide the dataset to 30 clients where each client has access to data samples from 4 classes. Total number of epochs is 250 and we fix the local iteration to be 40 per epoch; we set $\frac{K}{n} = 0.2$ and number of local epochs to be 2. AdaPeD outperforms the competing methods in parallel to the experiments in Section4, as can be seen in Table 5.

Table 5: Test accuracy (in %) for CIFAR-10 with $m = 30$ clients.

| Method | CIFAR-10 |
|---|---|
| FedAvg | $53.92 \pm 0.94$ |
| FedAvg+fine tuning Jiang et al. (2019) | $67.44 \pm 1.11$ |
| AdaPeD (Ours) | $\mathbf{73.86} \pm 0.39$ |
| pFedMe (Dinh et al., 2020) | $71.97 \pm 0.09$ |
| Per-FedAvg (Fallah et al., 2020) | $64.09 \pm 0.46$ |
| QuPeD (FP) (Ozkara et al., 2021) | $73.21 \pm 0.44$ |
| Federated ML (Shen et al., 2020) | $72.53 \pm 0.36$ |

**Additional Experiment Implementation Details.**

We use the same strategy as in Appendix K.1 to tune the main learning rates. We use 1e-4 weight decay.

- AdaPeD: We fine-tuned $\psi$ in between $0.5 - 5$ with 0.5 increments and set it to 4 for CIFAR-10/100 and to 3 for FEMNIST. We manually prevent $\psi$ becoming smaller than 1 so that local loss does not become dominated by the KD loss. We use $\eta_2 = 0.075$ and $\eta_1 = 0.075$ for CIFAR-10 and CIFAR-100 and $\eta_2 = 0.1$ and $\eta_1 = 0.1$ for FEMNIST.

- Per-FedAvg (Fallah et al., 2020) and pFedMe (Dinh et al., 2020): For Per-FedAvg, we used 0.1 as the learning rate and $\alpha = 0.0001$. For pFedMe we used the same learning rate schedule for main learning rate, $L = 3$ for the number of local approximation iterations; and we used $\lambda = 0.1$, $\eta = 0.1$.

- QuPeD Ozkara et al. (2021): We set $\lambda_p = 0.25$, $\eta_1 = 0.1$ for local learning rate and $\eta_2 = 0.1$ for global learning rate.

- Federated Mutual Learning Shen et al. (2020): Since authors do not discuss the hyperparameters in the paper, we used $\alpha = \beta = 0.25$.

