# OpenReview forum: "A Statistical Framework for Personalized Federated Learning and Estimation: Theory, Algorithms, and Privacy"
_ICLR.cc/2023/Conference — ICLR 2023 poster_

### Official Review · Reviewer_MwhW · 2022-10-25

**Confidence:** 3
**Clarity, Quality, Novelty And Reproducibility:** I found the paper quite difficult to …
**Correctness:** 3
**Technical Novelty And Significance:** 3
**Empirical Novelty And Significance:** 3
**Recommendation:** 5

**Strength And Weaknesses:**

Strengths:
1) The modeling assumptions of how local model parameters are drawn from a global distribution of parameters seems quite reasonable, albeit a little restrictive.
2) The proofs seemed correct. Although, I am not a theorist, so I wouldn't state this with a lot of certainty.
3) The authors seemed to have given a lot of thought to the practical problems in federated learning setting and have tried to tackle a lot of them systematically.

Weakness:
1) The paper tries to do too much in my opinion, which makes it quite difficult to follow at times.
2) The experiments seem sparse and handpicked, particularly for the personalized learning setting. Several questions that came to my mind are: what is the reason for picking m to be 50 and 66? Why didn't the authors try a range of values and reported all the results?
3) Similarly, for the estimation task, I'd like to see more experiments. This can be done easily with different choices of synthetic parameters.
4) Add more motivation for each estimation and learning setting. When are these problems relevant? Why should we care about them? This is not very clear from the paper.

**Summary Of The Paper:**

The authors propose a statistical framework for personalized learned and estimation in a federated learning setting. The authors first explicitly state the modeling assumptions of how local model parameters are drawn from a global distribution of parameters. They then develop private and non-private estimation and learning algorithms.

**Summary Of The Review:**

While I think the technical contributions of the paper are quite novel, it is very hard to read. This is in large part because the authors have tried to fit too many things into a single paper. For instance, have the authors considered writing this as 2 separate papers on personalized learning and estimation? Alternatively, it might be better suited as a long-form journal paper (say at JMLR or TMLR).

---

> ### Author Response · Authors · 2022-11-17
> **More experiments and motivation for the tasks**
>
> We thank the reviewer for their feedback and positive comments about technical content. We address your questions below point by point.
>
> - **The paper has many results and could be suitable for a journal.**  The paper presented a  statistical framework  leading to new personalized federated estimation and learning algorithms (e.g., AdaMix, AdaPeD); we also incorporated privacy (and communication) constraints into our algorithms and analyzed them. We believe that this a coherent theme for a paper which exposes the common structure for both estimation and learning. We hope that we have explained the core ideas clearly in the main part of the paper; we have given several examples and algorithms arising from it, as well as example numerics. We have proofs, additional details,  and numerics in the Appendices, as seen in several ICLR papers in the past. We therefore believe that this submission is suitable for this venue.
>
> - **The experiments seem sparse, the reason for client numbers.** These client numbers are representative of the performance we have seen with different number of clients. Our resources allowed us to do clients of this numbers, that is why we chose these client numbers; in particular, we had a server with 6 GPU's and could fit 11 clients per GPU hence the number 66 for FEMNIST; similarly for the earlier CIFAR-10 results. We did additional experiments with $m=30$ for FEMNIST and CIFAR-10, see newly added Appendix K.3 for the results. As can be observed from the results, the same trends reported in the earlier number of clients hold with these as well.
>
> - **More experiments for estimation.** Thank you for the feedback, we have done more experiments for estimation in newly added Appendix K.3, where we have chosen additional parameters in terms of number of clients and samples per client.
>
> - **More motivation for each estimation and learning setting.** The natural motivation for the overall setting is to enable personalized federated estimation and learning with limited local data which has heterogeneous statistics. This arises in several applications as described in [1]. The overall statistical framework is fairly general, but the specific choice of population models will be application dependent. In the estimation setting, when the parameters are discrete, (e.g., binary), in general it is a heterogeneous Bernoulli distribution, with a population distribution of the parameters which is supported in [0,1], without any special structure. For example, we used this to predict individual county political tendencies using historical election data in Section 4.  Gaussian distributions are applicable in several scenarios, and work well in several cases including in predicting individual baseball players', as mentioned in response to reviewer t7xg. Mixture models are appropriate for clustered distributions, where groups of users have similar statistics, but there are several groups. We have given methods for both discrete mixtures as well as Gaussian mixtures (inspiring AdaMix). Finally an exponential family on the Kullback-Leibler divergence, as explained in Appendix H.1, yields AdaPeD, which adaptively can combine clients using different learning models locally as it combines output probabilities rather than models directly. As seen in numerics, this performs well on several real data sets. We hope that these examples motivate the different estimation and learning settings. We believe that our introduction does motivate the personalized federated estimation and learning problems, but we will be happy to incorporate specific suggestions you may have given our response.
>
> We hope to have answered your concerns and questions adequately, please let us know if you have any further questions and we will be happy to answer. We would appreciate it if you could consider raising your score in light of our response.
>
> References
>
> [1]Kairouz, Peter, et al. "Advances and open problems in federated learning." Foundations and Trends in Machine Learning 14.1–2 (2021): 1-210.

---

### Official Review · Reviewer_3hXR · 2022-10-26

**Confidence:** 3
**Correctness:** 3
**Technical Novelty And Significance:** 3
**Empirical Novelty And Significance:** 3
**Recommendation:** 6

**Clarity, Quality, Novelty And Reproducibility:**

The paper is written clearly and easy to follow when including the appendix. I feel the paper is more suitable to a journal since a lot of important explanations are set aside in the appendix.
It is better to compare the upper bounds with the best bounds for the method without the privacy guarantee. By such a comparison, we can see the tightness of the present bounds roughly.

**Strength And Weaknesses:**

I think this paper is strong. It studies an important yet remarkably understudied problem that can build the bridge between Bayesian learning and federated learning. It gives two types of algorithms: AdaPeD and AdaMix. The algorithm and the theoretical results are sound. The algorithm is fairly natural in Bayesian perspective, which is to update the Gaussian posterior based on the prior and likelihood. The AdaPeD uses a knowledge distillation regularization. It shows we can federatedly train the model via MAP well when preserving user-level privacy.

**Summary Of The Paper:**

In this work, the authors developed a new Bayesian framework for personalized federated learning and derived theoretical bounds and novel algorithms for private personalized estimation. AdaPeD was proposed to use information divergence constraints along with adaptive weighting of local models and population models. AdaMix was proposed to adaptively weigh multiple global models and combines them with local data for personalization. Certain privacy analyses of new private personalized learning algorithms were done.

**Summary Of The Review:**

The paper is clearly written, but the notations are heavy. The contribution is very technical from the perspective of Bayesian Federated Learning.

---

> ### Author Response · Authors · 2022-11-17
> **Response**
>
> We thank the reviewer for their feedback and positive comments. We respond to your concerns below.
>
> - **Paper is more suitable for a journal.** The paper presented a  statistical framework  leading to new personalized federated estimation and learning algorithms (e.g., AdaMix, AdaPeD); we also incorporated privacy (and communication) constraints into our algorithms and analyzed them. We believe that this a coherent theme for a paper that exposes the common structure for both estimation and learning. We hope that we have explained the core ideas clearly in the main part of the paper; we have given several examples and algorithms arising from it, as well as example numerics. We have proofs, additional details,  and numerics in the Appendices, as seen in several ICLR papers in the past. We therefore believe that this submission is suitable for this venue.
>
> - **Tightness of the bound.** Thank you for asking this question. For the Gaussian case our upper bound matches the minmax lower bound so it is tight (see Remark 2). For the private case, there is a trade-off between the estimation error and privacy (see Theorem 2). When $\epsilon$ is very small (i.e., high privacy), each client depends more on the local estimator, where the parameter $a$ in Theorem 2 goes to one as $\epsilon\to 0$. However, when $\epsilon\to\infty$ (low privacy regime), the estimation error matches with that of the non-private case.
>
> We hope to have answered your concerns and questions adequately, please let us know if you have any further questions and we will be happy to answer. We would appreciate it if you could consider raising your score in light of our response.

---

### Official Review · Reviewer_zS9y · 2022-10-28

**Confidence:** 3
**Correctness:** 3
**Technical Novelty And Significance:** 3
**Empirical Novelty And Significance:** 3
**Recommendation:** 6

**Clarity, Quality, Novelty And Reproducibility:**

The ideas raised in the paper are original, and the paper is well-written and easy to read.

**Strength And Weaknesses:**

Strengths:
* The paper is theoretically grounded. Authors provide theoretical results for different population models.
* In contrast to many competitors, the paper considers privacy aspects of the problem and provides new algorithms with provable privacy guarantees

Weaknesses:
* "Based on (Xi, Agg(q1, . . . , qm)), client i outputs an estimate \hat{\theta_i} of \theta. " Should be the estimate of \theta_i ? Otherwise not clear what is \theta as it is not defined before.
* It seems that there is a missing opportunity to compare with another class of personalized models. Namely [1], when there is a split of shared and personalized parameters. Moreover, this model could be considered as a particular case of your statistical framework, when (for Gaussian population model) \sigma_\theta = [0,\infty] where 0 corresponds to the shared (common) part of parameters, and \sigma_x = 0. It is worth mentioning the paper in related work and comparing the results.
* The choice of datasets for experiments seems a bit low-key. I suggest considering CIFAR100.

Minor issues:
* page 6 first appearance of FedAvg — no link
*"... given 6 election data we did 1-fold cross-validation. " - what does 1-fold cross-validation mean? Did not you mean 6-fold cross-validation?


[1] Collins L. et al. Exploiting shared representations for personalized federated learning //International Conference on Machine Learning. – PMLR, 2021. – С. 2089-2099.


**Summary Of The Paper:**

In this paper, authors develop a statistical framework for personalized federated learning, in which they study different choices of population distribution. They provide theoretical bounds and novel algorithms for private personalized estimation, design, and conduct privacy analysis of new private personalized learning algorithms.

**Summary Of The Review:**

I am generally positive about the paper ideas, but there are some issues that are worth correcting (see above).

---

> ### Author Response · Authors · 2022-11-17
> **Fixed typos, discussion on Collins et al., experiments on CIFAR-100**
>
> We thank the reviewer for their feedback and positive comments. We address your questions below point by point.
>
> - **"Based on (Xi, Agg(q1, . . . , qm)), client i outputs an estimate $\hat{\theta_i}$ of $\theta$. " Should be the estimate of $\theta_i$? FedAvg citation, typo in 6-fold cross-validation.** Thank you pointing out the typos, we have fixed these in the revised draft where the edited parts are colored in blue; see pages 3, 6, 8.
>
> - **Comparison to Collins et al.** Thank you for the suggestion, and the reference that we were not aware of. On a quick read of the reference provided, it seems to build on the ``common representation'' framework advocated  in our earlier references, Du et al. ICLR 2021, Jain et al. NeurIPS 2021, etc., where they studied a linear version of the problem; in fact they seem to build on the work of  Du et al. ICLR 2021, which we referenced. From that viewpoint we  had related our framework to such (linear) common representation approaches (see Appendix J). Given  your suggestion we added a short discussion on Collins et al. in Appendix J of the revision.
>
> - **Experiments on CIFAR-100.** Thank you for the suggestion, we have added experiments on CIFAR-100, see Appendix K.3 for the results.  In this new dataset the classification task is more complex given the increased number of labels, still, AdaPeD outperforms the competing methods as in earlier experiments.
>
> We hope to have answered your concerns and questions adequately, please let us know if you have any further questions and we will be happy to answer. We would appreciate it if you could consider raising your score in light of our response.

---

> > ### Comment · Reviewer_zS9y · 2022-12-13
> > **Thanks for the answers**
> >
> > Dear authors,
> >
> > thanks for your comments and additional experiments. I tend to keep my score at the moment. I might adjust it later following the discussion with other reviewers.

---

### Official Review · Reviewer_t7xg · 2022-11-03

**Confidence:** 4
**Correctness:** 3
**Technical Novelty And Significance:** 3
**Empirical Novelty And Significance:** 3
**Recommendation:** 6

**Clarity, Quality, Novelty And Reproducibility:**

The paper presents an overall well investigated research work. To enhance clarity the authors need to answer the following questions:

Questions:

What does it mean statistically to “estimate i through the help of the server.”?

In the section ‘Personalized Estimation’, should it be “client i outputs an estimate i of i” instead?

Does the prior distribution necessarily need to have a density in general? How ‘realistic’ are the special case assumptions of Gaussian and Bernoulli as global population distributions?

In the statement of Theorem 2., should it be “[-r,r]d” instead?

Does the notation Ɛ mean expectation [In Theorem 2, 3, etc.]? If so, kindly maintain any one symbol throughout.
The authors also use the same notation for a different purpose in Section B.2.1.

Shouldn’t we have the sum also over the quantity on the right-hand side of equation (10) [Section 3.1]?

The quantity it may penalize the heterogeneity, but does not denote the variance. The authors should call it something else instead [Section 3.2, last paragraph].

Is the strict imposition of the value ‘1’ necessary in the definition of ‘neighborhood’ [Section A.3], since there is a clear possibility to generalize the result even if two datasets D and D' differ at multiple terms?

In Theorem 2, the upper bound on MSE  (5) loses its worth in a higher-dimensional setup. Can the authors talk about any remedy to the same?


**Strength And Weaknesses:**

Strength:

The language of the article is lucid, and the presentation is also of good quality. The discussion leading up to the theoretical analyses and the algorithms is precise. I find the statistical analysis rigorous and very well represented. Prior works and relevant references are well-placed throughout the paper.

Weakness/Issues:

The authors have altered the standard structure of the article, as guided by ICLR instructions. The abstract should not be full page-wide. This is a violation of the code and gives them an undue advantage over others.

The current article looks incomplete, lacking a ‘Conclusion’ section. Also, sufficient discussion regarding limitations and future work is missing.

I suggest the authors present accompanying codes maintaining anonymity.

It would be very helpful if the problem statement is presented more precisely in the introduction. The authors provide a lot of references to prior work. However, amidst such a crowd, the motivation somehow fades.

As I have acknowledged, the discussion is quite rigorous. However, the is significant room for improvement when it comes to organization.

The empirical results seem insufficient, and I suggest the authors put more datasets to test if feasible.


**Summary Of The Paper:**

The paper proposes algorithms that search for suitable personalized models in a client-server type federated learning setup. The algorithms are inspired by the classical theory of parametric Bayesian risk minimization. In the personalized parameter estimation regime, the authors assume two population distributions: Gaussian and Bernoulli. Under Gaussian distribution, they report that in case the parent population becomes degenerate (i.e., variation tending to nullity), the global average turns out to be the ‘best’ estimator. Moreover, the posterior personalized mean estimator in this setup also turns out to be optimal in general. If the parent population follows a Bernoulli law, having sufficient observations from local sub-populations suggests against collaborating. The following ‘personalization’ algorithms utilize different prior distributions and regularization schemes to ensure client privacy.

**Summary Of The Review:**

The paper may be considered for acceptance provided the authors address the above listed concerns.

---

> ### Author Response · Authors · 2022-11-17
> **Response 2/2**
>
> - **How ‘realistic’ are the special case assumptions of Gaussian and Bernoulli as global population distributions?** For the Bernoulli (binary) setting actually we do not restrict the population distribution, as long as it is a distribution whose support is in [0,1]. Our model, Beta distribution is a good approximation for many cases, particularly for unimodal or bimodal distributions. For example, we used this to predict individual county political tendencies using historical election data in Section 4. Using a Gaussian model for population distribution works well in several applications. As an example for Gaussian case, Efron [2, page 8] provides a baseball statistics example where such modeling can give good predictions for individual players. Eventually choice of population models will be application dependent, but we advocate a parametric family of models for computational ease, in federated estimation and learning. For the Gaussian setting, parametrization by a population mean vector is useful: since a client has limited amount of data it is better to do a point estimate of mean/variance. Our statistical framework extends to non-parametric priors, but its estimation may require use of MCMC methods (or other non-parametric estimation algorithms). However this would be computationally expensive and usually not feasible in FL settings as we also explain in the paper. As mentioned, in the paper we have investigated population models other that Gaussian and Bernoulli. For example, we examined discrete mixture models as mentioned above (Appendix D for estimation, Appendix G for learning). The Gaussian mixture population model inspires the AdaMix algorithm. Mixture distributions model clustered user data distributions, and are widely used in many applications. The population model inspiring  AdaPeD is from an exponential family on the Kullback-Leibler divergence, as explained in Appendix H.1. In this case, the model relates  output probabilities, corresponding to  classification tasks. As seen in numerics, this performs well on several real data sets.
>
> We hope to have answered your concerns and questions adequately, please let us know if you have any further questions and we will be happy to answer. We would appreciate it if you could consider raising your score in light of our response.\\
>
> References
>
> [1] Efron, Bradley. Large-Scale Inference: Empirical Bayes Methods for Estimation, Testing, and Prediction (Institute of Mathematical Statistics Monographs) (2010). Cambridge: Cambridge University Press.
>
> [2] Dwork, Cynthia, and Aaron Roth. ``The algorithmic foundations of differential privacy." Foundations and Trends® in Theoretical Computer Science 9.3–4 (2014): 211-407.
>
> [3] Zhang, Yuchen, et al. ``Information-theoretic lower bounds for distributed statistical estimation with communication constraints." Advances in Neural Information Processing Systems 26 (2013).

---

> ### Author Response · Authors · 2022-11-17
> **Response 1/2**
>
> We thank the reviewer for their feedback and positive comments on presentation, language and content. Below we answer the concerns and questions point by point.
>
> - **Revised Version.** Thank you for your close reading of our work. We fixed the typos/issues and uploaded a revision to the paper, we wrote the newly added parts in blue. In particular, we fixed the typos in response to the following questions:
>
>   - ``In the section ‘Personalized Estimation’, should it be “client i outputs an estimate i of i” instead?'' - Indeed it should be \theta_i , see page 3 of the revision.
>   -  ``In the statement of Theorem 2., should it be “[-r,r]d” instead?'' - Indeed it should be [-r,r]^d, see page 4 of the revision.
>   -   ``Does the notation epsilon mean expectation [In Theorem 2, 3, etc.]? If so, kindly maintain any one symbol throughout. The authors also use the same notation for a different purpose in Section B.2.1.'' - Yes it means expectation, we have made the notation consistent in the revision.
>   -   ``Shouldn’t we have the sum also over the quantity on the right-hand side of equation (10) [Section 3.1]?'' Yes, we have corrected this typo in the revision, see page 6.
>   -  ``The quantity it may penalize the heterogeneity, but does not denote the variance. The authors should call it something else instead [Section 3.2, last paragraph].'' We have changed the terminology to `dissimilarity term' to avoid confusion, see page 7.
>
> - We added a Conclusion part for the paper (including some open questions & future directions) in page 9, and in order to fit that into the 9 pages, we made minor changes in the text (indicated in blue) throughout the paper. Moreover, in the introduction we add more pointers to precise statements in later sections, if you have additional specific suggestions about this aspect, we are happy to incorporate.
>
> - **Discussion is rigorous but organization needs to be improved.** Thank you appreciating the rigor of our discussion, we would like to ask whether your concern remains after the revision. If that is the case, we are happy to incorporate your additional suggestions to further improve the organization.
>
> - **Experiments on more datasets.** We have added experiments on Cifar-100, please see Appendix K.3. In this new dataset the classification task is more complex given the increased number of labels, still, AdaPeD outperforms the competing methods as in earlier experiments.
>
> - **What does it mean statistically to “estimate i through the help of the server.”?** It means that each client computes a personalized estimate with the help of a central server that intermediates between clients. This is the distributed architecture in contrast to the decentralized (peer to peer) architecture, which does not have any central server to help co-ordinate.
>
> - **Does the prior distribution necessarily need to have a density in general?** No, there are no restriction on prior/population distribution to have density; in fact, we have analyzed the case of discrete mixture model population distribution, for estimation it can be found in Appendix D and for learning it can be found in Appendix G. We are happy to explain this further if needed.
>
> - **Is the strict imposition of the value ‘1’ necessary in the definition of ‘neighborhood’ [Section A.3], since there is a clear possibility to generalize the result even if two datasets D and D' differ at multiple terms?** Thank you for your comment. We follow the standard definition of differential privacy (DP), where two datasets are neighboring if they differ in a single item. However, this definition covers the general case (when datasets differ in multiple items) using the group property of the DP [Theorem 2.2,2]: Any $\left(\epsilon,\delta\right)$-DP mechanism is $\left(k\epsilon,k\delta\right)$-DP for group of size $k$ (i.e., datasets differ in $k$ items). This can be proven directly by repeating the DP definition $k$ times.
>
> - **In Theorem 2, the upper bound on MSE (5) loses its worth in a higher-dimensional setup. Can the authors talk about any remedy to the same?** In the case where the samples are Gaussian random vectors in $d$ dimensions, fundamentally there is a linear dependence on $d$ for mean estimation as it is minimax optimal. This is established information theoretically, see for example [3] which shows a lower bound for estimation error and discusses its dependency on $d$. We are happy to explain further if needed.

---

### Author Response · Authors · 2022-11-29
**Follow up**

Dear Reviewers,

We hope that we have clarified all the questions in the initial review. In addition to our clarifications, we have revised the paper and also added numerics on other data sets as well as synthetic data, in the revision. We had not seen any further comments by the reviewers. If you had any further questions, please let us know. We would appreciate it if you could consider increasing your evaluation scores, given our responses.

---

### Decision · Program_Chairs · 2023-01-20

**Decision:**

Accept: poster

**Justification For Why Not Higher Score:**

The audience for this paper may not be very wide.


**Justification For Why Not Lower Score:**

It addresses an important topic from a well-grounded theoretical perspective, both in terms of estimation and privacy.

**Metareview: Summary, Strengths And Weaknesses:**

This paper presents privacy-preserving empirical and hierarchical Bayes algorithms. The analysis and development are both fine, and the topic is interesting, at least for me, relative to the standard 'propagate gradients' idea of most federated learning algorithms. The only complaint was that more experiments might be necessary. However, it is my feeling that the paper is good enough as is.


**Note From Pc:**

if the above contains the word "oral" or "spotlight" please see: "oral" presentation means -> notable-top-5% and "spotlight" means -> notable-top-25%. As stated in our emails, we are disassociating presentation type from AC recommendations